# Contextual Gaussian Process Bandits with Neural Networks

**Haoting Zhang   Jinghai He   Rhonda Righter   Zuo-Jun Max Shen   Zeyu Zheng**
Department of Industrial Engineering & Operations Research
University of California, Berkeley
Berkeley, CA 94720
`haoting_zhang,jinghai_he,rrighter,maxshen,zyzheng@berkeley.edu`

## Abstract

Contextual decision-making problems have witnessed extensive applications in various fields such as online content recommendation, personalized healthcare, and autonomous vehicles, where a core practical challenge is to select a suitable surrogate model for capturing unknown complicated reward functions. It is often the case that both high approximation accuracy and explicit uncertainty quantification are desired. In this work, we propose a neural network-accompanied Gaussian process (NN-AGP) model, which leverages neural networks to approximate the unknown and potentially complicated reward function regarding the contextual variable, and maintains a Gaussian process metamodel with respect to the decision variable. Our model is shown to outperform existing approaches by offering better approximation accuracy thanks to the use of neural networks and possessing explicit uncertainty quantification from the Gaussian process. We also analyze the maximum information gain of the NN-AGP model and prove regret bounds for the corresponding algorithms. Moreover, we conduct experiments on both synthetic and practical problems, illustrating the effectiveness of our approach.

## 1   Introduction

Various applications, including online content recommendation [1], healthcare [37, 15, 36], and autonomous vehicles [7], demand the sequential selection of a decision variable, conditional on the observed contextual variable representing the state of the environment in each round. These applications can generally be framed as contextual bandit problems [5, 59, 63, 2], especially when the reward function associated with each pair of decision and contextual variables is unknown. When both the decision and contextual variables are drawn from continuous sets, a significant challenge is selecting an appropriate surrogate model to approximate the reward function, considering both approximation accuracy and uncertainty quantification. A common approach to alleviate this issue is to employ a Gaussian process (GP) to model the reward function [61, 46], yielding the GP bandit method [80, 81]. Indeed, GP has proven to be an effective surrogate model to address the exploration-exploitation trade-off in estimating the unknown function while optimizing over it; see [88, 75, 90, 43, 51]. On the other hand, most of the existing GP bandit literature does not take the exogenous contextual variable into consideration, despite its critical role in capturing effects beyond the decision variable that influence the reward – effects that are integral to many of the applications previously mentioned [50, 62]. When the contextual variable is included in GP bandit problems, previous work largely adopts a GP to jointly model the reward function with both contextual and decision variables, employing a composite kernel that is either the sum or product of two separate kernels; see [57, 9].

While GP-based bandit methods have proven effective in various applications [3, 8, 92, 93, 86, 6], they may fall short in scenarios where the reward function exhibits intricate dependence on complex

37th Conference on Neural Information Processing Systems (NeurIPS 2023).

contextual variables, for example, time-varying rewards [13, 31] and graph-structured contextual variables [71]. Specifically, it is a challenge to pre-define an appropriate composite kernel function for the joint GP, which is critical for the performance of the corresponding bandit algorithms, as documented by [21, 82]. Neural networks (NN), on the other hand, have been utilized elsewhere as surrogate models for the reward function in bandit problems [16, 54, 55], thanks to their flexibility and strong approximation power. However, they bring their own set of challenges. The "black-box" nature of the neural network hinders explicit uncertainty quantification and complicates theoretical analysis of the associated algorithms. In particular, the acquisition functions guiding point selection in these algorithms necessitate an approximation of uncertainty [99, 54]. Although this approximation tends to be accurate when the NN's width is large, this can also lead to overparameterization.

**Contribution.** This paper proposes a *neural network-accompanied Gaussian process* (NN-AGP) model for solving contextual bandit problems, especially when the reward functions have intricate dependence on complex contextual variables. The proposed model is an inner product of a neural network and a multi-output GP, where the neural network captures the dependence of the reward function on the contextual variables, and the GP is employed to model the mapping from the decision variable to the reward. Our model generates a joint GP with both contextual and decision variables, which outperforms the existing GP-based bandit methods by specifying the data-driven kernel function through the lens of neural networks, thereby leading to an accurate approximation for the reward function. Moreover, compared with entirely relying on NN's, our model stands out due to the explicit GP expression with respect to the decision variable. This feature enables bandit algorithms with NN-AGP to be implemented efficiently with existing GP-based acquisition functions and provides a theoretical guarantee of the regret bounds. Our main contributions can be summarized as follows:

1. We propose an NN-AGP model and its upper confidence bound (UCB) algorithm for contextual bandit problems, referred to as NN-AGP-UCB. Our algorithm offers a data-driven procedure to specify the kernel functions of the joint GP, thereby achieving superior accuracy and model flexibility. We also prove the upper bound for both the maximum information gain of the NN-AGP model and the regret of the NN-AGP-UCB algorithm.

2. We conduct the experiments to evaluate our approach for complex reward functions, including those with time-varying dependence on sequenced and graph-structured contextual variables. Experimental results demonstrate the superiority of our approach over existing approaches that entirely rely on either GP or NN.

## 1.1 Related work

Since the seminal work by [80], the Gaussian process (GP) bandit problem has been extensively studied, where the bandit feedback is modeled as a GP regarding the decision variable (arms to be pulled). Some recent work includes [35, 32, 13, 12, 11, 18, 65]. In addition, GP bandits are also related to Bayesian optimization (BO) problems [38, 19, 94, 39, 34, 28, 85, 52, 23], where both problems consider optimizing black-box functions and therefore require surrogate models (GP in particular). When the number of decision variables is finite, the black-box optimization problem is also known as Ranking & Selection (R&S) [44, 73, 70, 4, 87], where GP models are widely employed as well; see [20, 58, 79, 64]. Our NN-AGP model can also be employed in (contextual) BO or R&S, but the discussion is beyond the scope of this work.

In this work, we specifically take the exogenous contextual variables into consideration. Previous work employs multiplicative and additive kernels to incorporate continuous context spaces into the scalar GP; see [57]. Other work considers safe contextual Bayesian optimization, employing a similar strategy to construct composite kernels; see [41, 9]. Another line of research explores distributionally robust BO [56, 83, 53], where the contextual variable distribution is selected from an ambiguity set. The methodology of representing the objective function using a joint GP has also been widely used in contextual policy search; see [72, 21, 40].

The connection between GP and NN has been explored in [60, 68], documenting that NN's with infinite width approach a GP model when the weight parameters are assigned with Gaussian priors. In addition, deep GP's have been proposed to enhance the model flexibility of neural networks where variational inference is employed; see [27, 26, 91]. GP models in which the parameters are represented by neural networks are studied in [98, 97].

## 2  Main procedure

We consider a problem of sequentially selecting a system's input variable (decision variable) for $T$ (not necessarily known a priori) rounds. In each round, we receive a contextual variable $\boldsymbol{\theta}_t \in \Theta \subset \mathbb{R}^{d'}$ from a set $\Theta$, and select a decision variable $\mathbf{x}_t \in \mathcal{X} \subset \mathbb{R}^d$ from a set $\mathcal{X}$ of decisions. We then receive an observation

$$y_t = f(\mathbf{x}_t; \boldsymbol{\theta}_t) + \epsilon_t,$$

where $f(\mathbf{x}; \boldsymbol{\theta})$ is the reward (objective) function and $\epsilon_t \overset{i.i.d.}{\sim} \mathcal{N}(0, \sigma_\epsilon^2)$ denotes the noise that is independent of both the contextual variables and decision variables. We consider the scenarios when both $\Theta$ and $\mathcal{X}$ are continuous and $\boldsymbol{\theta}_t$'s are fully exogenous. That is, the selection of the decision variable in each round does not influence the future contextual variables. Although we focus on unconstrained problems in this work, our proposed model can be employed to approximate the constraints in optimization problems as well; see [9]. We also note that, for the description and discussion of our approach, the contextual variable is represented by a vector. However, we show through experiments in Section 4 that our approach is applicable to contextual variables with other structures.

Since $f(\mathbf{x}, \boldsymbol{\theta})$ is unknown, we will not generally be able to choose the optimal action, and will thus incur regret $r_t = \sup_{\mathbf{x}' \in \mathcal{X}} f(\mathbf{x}', \boldsymbol{\theta}_t) - f(\mathbf{x}_t, \boldsymbol{\theta}_t)$, indicating the difference between the optimal reward and the reward we actually receive in each round. After $T$ rounds, the cumulative regret is $\mathcal{R}_T = \sum_{t=1}^{T} r_t$. Our goal is to develop an algorithm that achieves sub-linear contextual regret, i.e., $\mathcal{R}_T/T \to 0$ for $T \to \infty$, which requires a statistical model of the reward function with respect to both context variables and decision variables.

We specifically consider a reward function in the form

$$f(\mathbf{x}; \boldsymbol{\theta}) = \boldsymbol{g}(\boldsymbol{\theta})^\top \mathbf{p}(\mathbf{x}), \tag{1}$$

where $\boldsymbol{g}(\boldsymbol{\theta})$ and $\mathbf{p}(\mathbf{x})$ are both $m$-dimensional vector-valued (unknown) functions, and $m \in \mathbb{N}$ is a user-selected quantity to indicate the complexity of the function. There are two main reasons for considering this reward function. First, this formulation is a generalization of linear contextual bandits, where the reward function is the inner product of the contextual variables and the unknown parameters. Here, we assume that the inner product is taken with two vector-valued functions with respect to decision variables and the contextual variable, which is similar in spirit to [24, 96, 42, 100]. Second, this reward function is consistent with the tensor-product approximation of a general function; see [29, 30, 47]. Therefore, further analysis on model mis-specification of the reward function can be supported by existing results of tensor-product approximation.

In this work, we specifically assume that $\boldsymbol{g}(\boldsymbol{\theta})$ is a vector-valued deterministic mapping from $\mathbb{R}^{d'}$ to $\mathbb{R}^m$, represented by a neural network with a given structure and some weight parameter $\mathbf{W}$. In addition, $\mathbf{p}(\mathbf{x})$ is a multi-output Gaussian process (MGP) defined on $\mathcal{X} \subset \mathbb{R}^d$. The MGP model is a generalization of the scalar-valued GP, where the output $\mathbf{p}(\mathbf{x})$ at each $\mathbf{x}$ is an $m$-dimensional vector. The MGP model captures not only the dependence between two outputs but also the dependence between different entries of each output. Thus, the covariance of an MGP $\mathbf{p}(\mathbf{x})$ is represented by a matrix-valued covariance function, denoted by $\mathcal{K}(\mathbf{x}, \mathbf{x}')$, and the vector of parameters involved in the MGP is denoted by $\Phi$. We postpone the detailed description of the NN-AGP model to Section 3.1 and conclude our brief introduction of NN-AGP with an associated proposition, which follows easily from the fact that the normal distribution is preserved under linear transformations.

**Proposition 1.** *The reward function $f(\mathbf{x}; \boldsymbol{\theta})$ is a scalar-valued mean-zero Gaussian process with respect to $\mathbf{x}$ and $\boldsymbol{\theta}$. The kernel function of this Gaussian process is*

$$\tilde{K}((\mathbf{x}, \boldsymbol{\theta}), (\mathbf{x}', \boldsymbol{\theta}')) = \boldsymbol{g}(\boldsymbol{\theta})^\top \mathcal{K}(\mathbf{x}, \mathbf{x}') \boldsymbol{g}(\boldsymbol{\theta}'),$$

*where $\mathcal{K}(\mathbf{x}, \mathbf{x}')$ is the covariance of the MGP.*

Next, we provide a bandit algorithm with the NN-AGP model, during which the surrogate model is sequentially learned from data. We name the algorithm *neural network-accompanied Gaussian process upper confidence bound* (NN-AGP-UCB). Suppose we are now in round $t$ and observe the contextual variable $\boldsymbol{\theta}_t$. In addition, we also have the historic data $\mathcal{D}_{t-1} = \{(\boldsymbol{\theta}_1, \mathbf{x}_1, y_1), (\boldsymbol{\theta}_2, \mathbf{x}_2, y_2), \dots, (\boldsymbol{\theta}_{t-1}, \mathbf{x}_{t-1}, y_{t-1})\}$ in hand. Denote by $\mathbf{y}_{t-1} = (y_1, y_2, \dots, y_{t-1})$

the vector of observations. The selection of the next decision variable $\mathbf{x}_t$ depends on the posterior distribution of the reward function, which is $f(\mathbf{x}; \boldsymbol{\theta}_t) \mid \mathcal{D}_{t-1} \sim \mathcal{N}\left(\mu_{t-1}(\mathbf{x}; \boldsymbol{\theta}_t), \sigma_{t-1}^2(\mathbf{x}; \boldsymbol{\theta}_t)\right).$ Here

$$
\begin{aligned}
\mu_{t-1}(\mathbf{x}; \boldsymbol{\theta}_t) &= \tilde{\mathcal{K}}_{(\mathbf{x}; \boldsymbol{\theta}_t)}^{\top} \left[\tilde{\mathcal{K}}_{\mathcal{D}_{t-1}} + \sigma_\epsilon^2 I_{t-1}\right]^{-1} \mathbf{y}_{t-1}, \\
\sigma_{t-1}^2(\mathbf{x}; \boldsymbol{\theta}_t) &= \boldsymbol{g}(\boldsymbol{\theta}_t)^{\top} \mathcal{K}(\mathbf{x}, \mathbf{x}) \boldsymbol{g}(\boldsymbol{\theta}_t) - \tilde{\mathcal{K}}_{(\mathbf{x}; \boldsymbol{\theta}_t)}^{\top} \left[\tilde{\mathcal{K}}_{\mathcal{D}_{t-1}} + \sigma_\epsilon^2 I_{t-1}\right]^{-1} \tilde{\mathcal{K}}_{(\mathbf{x}; \boldsymbol{\theta}_t)},
\end{aligned}
\tag{2}
$$

where $\tilde{\mathcal{K}}_{(\mathbf{x}; \boldsymbol{\theta}_t)}$ denotes the covariance vector between $f(\mathbf{x}; \boldsymbol{\theta}_t)$ and $\{f(\mathbf{x}_\tau; \boldsymbol{\theta}_\tau)\}_{\tau=1}^{t-1}$. In addition, for the $(t-1) \times (t-1)$-dimensional covariance matrix for historical data $\tilde{\mathcal{K}}_{\mathcal{D}_{t-1}}$, the $(i, j)$-th entry is $\boldsymbol{g}(\boldsymbol{\theta}_i)^{\top} \mathcal{K}(\mathbf{x}_i, \mathbf{x}_j) \boldsymbol{g}(\boldsymbol{\theta}_j)$ as in **Proposition 1**. The required parameters in (2), including the weight parameters $\mathbf{W}$ of the neural network $\boldsymbol{g}(\boldsymbol{\theta})$, the parameters $\Phi$ involved in the MGP $\mathbf{p}(\mathbf{x})$ and the variance $\sigma_\epsilon^2$ of the noise $\epsilon_t$, are all learned and updated with the observations through (5), which we will discuss in Section 3.1.

In terms of the acquisition function (the function that decides the decision variable in the following iteration), we employ the contextual Gaussian process-upper confidence bound (CGP-UCB) introduced in [57]. That is, we decide $\mathbf{x}_t$ as

$$
\mathbf{x}_t = \arg\max_{\mathbf{x} \in \mathcal{X}} \left\{\mu_{t-1}(\mathbf{x}; \boldsymbol{\theta}_t) + \beta_t^{1/2} \sigma_{t-1}(\mathbf{x}; \boldsymbol{\theta}_t)\right\},
\tag{3}
$$

where $\mu_{t-1}(\mathbf{x}; \boldsymbol{\theta}_t)$ and $\sigma_{t-1}(\mathbf{x}; \boldsymbol{\theta}_t)$ are the posterior mean and standard deviation of $f(\mathbf{x}; \boldsymbol{\theta}_t)$ as calculated in (2). The optimization problem (3) can be solved efficiently by global search heuristics, as suggested in [17]. In addition, $\beta_t$ is a user-selected hyper-parameter in each round, addressing the exploration-exploitation trade-off; see discussions in Section 3.2. The procedure for NN-AGP-UCB is summarized in **Algorithm 1**. We note that other commonly selected acquisition functions for GP bandit problems or Bayesian optimization can be employed with NN-AGP as well, including knowledge gradient [76, 89, 33] and Thompson sampling [25, 74]. We postpone the discussion of these acquisition functions to the supplements.

---

**Algorithm 1** NN-AGP-UCB

---

**Input:** Initial values of $\left(\mathbf{W}, \Phi, \sigma_\epsilon^2\right)$;
**for** $t = 1, 2, \ldots, T$ **do**
  Observe the contextual variable $\boldsymbol{\theta}_t$;
  Choose $\mathbf{x}_t = \arg\max_{\mathbf{x} \in \mathcal{X}} \left\{\mu_{t-1}(\mathbf{x}; \boldsymbol{\theta}_t) + \beta_t^{1/2} \sigma_{t-1}(\mathbf{x}; \boldsymbol{\theta}_t)\right\}$;
  Sample $y_t$ at $(\boldsymbol{\theta}_t, \mathbf{x}_t)$;
  Update $\left(\hat{\mathbf{W}}_t, \hat{\Phi}_t, \hat{\sigma}_{\epsilon;t}^2\right)$ as in (5) ;
**end for**

---

## 3 Statistical properties

### 3.1 Specification of NN-AGP

We describe the NN-AGP model here, which is employed as the surrogate for the reward function. Recall that the reward function with the pair of contextual and decision variables $(\boldsymbol{\theta}, \mathbf{x})$ is $f(\mathbf{x}; \boldsymbol{\theta}) = \boldsymbol{g}(\boldsymbol{\theta})^{\top} \mathbf{p}(\mathbf{x})$, where $\boldsymbol{g}(\boldsymbol{\theta})$ is a vector-valued neural network (with weight parameters $\mathbf{W}$) from $\mathbb{R}^{d'}$ to $\mathbb{R}^m$ and $\mathbf{p}(\mathbf{x})$ is an $m$-dimensional output Gaussian process defined on $\mathcal{X} \subset \mathbb{R}^d$. To be more specific, the $m$ outputs $\mathbf{p} = (\mathbf{p}_1, \ldots, \mathbf{p}_m)^{\top}$ are assumed to follow a multi-output Gaussian process (MGP) as

$$
\mathbf{p}(\mathbf{x}) \sim \mathcal{MGP}\left(\mathbf{0}, \mathcal{K}(\mathbf{x}, \mathbf{x}')\right).
$$

Here, $\mathcal{K}(\mathbf{x}, \mathbf{x}')$ denotes the covariance matrix of $\mathbf{p}(\mathbf{x})$ and $\mathbf{p}(\mathbf{x}')$, defined as $\mathcal{K}(\mathbf{x}, \mathbf{x}') \doteq \begin{pmatrix} K_{11}(\mathbf{x}, \mathbf{x}') & \cdots & K_{1m}(\mathbf{x}, \mathbf{x}') \\ \vdots & \ddots & \vdots \\ K_{m1}(\mathbf{x}, \mathbf{x}') & \cdots & K_{mm}(\mathbf{x}, \mathbf{x}') \end{pmatrix}$ which is positive and semi-definite. The $(l, l')$-th entry $K_{ll'}(\mathbf{x}, \mathbf{x}')$ represents the covariance (similarity) between outputs $\mathbf{p}_l(\mathbf{x})$ and $\mathbf{p}_{l'}(\mathbf{x}')$. The NN-AGP

model results in a scalar-valued Gaussian process with both the contextual and decision variables and therefore facilitates explicit acquisition functions and theoretical analysis. This GP representation arises from the linear structure between the MGP and the mapping $g(\boldsymbol{\theta})$.

To specify the covariance matrix, we adopt the collaborative multi-output Gaussian process model [69] as a representative, which generalizes the commonly-used linear model of coregionalization (LMC) and the intrinsic coregionalization model (ICM); see [67]. That is, the MGP is determined by the linear transformation of multiple independent scalar-valued Gaussian processes as

$$\mathbf{p}_l(\mathbf{x}) = \sum_{q=1}^{Q} a_{l,q} u_q(\mathbf{x}) + v_l(\mathbf{x}). \tag{4}$$

Here, $\mathbf{p}_l(\mathbf{x})$ is the $l$-th element of $\mathbf{p}(\mathbf{x})$, $Q$ is the number of involved GP's, $\{u_q(\mathbf{x})\}_{q=1}^{Q}$ and $\{v_l(\mathbf{x})\}_{l=1}^{m}$ are independent scalar-valued GP's, and the $a_{l,q}$'s are coefficient parameters. In this way, the correlation between different entries in the MGP $\mathbf{p}(\mathbf{x})$ is captured by $\{u_q(\mathbf{x})\}_{q=1}^{Q}$ through the $a_{l,q}$'s. Morever, $v_l(\mathbf{x})$ represents specific independent features of $\mathbf{p}_l(\mathbf{x})$ itself, for $l = 1, 2, \ldots, m$. Suppose the kernel functions of the $u_q(\mathbf{x})$'s and $v_l(\mathbf{x})$'s are $k_q(\mathbf{x}, \mathbf{x}')$'s and $\tilde{k}_l(\mathbf{x}, \mathbf{x}')$'s and all these kernel functions are less than or equal to one, as is regularly assumed [80, 81, 57]. Then the matrix-valued kernel function of $\mathbf{p}(\mathbf{x})$ is $\mathcal{K}(\mathbf{x}, \mathbf{x}') = \sum_{q=1}^{Q} \mathbf{A}_q k_q(\mathbf{x}, \mathbf{x}') +$ Diag $\left\{\tilde{k}_1(\mathbf{x}, \mathbf{x}'), \ldots, \tilde{k}_m(\mathbf{x}, \mathbf{x}')\right\}$. Here, $\mathbf{A}_q$ denotes the semi-definite matrix in which the $(l, l')$-th entry is $a_{l,q} a_{l',q}$. In some applications, the parameters involved in the kernel functions $k_q(\mathbf{x}, \mathbf{x}')$ and $\tilde{k}_l(\mathbf{x}, \mathbf{x}')$, and the coefficients $a_{l,q}$'s are not known in advance. We denote these unknown parameters and coefficients in the MGP as $\Phi$. In addition to the model (4), other types of MGP's can be employed in our methodology as well [14], while the selection of model (4) enables the theoretical analysis of our approach.

In terms of the mapping $g(\boldsymbol{\theta})$, since we have no prior knowledge, we select the neural network as the surrogate model, due to 1) its strong approximation power for intricate dependence on complex variables; 2) its flexibility of adaptation to different application scenarios (e.g. time-series or graph-structured contextual variables) and 3) the availability of fruitful methods and tools for the training procedure. Given the data $\mathcal{D}_t = \{(\boldsymbol{\theta}_1, \mathbf{x}_1, y_1), (\boldsymbol{\theta}_2, \mathbf{x}_2, y_2), \ldots, (\boldsymbol{\theta}_t, \mathbf{x}_t, y_t)\}$, the learning of unknown parameters in the MGP and the weight parameters in the neural network (as well as the noise variance) is through maximum likelihood estimation (MLE). That is

$$\left(\hat{\mathbf{W}}_t, \hat{\Phi}_t, \sigma_{\epsilon;t}^2\right) = \arg \max_{(\mathbf{W}, \Phi, \sigma_\epsilon^2)} L_t\left(\mathbf{W}, \Phi, \sigma_\epsilon^2\right), \tag{5}$$

where the (normalized) likelihood function is $L_t = -\ln\left[\left|\tilde{K}_{\mathcal{D}_t} + \sigma_\epsilon^2 I_t\right|\right] - \mathbf{y}_t^\top \left[\tilde{K}_{\mathcal{D}_t} + \sigma_\epsilon^2 I_t\right]^{-1} \mathbf{y}_t$.

Here, $\mathbf{y}_t = (y_1, y_2, \ldots, y_t)$ is the vector of observations and $\tilde{K}_{\mathcal{D}_t}$ is the covariance matrix of the data $\mathcal{D}_t$. The parameters $\mathbf{W}$ and $\Phi$ are contained in this covariance matrix. That is, instead of pre-defining a kernel function of the GP, the kernel function of NN-AGP is specified through learning the neural network from the data, yielding better approximation accuracy. We include a discussion of the consistency of training NN-AGP in the supplements.

## 3.2 Cumulative regret

Recall that the cumulative regret is defined as $\mathcal{R}_T = \sum_{t=1}^{T} \{\sup_{\mathbf{x}' \in \mathcal{X}} f(\mathbf{x}', \boldsymbol{\theta}_t) - f(\mathbf{x}_t, \boldsymbol{\theta}_t)\}$. Here we provide an upper bound of $\mathcal{R}_T$ with NN-AGP-UCB.

**Theorem 1.** *Suppose $\delta \in (0, 1)$ and the following.*

1. *The decision variable $x \in \mathcal{X} \subseteq [0, r]^d$ and $\mathcal{X}$ is convex and compact. The contextual variable $\boldsymbol{\theta} \in \Theta \subseteq \mathbb{R}^{d'}$ and $\Theta$ is convex and compact; $g(\boldsymbol{\theta})$ is a known continuous mapping of $\boldsymbol{\theta} \in \Theta$; $\mathbf{p}(\mathbf{x})$ is sampled from a known MGP prior as in (4) and the variance of the noise $\sigma_\epsilon^2$ is known. That is, these parameters do not need learning and updating from data.*

2. *For the components of the MGP, there exist constants* $\{a_q\}_{q=1}^{Q}$, $\{b_q\}_{q=1}^{Q}$, $\{\tilde{a}_l\}_{l=1}^{m}$, $\left\{\tilde{b}_l\right\}_{l=1}^{m}$ *satisfying*

$$\mathbb{P}\left\{\sup_{x\in\mathcal{X}}\left|\frac{\partial u_q(x)}{\partial x_j}\right| > L_q\right\} \leqslant a_q e^{-(L_q/b_q)^2}; \mathbb{P}\left\{\sup_{x\in\mathcal{X}}\left|\frac{\partial v_l(x)}{\partial x_j}\right| > \tilde{L}_l\right\} \leqslant \tilde{a}_l e^{-(\tilde{L}_l/\tilde{b}_l)^2} \quad (6)$$

$\forall L_q, \tilde{L}_l > 0$ *and* $\forall j = 1, 2, \ldots, d$, $q = 1, 2, \ldots, Q$ *and* $l = 1, 2, \ldots, m$.

3. *We choose as a hyper-parameter in (3)*

$$\beta_t = 2\log\left(t^2 2\pi^2/(3\delta)\right) + 2d\log\left(\tilde{M}t^2 dbr \sqrt{\log(4da/\delta)}\right),$$

*where* $d$ *and* $r$ *are the dimension and the upper bound of the decision variable,* $a = \sum_{q=1}^{Q} a_q + \sum_{l=1}^{m} \tilde{a}_l$, $b = \sum_{q=1}^{Q} b_q + \sum_{l=1}^{m} \tilde{b}_l$, $\tilde{M} = \sup_{\boldsymbol{\theta}\in\Theta}\left\{\{|\sum_{l=1}^{m} \boldsymbol{g}_l(\boldsymbol{\theta})a_{l,q}|\}_{q=1}^{Q}, \{|\boldsymbol{g}_l(\boldsymbol{\theta})|\}_{l=1}^{m}\right\}$, *where* $\boldsymbol{g}_l$ *denotes the l-th entry of* $\boldsymbol{g}(\boldsymbol{\theta})$.

*Then the cumulative regret is bounded with high probability as*

$$\mathbb{P}\left\{\mathcal{R}_T \leqslant \sqrt{\frac{8CT\beta_T\gamma_T}{\log\left(1 + C\sigma_\epsilon^{-2}\right)}} + \frac{\pi^2}{6}, \forall T \geqslant 1\right\} \geqslant 1 - \delta.$$

*Here* $C = \left(\left(\sum_{q=1}^{Q}\sum_{l=1}^{m} a_{l,q}^2\right) + 1\right)\sup_{\boldsymbol{\theta}\in\Theta}\|\boldsymbol{g}(\boldsymbol{\theta})\|_2^2$. *In addition,* $\gamma_T$ *is the maximum information gain associated with the NN-AGP* $f(\mathbf{x}; \boldsymbol{\theta})$*, defined by (7).*

We postpone the discussion of the maximum information gain $\gamma_T$ to Section 3.3, with some specific kernels employed in the MGP component of the NN-AGP model. We note that GP's with commonly-selected kernel functions, including the Matérn kernel and the radial basis function kernel, satisfy the condition (6) and further discussions can be found in **Theorem 5** in [45]. A detailed proof of **Theorem 1** is contained in the supplements. Note that **Theorem 1** assumes that $\boldsymbol{g}(\boldsymbol{\theta})$ is exactly known so does not consider the error of approximating $\boldsymbol{g}(\boldsymbol{\theta})$ with the neural networks. In the supplements, we include a detailed discussion of the algorithm that considers the neural network approximation error, as well as the corresponding regret bounds.

At the end of this section, we compare our regret bound with existing work. Specifically, NN-AGP-UCB has the same bound of $\tilde{\mathcal{O}}\left(\sqrt{T\gamma_T}\right)$ as CGP-UCB, but is superior when the contextual variable dimension is high; see details in the supplements. In terms of the algorithms which entirely rely on NN, we note that NeuralUCB [99], Neural TS [98], and Neural LinUCB [95] all consider the scenarios when the decision variable $\mathbf{x}$ is selected from a finite set. In comparison, we consider that $\mathbf{x}$ is selected from a continuous set. When performed on a finite feasible set $\mathcal{X}$, our NN-AGP-UCB also has a regret bound of $\tilde{\mathcal{O}}\left(\sqrt{T\gamma_T}\right)$, where the maximum information gain $\gamma_T$ further depends on the kernel function of the GP component used in NN-AGP. When the kernel function has an exponential eigendecay (see Definition 1), NN-AGP-UCB has a regret bound of $\tilde{\mathcal{O}}\left(\sqrt{T}\right)$, matching the regret bound of NeuralUCB, Neural TS and Neural LinUCB as well.

## 3.3 Maximum information gain

In this section, we discuss the information gain $\gamma_T$ of the proposed NN-AGP model. The maximum information gain is defined as

$$\gamma_T = \sup_{\{(\boldsymbol{\theta}_t, \mathbf{x}_t)\}_{t=1}^{T} \subseteq \Theta \times \mathcal{X}} I\left(\mathbf{y}_T; f_T\right), \quad (7)$$

where $f_T$ is the reward function evaluated at $\{(\boldsymbol{\theta}_t, \mathbf{x}_t)\}_{t=1}^{T}$; $\mathbf{y}_T$ denotes the observations; $I\left(\mathbf{y}_T; f_T\right) = H\left(\mathbf{y}_T\right) - H\left(\mathbf{y}_T \mid f_T\right)$ is the mutual information between $\mathbf{y}_T$ and $f_T$; $H(\cdot) = \mathbb{E}[-\log p(\cdot)]$ is the entropy of a random element, where $p$ is the probability density function.

For ease of notation, here we consider the scenario when $Q = 1$ and there is no $\{v_l(\mathbf{x})\}$ in the MGP defined in (4), i.e., $\mathbf{p}_l(\mathbf{x}) = a_l u(\mathbf{x})$. We also impose more variability so that $\mathbf{A} = \mathbf{A}_1$ is a semi-definite positive matrix and not necessarily a rank-one matrix, analogous to [14]. We provide a more general discussion of the maximum information gain of the NN-AGP in the supplements. We first provide a proposition on the kernel function.

**Proposition 2.** *When $Q = 1$ and there is no $\{v_l(\mathbf{x})\}$ in the MGP, the kernel function of the NN-AGP is a product of two kernel functions. That is $\tilde{K}\left((\mathbf{x}, \boldsymbol{\theta}), (\mathbf{x}', \boldsymbol{\theta}')\right) = \tilde{k}(\boldsymbol{\theta}, \boldsymbol{\theta}')k(\mathbf{x}, \mathbf{x}')$, where $\tilde{k}(\boldsymbol{\theta}, \boldsymbol{\theta}') = \boldsymbol{g}(\boldsymbol{\theta})^\top \mathbf{A}\boldsymbol{g}(\boldsymbol{\theta})$ is a finite rank kernel with respect to $\boldsymbol{\theta}$. Furthermore, suppose that both $\Theta$ and $\mathcal{X}$ are compact, $\boldsymbol{g}(\boldsymbol{\theta})$ is a continuous mapping and $k(\mathbf{x}, \mathbf{x}')$ is a semi-definite kernel function. Then the kernel function $\tilde{K}\left((\mathbf{x}, \boldsymbol{\theta}), (\mathbf{x}', \boldsymbol{\theta}')\right)$ possesses a Mercer decomposition:*

$$\tilde{K}\left((\mathbf{x}, \boldsymbol{\theta}), (\mathbf{x}', \boldsymbol{\theta}')\right) = \sum_{j=1}^{\infty}\sum_{i=1}^{m} \mu_i \lambda_j \phi_i\left(\boldsymbol{\theta}\right)\psi_j\left(\mathbf{x}\right)\phi_i\left(\boldsymbol{\theta}'\right)\psi_j\left(\mathbf{x}'\right).$$

*Here, $\{(\mu_i, \phi_i)\}_{i=1}^{m}$ and $\{(\lambda_j, \psi_j)\}_{j=1}^{m}$ are the Mercer decompositions for $\tilde{k}\left(\boldsymbol{\theta}, \boldsymbol{\theta}'\right)$ and $k\left(\mathbf{x}, \mathbf{x}'\right)$. That is, $\tilde{k}\left(\boldsymbol{\theta}, \boldsymbol{\theta}'\right) = \sum_{i=1}^{m}\mu_i \phi_i(\boldsymbol{\theta})\phi_i\left(\boldsymbol{\theta}'\right), k\left(\mathbf{x}, \mathbf{x}'\right) = \sum_{j=1}^{\infty}\lambda_j\psi_j\left(\mathbf{x}\right)\psi_j\left(\mathbf{x}'\right),$ where the eigenvalues are $\mu_1 \geqslant \mu_2 \geqslant \ldots \geqslant \mu_m \geqslant 0$ and $\lambda_1 \geqslant \lambda_2 \geqslant \ldots \geqslant 0$.*

With the Mercer decomposition of the NN-AGP kernel function, we provide the bound of the maximum information gain. Specifically, we consider two scenarios for the employed kernel in the MGP, analogous to [22, 84].

**Definition 1.** *Consider the eigenvalues $\{\lambda_j\}_{j=1}^{\infty}$ of the kernel function $k\left(\mathbf{x}, \mathbf{x}'\right)$ in decreasing order.*

- *For some $C_p > 0, \alpha_p > 1, k$ is said to have a $(C_p, \alpha_p)$ polynomial eigendecay, if for all $j \in \mathbb{N}$, we have $\lambda_j \leqslant C_p j^{-\alpha_p}$. An example is the Matérn kernel.*
- *For some $C_{e,1}, C_{e,2}, \alpha_e > 0, k$ is said to have a $(C_{e,1}, C_{e,2}, \alpha_e)$ exponential eigendecay, if for all $j \in \mathbb{N}$, we have $\lambda_j \leqslant C_{e,1} \exp\left(-C_{e,2}j^{\alpha_e}\right)$. An example is the radial basis function kernel.*

**Theorem 2.** *Suppose that 1) $\tilde{K}\left((\mathbf{x}, \boldsymbol{\theta}), (\mathbf{x}', \boldsymbol{\theta}')\right)$ satisfies the conditions in **Proposition 2**; 2) $\forall \mathbf{x}, \mathbf{x}' \in \mathcal{X}, |k\left(\mathbf{x}, \mathbf{x}'\right)| \leqslant \bar{k}$ for some $\bar{k} > 0$ and 3) $\forall j \in \mathbb{N}, \forall \mathbf{x} \in \mathcal{X}, |\psi_j(\mathbf{x})| \leqslant \psi,$ for some $\psi > 0$. If $k\left(\mathbf{x}, \mathbf{x}'\right)$ has a $(C_p, \alpha_p)$ polynomial eigendecay, then*

$$\gamma_T \leqslant m\left(\left(\frac{\bar{\mu}\phi^2\psi^2 C_p T}{\sigma_\epsilon^2}\log^{-1}\left(1 + \frac{\bar{\bar{k}}\bar{k}T}{m\sigma_\epsilon^2}\right)\right)^{\frac{1}{\alpha_p}} + 1\right)\log\left(1 + \frac{\bar{\bar{k}}\bar{k}T}{m\sigma_\epsilon^2}\right).$$

*If $k\left(\mathbf{x}, \mathbf{x}'\right)$ has a $(C_{e,1}, C_{e,2}, \alpha_e)$ exponential eigendecay, then*

$$\gamma_T \leqslant m\left(\left(\frac{2}{C_{e,2}}\left(\log(T) + C_{\alpha_e}\right)\right)^{\frac{1}{\alpha_e}} + 1\right)\log\left(1 + \frac{\bar{\bar{k}}\bar{k}T}{m\sigma_\epsilon^2}\right),$$

*where*

$$C_{\alpha_e} = \begin{cases} \log\left(\dfrac{C_{e,1}m\bar{\mu}\phi^2\psi^2}{\sigma_\epsilon^2 C_{e,2}}\right) & \text{if } \alpha_e = 1 \\[3mm] \log\left(\dfrac{2C_{e,1}m\bar{\mu}\phi^2\psi^2}{\sigma_\epsilon^2\alpha_e C_{e,2}}\right) + \left(\dfrac{1}{\alpha_e} - 1\right)\left(\log\left(\dfrac{2}{C_{e,2}}\left(\dfrac{1}{\alpha_e} - 1\right)\right) - 1\right) & \text{otherwise.} \end{cases}$$

*Here, $\bar{\mu} = \frac{1}{m}\sum_{i=1}^{m}\mu_i$ denotes the mean of the eigenvalues of the kernel function $\tilde{k}\left(\boldsymbol{\theta}, \boldsymbol{\theta}'\right)$; $\bar{\bar{k}} = \sup_{\boldsymbol{\theta}, \boldsymbol{\theta}' \in \Theta}\left|\tilde{k}\left(\boldsymbol{\theta}, \boldsymbol{\theta}'\right)\right|$ and $\phi = \sup_{\boldsymbol{\theta} \in \Theta}|\phi(\boldsymbol{\theta})|$. Moreover, the maximum information gain of the NN-AGP is $\mathcal{O}\left(m\gamma_{\mathbf{x};T}\right)$, where $\gamma_{\mathbf{x};T}$ is the maximum information of $k\left(\mathbf{x}, \mathbf{x}'\right)$.*

A more detailed discussion of the Mercer decomposition of NN-AGP is contained in the supplements, as well as the proofs of **Proposition 2** and **Theorem 2**. At the end of this section, we compare our results on maximum information gain with [57]. For the composite kernel that is a product of two kernels, the upper bound is $(d + d')\gamma_{\mathbf{x};T} + (d + d')\log T$. Here, $\gamma_{\mathbf{x};T}$ is the maximum information gain of the kernel function with the decision variable, and $d'$ and $d$ are the dimensions of the contextual variable and the decision variable. That is, the information gain (as well as the cumulative regret) increases as the dimension of the contextual variable increases. In comparison, the maximum information in **Theorem 2** does not depend on $d'$. Thus, the NN-AGP model has lower cumulative regret than the classical strategy in [57] when the dimension of the contextual variable is relatively high.

# 4 Experiments

In this section, we conduct experiments to show the practicality of our neural network-accompanied Gaussian process upper bound confidence (NN-AGP-UCB) approach. We apply different neural networks to different problems, including the fully-connected neural network (FCN) [77] to a synthetic reward function, the long short-term memory (LSTM) [48] neural network to a queuing problem with time sequence contextual variables, and the graph convolutional neural network (GCN) [78] to a pricing problem with diffusion networks. We add a noise $\epsilon_t \overset{i.i.d.}{\sim} \mathcal{N}(0, 0.01)$ to the ground-truth value of the reward in the first set of experiments. For the latter two sets of experiments, the reward is generated by stochastic simulation (and therefore is "black-box" with contextual/decision variables) and we postpone the description of the full dynamic to the supplements. We also provide additional experiments in the supplements, including 1) sensitivity on the structure of reward functions; 2) comparison with contextual variables possessing different dimensions; 3) regression tasks on complex functions and 4) finite decision/contextual variables with real data.

The baseline approaches includes CGP-UCB [57], NeuralUCB [99] and NN-UCB [54]. The experiment results provided are mean performances based on repeating the experiments 15 times. Standard deviations (represented by a shadow associated with the mean-value line) are also included. In each iteration, the exogenous contextual variable $\boldsymbol{\theta}_t$ is randomly selected from $\Theta$ with equal probability. In terms of the initialization, we randomly select decision variables independently of observed contextual variables for each approach in the first 20 iterations to attain surrogates. The specific description of the employed surrogate model in each approach is postponed to the supplements.

## 4.1 Synthetic reward function

In this section, we consider two synthetic reward functions

$$R_1(\mathbf{x}, \boldsymbol{\theta}) = -\sqrt{|\sin(\|\mathbf{x}\|)\, \boldsymbol{\theta}^3 \exp(\cos(\|\mathbf{x}\| + \|\boldsymbol{\theta}\|))|};$$
$$R_2(\mathbf{x}, \boldsymbol{\theta}) = -\sqrt{|\sin(\|\mathbf{x}\|)\mathbf{x}^3 \exp(\|\mathbf{x}\| + \cos(\|\boldsymbol{\theta}\|))|}.$$

For NN-AGP-UCB, we consider $m = 2, 3, 5$ to study the effects of model selection on the algorithm performance. For CGP-UCB, we consider both additive kernels and multiplicative kernels. Because both NeuralUCB and NN-UCB are designed for contextual bandits with finite arms, we adapt them to the problem we consider in Section 2 and postpone the details to the supplements. The experimental results of the average regret $\mathcal{R}_T/T$ are illustrated in **Figure 1** and **Figure 2**, which provide the following insights. **1. Comparison with baseline approaches.** For both reward functions, NN-AGP-UCB outperforms both the CGP-UCB and NN-based approaches. The advantage comes from 1) strong approximation power regarding $\boldsymbol{\theta}$ due to NN and 2) explicit inference regarding $\mathbf{x}$ due to GP. **2. Effects of the dimension $m$.** Generally, when $m$ increases, the model flexibility increases, and the regret might be smaller, although this improvement might not be significant in some scenarios. Furthermore, a relatively large $m$ might result in overparameterization especially when there are not enough iterations. A suggested selection of $m$ is $\lceil d/3 + d'/10 \rceil + 3$ considering both the algorithm performance and training complexity of models. **3. Breaking the linear assumption.** Recall that we assume the reward function is of the form of $R(\mathbf{x}; \boldsymbol{\theta}) = \boldsymbol{g}(\boldsymbol{\theta})^\top \mathbf{p}(\mathbf{x})$. However, the reward functions here break this linear assumption and yet our NN-AGP-UCB is still applicable and outperforms the baseline approaches. Moreover, additional experiments show that NN-AGP 1) is not sensitive on the structure of the reward functions; 2) has a greater advantage with higher-dimensional contextual variables and 3) achieves better approximation accuracy for complex functions, compared with a joint GP with composite kernels; see the supplements for details. In addition, when the dimension of the input increases, the uncertainty of the regret will increase as well; see **Figures 1 & 2** for comparison. We also include the recorded computational time of these bandit algorithms in the supplements.

## 4.2 Queuing problem with time sequence contextual variables

In this section, we show through experiments that the NN-AGP model is applicable to contextual GP bandits when the objective function depends on the sequence of the contextual variables. That is, the reward function at time $t$ can be approximated by

$$f_t(\mathbf{x}; \boldsymbol{\theta}_1, \boldsymbol{\theta}_2, \ldots, \boldsymbol{\theta}_t) \approx \boldsymbol{g}_t(\boldsymbol{\theta}_1, \boldsymbol{\theta}_2, \ldots, \boldsymbol{\theta}_t)^\top \mathbf{p}(\mathbf{x}).$$

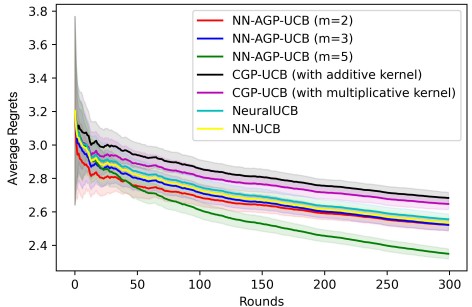 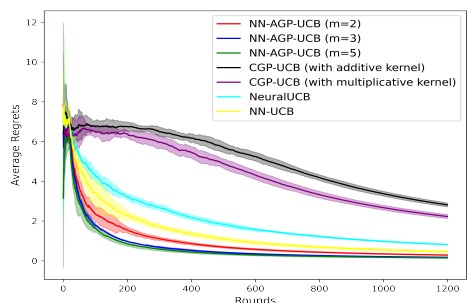

Figure 1: Average regret of using $R_1(\mathbf{x}; \boldsymbol{\theta})$ with $\mathcal{X} = [-1, 1]^2$ and $\Theta = [-1, 1]^3$.

Figure 2: Average regret of using $R_2(\mathbf{x}; \boldsymbol{\theta})$ with $\mathcal{X} = [-1, 1]^5$ and $\Theta = [-1, 1]^{15}$.

Here, $\boldsymbol{g}_t(\boldsymbol{\theta}_1, \boldsymbol{\theta}_2, \ldots, \boldsymbol{\theta}_t)$ is modeled by an LSTM neural network. We consider a discrete-time queuing problem. In each time epoch, a contextual variable is first revealed. For example, the contextual variable might includes traffic and weather conditions that affect the arrival process of the queuing system. The number of customers arriving at this epoch depends on the entire sequence of the revealed contextual variables up to now. The agent decides the service rate of the server and the service price for customers (decision variables). The reward function (might be negative) is the expected net income (the income brought by serving customers minus the service cost and penalty for losing customers). We let $\mathcal{X} = [0, 5]^2$ and sample $\boldsymbol{\theta}_t$ from multivariate normal distributions. We present the cumulative rewards in **Figure 3** and **Figure 4** of NN-AGP-UCB with LSTM. The baseline CGP-UCB adopts both additive and multiplicative kernels with the current contextual variable. We also apply kernel functions specifically designed for time series [10] to construct the composite kernel. Experimental results indicate that 1) employing a specific time series kernel enhances the performance of CGP-UCB and 2) NN-AGP-UCB with LSTM outperforms the classical CGP-UCB approaches with different composite kernels.

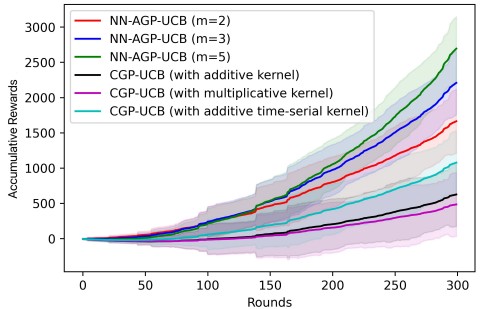 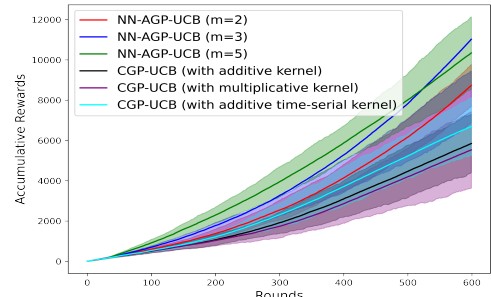

Figure 3: Cumulative rewards for a queuing problem with $\boldsymbol{\theta}_t \overset{i.i.d.}{\sim} \mathcal{N}(\mathbf{0}, I_3)$.

Figure 4: Cumulative rewards for a queuing problem with $\boldsymbol{\theta}_t \overset{i.i.d.}{\sim} \mathcal{N}(\mathbf{0}, I_{10})$.

## 4.3 Pricing with diffusion network

In this section, we show the NN-AGP model is applicable to contextual GP bandits with graph-structured contextual variables. That is, the contextual variable is summarized by a network structure

$$\boldsymbol{\theta}_t = (V_t, E_t),$$

where $V_t$ denotes the set of nodes and $E_t$ denotes the set of directed/undirected edges of a network. To approximate $\boldsymbol{g}(\boldsymbol{\theta})$ with a graph-structured contextual variable, we apply the GCN model. We consider a pricing problem with a diffusion network, where each node represents a user who decides

to adopt a service or not and the edge between two nodes indicates whether the choices of these two users influence each other. In each iteration, the network structure $\theta_t$ is first presented, and then the agent decides the price of the service. The reward function is the expected income for the service adoption from the users. The detailed description can be found in [66]. We let $\mathcal{X} = [0, 30]$ and $\theta_t$ represents an undirected graph with 5 and 10 nodes where each edge exists with probability 1/2. We present the cumulative rewards in **Figure 5** and **Figure 6** of NN-AGP-UCB with GCN. The baseline CGP-UCB adopts both additive and multiplicative kernels and in terms of the contextual variable, we consider 1) vectorizing the adjacency matrix that summarizes the network structure and 2) applying kernel functions that are specifically designed for graphs [49]. Experimental results indicate that NN-AGP-UCB with GCN outperforms the classical CGP-UCB approach adopting different kernels, and the advantages become greater for networks with more nodes.

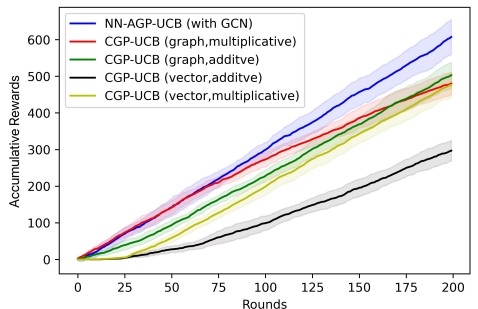
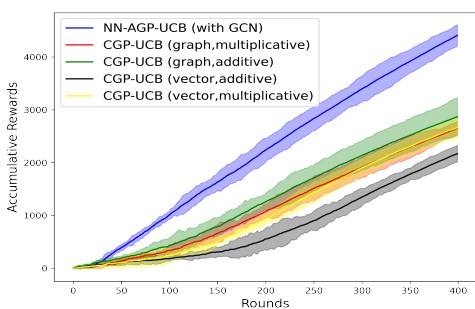

Figure 5: Cumulative rewards with a 5-node network diffusion problem.

Figure 6: Cumulative rewards with a 10-node network diffusion problem.

## 5   Conclusion & impact

We propose a neural network accompanied Gaussian process (NN-AGP) model to address contextual GP bandit problems. The advantages of our approach include 1) flexibility of employing different neural networks appropriate for applications with diverse structures of contextual information; 2) approximation accuracy for the reward function and better performance on cumulative rewards/regrets; 3) tractability of a GP representation regarding the decision variable, thus supporting explicit uncertainty quantification and theoretical analysis. Our approach has potential application to healthcare, where doctors need to develop therapy plans based on patient information to achieve optimal treatment effects. When complex and sparse genetic information is employed, it necessitates the use of neural networks. Another potential application is for Automated Guided Vehicles (AGVs) to enhance workplace safety and reduce carbon emissions, where environmental information is provided to the AGV, and the AGV takes actions accordingly.

In terms of limitations, since NN-AGP retains a GP structure, it suffers from computational complexity with large data sets. To alleviate the computational burden, we consider sparse NN-AGP for future work; see a discussion in the supplements. In addition, incorporating NN into bandit problems generally requires sufficient data to approximate the unknown reward function, thereby bringing the cold-start issue to NN-AGP. To address the challenge, we also include a discussion on employing transfer learning technologies in the supplements. Other potential future work includes 1) adapting NN-AGP to multi-objective/constrained optimization problems and 2) employing NN-AGP in a federated contextual bandit problem with multiple decentralized users.

## Acknowledgements

We thank the anonymous reviewers and chairs for constructive comments that helped improve this work. H. Zhang and J. He were partially supported by the Berkeley graduate fellowship. Z.J.M. Shen was partially supported by NSFC Grant 71991462. R. Righter is supported by the Ron Wolff Chaired Professorship.

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
