# Supplementary Materials

We provide the supplements of "Contextual Gaussian Process Bandits with Neural Networks" here. Specifically, we discuss alternative acquisition functions that can be incorporated with the neural network-accompanied Gaussian process (NN-AGP) model in Section 6. In Section 7, we discuss the bandit algorithm with NN-AGP, where the neural network approximation error is considered. In Section 8, we provide the detailed proof of theorems. We provide the experimental details and include additional numerical experiments in Section 9. Last we discuss the limitations of NN-AGP and propose the potential approaches to addressing the limitations for future work, including sparse NN-AGP for alleviating computational burdens and transfer learning with NN-AGP to address cold-start issue; see Section 10.

## 6  Acquisition functions

In the main text, we employ the upper confidence bound function as the acquisition function in the contextual Bayesian optimization approach. Here, we provide two alternative choices: Thompson sampling (TS) and knowledge gradient (KG). We describe the two procedures of the contextual GP bandit problems with NN-AGP, where the acquisition function is replaced by TS or KG. Both of them utilize the posterior distribution of the NN-AGP model

$$f\left(\mathbf{x};\boldsymbol{\theta}_t\right) \mid \mathcal{D}_{t-1} \sim \mathcal{N}\left(\mu_{t-1}\left(\mathbf{x};\boldsymbol{\theta}_t\right), \sigma_{t-1}^2\left(\mathbf{x};\boldsymbol{\theta}_t\right)\right) \tag{8}$$

with

$$\mu_{t-1}\left(\mathbf{x};\boldsymbol{\theta}_t\right) = \tilde{\mathcal{K}}_{(\mathbf{x};\boldsymbol{\theta}_t)}^{\top}\left[\tilde{\mathcal{K}}_{\mathcal{D}_{t-1}} + \sigma_\epsilon^2 I_{t-1}\right]^{-1}\mathbf{y}_{t-1},$$

$$\sigma_{t-1}^2\left(\mathbf{x};\boldsymbol{\theta}_t\right) = \boldsymbol{g}(\boldsymbol{\theta}_t)^{\top}\mathcal{K}\left(\mathbf{x},\mathbf{x}\right)\boldsymbol{g}(\boldsymbol{\theta}_t) - \tilde{\mathcal{K}}_{(\mathbf{x};\boldsymbol{\theta}_t)}^{\top}\left[\tilde{\mathcal{K}}_{\mathcal{D}_{t-1}} + \sigma_\epsilon^2 I_{t-1}\right]^{-1}\tilde{\mathcal{K}}_{(\mathbf{x};\boldsymbol{\theta}_t)}.$$

### 6.1  NN-AGP-TS

Thompson sampling (TS) is a heuristic for choosing actions in the multi-armed bandit problem. It chooses the action that maximizes the expected reward with respect to a random belief that is drawn for a posterior distribution. Besides the multi-armed bandit problems, TS has also achieved both theoretical and practical success in BO and Gaussian process regression. For more detailed discussions on TS, we refer to [87, 88].

Specifically, we propose a neural network-accompanied Gaussian process Thompson sampling (NN-AGP-TS) approach to address contextual GP bandits. The approach works as follows. In each iteration, NN-AGP-TS first fits an NN-AGP model with the historic data. Then, given the current contextual variable, a realization of the Gaussian process with respect to $\mathbf{x} \in \mathcal{X}$ is sampled from the posterior distribution conditional on the historic data[1]. That is,

$$\hat{f}\left(\mathbf{x};\boldsymbol{\theta}_t\right) \sim \mathcal{N}\left(\mu_{t-1}\left(\mathbf{x};\boldsymbol{\theta}_t\right), \sigma_{t-1}^2\left(\mathbf{x};\boldsymbol{\theta}_t\right)\right), \mathbf{x} \in \mathcal{X}.$$

The realization $\hat{f}\left(\mathbf{x};\boldsymbol{\theta}_t\right)$ is a deterministic function and adopts a closed-form expression with respect to $\mathbf{x}$. Thus, efficient optimization approaches (e.g. global heuristic search) can be applied to find the next point to sample $\mathbf{x}_t$ by solving the optimization problem

$$\mathbf{x}_t = \arg\max_{\mathbf{x}\in\mathcal{X}} \hat{f}\left(\mathbf{x};\boldsymbol{\theta}_t\right).$$

The complete procedure of NN-AGP-TS is summarized as in **Algorithm 2**.

Different from the upper confidence bound (UCB)-based algorithms, the TS method considers a Bayesian cumulative regret

$$\tilde{\mathcal{R}}_T = \sum_{t=1}^{T}\mathbb{E}\left[\sup_{\mathbf{x}'\in\mathcal{X}} f\left(\mathbf{x}',\boldsymbol{\theta}_t\right) - f\left(\mathbf{x}_t,\boldsymbol{\theta}_t\right)\right],$$

---

[1]An efficient implementation of sampling Gaussian processes given the mean and covariance functions can be found in https://www.r-bloggers.com/2019/07/sampling-paths-from-a-gaussian-process/.

---
**Algorithm 2** NN-AGP-TS
---
**Input:** A prior of $\left(\mathbf{W}, \Phi, \sigma_\epsilon^2\right)$;
**for** $t = 1, 2, \ldots, T$ **do**
    Observe the contextual variable $\boldsymbol{\theta}_t$;
    Sample a realization $\hat{f}\left(\mathbf{x}; \boldsymbol{\theta}_t\right) \sim \mathcal{N}\left(\mu_{t-1}\left(\mathbf{x}; \boldsymbol{\theta}_t\right), \sigma_{t-1}^2\left(\mathbf{x}; \boldsymbol{\theta}_t\right)\right), \mathbf{x} \in \mathcal{X}$;
    Select $\mathbf{x}_t = \arg\max_{\mathbf{x} \in \mathcal{X}} \hat{f}\left(\mathbf{x}; \boldsymbol{\theta}_t\right)$;
    Sample $y_t$ at $\left(\boldsymbol{\theta}_t, \mathbf{x}_t\right)$;
    Update $\left(\hat{\mathbf{W}}_t, \hat{\Phi}_t, \hat{\sigma_{\epsilon;t}^2}\right)$;
**end for**
---

where the expectation is taken over the prior distribution of $f(\mathbf{x}; \boldsymbol{\theta})$. We provide the upper bound of the cumulative regret for the NN-AGP-TS as a sanity check. For simplicity, we consider the scenario when $|\mathcal{X}|$ is finite. For a more general setting when $\mathcal{X}$ is continuous, the strategy of discretization that is adopted in the proof of **Theorem 1** can be applied as well.

**Theorem 3.** *Suppose that $g(\boldsymbol{\theta})$ is a known continuous mapping of $\boldsymbol{\theta} \in \Theta$; $\mathbf{p}(\mathbf{x})$ is sampled from a known MGP prior and the variance of the noise $\sigma_\epsilon^2$ is known. In addition, $|\mathcal{X}|$ is finite and $\Theta$ is a convex and compact set. The Bayesian cumulative regret of NN-AGP-TS is bounded by*

$$\tilde{\mathcal{R}}_T \leqslant C + 2\sqrt{\frac{CT\gamma_T}{\log\left(1 + C\sigma_\epsilon^{-2}\right)} \log\left(\frac{(T^2 + 1)|\mathcal{X}|}{\sqrt{2\pi}}\right)},$$

*where $C = \left(\left(\sum_{q=1}^{Q} \sum_{l=1}^{m} a_{l,q}^2\right) + 1\right) \sup_{\boldsymbol{\theta} \in \Theta} \|g(\boldsymbol{\theta})\|_2^2$ is a constant.*

The proof is largely based on the methodology proposed in [87] (therefore we will not present here considering the length of the supplements) and the upper bound of the posterior variance $\sigma_t^2(\mathbf{x}; \boldsymbol{\theta}_t)$. We postpone the discussion on the upper bound of posterior variance to Section 8, which is also used in the proof of **Theorem 1**.

## 6.2 NN-AGP-KG

In this section, we present the procedure of the neural network-accompanied knowledge gradient (NN-AGP-KG) approach, where the acquisition function employed in each iteration is contextual knowledge gradient (C-KG). The C-KG function at time $t$ is defined as

$$\text{C-KG}_t\left(\mathbf{x}; \boldsymbol{\theta}_t\right) = \mathbb{E}_{y_t}\left[\mu_t^*\left(\boldsymbol{\theta}_t\right) - \mu_{t-1}^*\left(\boldsymbol{\theta}_t\right) \mid \mathbf{x}_t = \mathbf{x}\right], \tag{9}$$

where $\mu_{t-1}^*\left(\boldsymbol{\theta}_t\right) = \max_{\mathbf{x} \in \mathcal{X}} \mu_{t-1}\left(\mathbf{x}; \boldsymbol{\theta}_t\right)$ and $\mu_t^*\left(\boldsymbol{\theta}_t\right) = \max_{\mathbf{x} \in \mathcal{X}} \mu_t\left(\mathbf{x}; \boldsymbol{\theta}_t\right)$. The difference between $\mu_{t-1}^*\left(\boldsymbol{\theta}_t\right)$ and $\mu_t^*\left(\boldsymbol{\theta}_t\right)$ is that, given the data-set $\mathcal{D}_{t-1}$, $\mu_{t-1}^*\left(\boldsymbol{\theta}_t\right)$ is deterministic, while $\mu_t^*\left(\boldsymbol{\theta}_t\right)$ depends on $y_t$ and therefore is random. Thus, the C-KG function requires taking the expectation with the unrevealed observation $y_t$. As is the regular KG acquisition function, simulated samples of $y_t$ as in (8) are required to approximate both the value of C-KG$_t\left(\mathbf{x}; \boldsymbol{\theta}_t\right)$ and the gradient $\nabla_{\mathbf{x}}$ C-KG$_t\left(\mathbf{x}; \boldsymbol{\theta}_t\right)$.

We present the procedure of selecting the decision variable $\mathbf{x}_t$ in each iteration with the C-KG and the NN-AGP model in **Algorithm 3**. For more detailed discussions on the knowledge gradient, including the statistical properties, we refer to [50, 93].

At the end of this section, we note that some classical acquisition functions that are widely adopted in Bayesian optimization might not be directly employed when the objective function involves contextual variables. An example is the expected improvement (EI) acquisition function, which is formulated as

$$\text{EI}(\mathbf{x}) = \mathbb{E}\left[\max\left\{f(\mathbf{x}) - f^*, 0\right\}\right],$$

where $f^*$ denotes the maximum values among the observations up to now, and the expectation is taken with the posterior distribution of the GP model $f$ conditional on the observed data. In other words, $f^*$ serves as a so-called incumbent that guides how to select the next point. However, when the objective function involves contextual variables, it is very likely that there are no observed data under the current value of the contextual variable. Thus, we can not define an incumbent for the EI function with the current contextual variable. A similar difficulty appears in the probability of improvement (PI) acquisition function as well. The comparison between different acquisition functions with our NN-AGP model will be contained in future work.

**Algorithm 3** Selection of $\mathbf{x}_t$ based on C-KG

---

**Input:** Number of points for the multi-start global optimization approach $\mathfrak{R}$; Number of iterations in the gradient descent for each starting point $\mathfrak{T}$; A rule to decide the step size $\alpha_t$;
**for** $\mathfrak{r} = 1, 2, \ldots, \mathfrak{R}$ **do**
    Select $\mathbf{x}_0^{(\mathfrak{r})}$ uniformly at random from $\mathcal{X}$;
    **for** $\mathfrak{t} = 1, 2, \ldots, \mathfrak{T}$ **do**
        Attain the stochastic gradient estimate of $\nabla_{\mathbf{x}}$ C-KG$_t\left(\mathbf{x}_{\mathfrak{t}}^{(\mathfrak{r})}; \boldsymbol{\theta}_t\right)$ as in **Algorithm 5**, denoted
        as $\mathfrak{G}$;
        Decide the step size $\alpha_t$;
        $\mathbf{x}_{\mathfrak{t}}^{(\mathfrak{r})} = \mathbf{x}_{\mathfrak{t}-1}^{(\mathfrak{r})} + \alpha_t \mathfrak{G}$;
    **end for**
    Estimate C-KG$_t\left(\mathbf{x}_{\mathfrak{T}}^{(\mathfrak{r})}; \boldsymbol{\theta}_t\right)$ as in **Algorithm 4**;
**end for**
Return $\mathbf{x}_t = \arg\max_{\mathbf{x}_{\mathfrak{T}}^{(\mathfrak{r})}}$ C-KG$_t\left(\mathbf{x}_{\mathfrak{T}}^{(\mathfrak{r})}; \boldsymbol{\theta}_t\right)$.

---

**Algorithm 4** Estimation of C-KG

---

**Input:** The conditional mean $\mu_{t-1}\left(\mathbf{x}; \boldsymbol{\theta}_t\right)$ and variance $\sigma_{t-1}^2\left(\mathbf{x}; \boldsymbol{\theta}_t\right)$ as attained in (2);
Attain $\mu_{t-1}^*\left(\boldsymbol{\theta}_t\right) = \max_{\mathbf{x} \in \mathcal{X}} \mu_{t-1}\left(\mathbf{x}; \boldsymbol{\theta}_t\right)$;
**for** $\mathfrak{j} = 1, 2, \ldots, \mathfrak{J}$ **do**
    Sample $y_t \sim \mathcal{N}\left(\mu_{t-1}\left(\mathbf{x}; \boldsymbol{\theta}_t\right), \sigma_{t-1}^2\left(\mathbf{x}; \boldsymbol{\theta}_t\right)\right)$;
    Update $\mu_t\left(\mathbf{x}'; \boldsymbol{\theta}_t\right), \mathbf{x}' \in \mathcal{X}$ with $\mathcal{D}_{t-1} \cup \{(\boldsymbol{\theta}_t, \mathbf{x}, y_t)\}$;
    Attain $\mu_t^*\left(\boldsymbol{\theta}_t\right) = \max_{\mathbf{x}' \in \mathcal{X}} \mu_t\left(\mathbf{x}'; \boldsymbol{\theta}_t\right)$;
    $\Delta^{(\mathfrak{j})} = \mu_t^*\left(\boldsymbol{\theta}_t\right) - \mu_{t-1}^*\left(\boldsymbol{\theta}_t\right)$;
**end for**
Return C-KG$_t\left(\mathbf{x}; \boldsymbol{\theta}_t\right) = \frac{1}{\mathfrak{J}} \sum_{\mathfrak{j}=1}^{\mathfrak{J}} \Delta^{(\mathfrak{j})}$.

---

# 7 NN-AGP with neural network error

In the main text, we provide an NN-AGP-UCB algorithm in Section 3.2 and provide the upper bound of the regrets in **Theorem 1**, where the error of approximating the mapping $\boldsymbol{g}(\boldsymbol{\theta})$ using the neural networks is not taken into consideration. In this section, we provide a detailed discussion on incorporating the neural network approximation error into consideration. Specifically, we select the MGP model $\mathbf{p}(\mathbf{x})$ in (4) as well. That is,

$$
\begin{aligned}
f(\mathbf{x}; \boldsymbol{\theta}) &= \boldsymbol{g}(\boldsymbol{\theta})^\top \mathbf{p}(\mathbf{x}) \\
&= \sum_{q=1}^Q \left( \sum_{l=1}^m \boldsymbol{g}_l(\boldsymbol{\theta}) a_{l,q} u_q(\mathbf{x}) \right) + \sum_{l=1}^m \boldsymbol{g}_l(\boldsymbol{\theta}) v_l(\mathbf{x})
\end{aligned}
$$

where $u_q$'s and $v_l$'s are independent scalar Gaussian processes with known kernel functions, and $a_{l,q}$'s are also known in advance. We note that the prior knowledge of the Gaussian process model is a regular assumption in Gaussian process bandits; see [68, 96]. Meanwhile, in terms of $\boldsymbol{g}(\boldsymbol{\theta}) = (\boldsymbol{g}_1(\boldsymbol{\theta}), \boldsymbol{g}_2(\boldsymbol{\theta}), \ldots, \boldsymbol{g}_m(\boldsymbol{\theta}))^\top$, we use

$$
\hat{\boldsymbol{g}}_t(\boldsymbol{\theta}) = (\hat{\boldsymbol{g}}_{1;t}(\boldsymbol{\theta}), \hat{\boldsymbol{g}}_{2;t}(\boldsymbol{\theta}), \ldots, \hat{\boldsymbol{g}}_{m;t}(\boldsymbol{\theta}))^\top
$$

as the approximation, where $\hat{\boldsymbol{g}}_t$ denotes the learned neural network in round $t$ with the historical data set $\mathcal{D}_{t-1}$. Here, we select the rectified linear unit (ReLU) function as the activation function in the neural networks and assume that each entry of the deterministic mapping $\boldsymbol{g}(\boldsymbol{\theta})$ is an $\boldsymbol{\alpha}$-Hölder function. That is, given a Hölder index $\boldsymbol{\alpha} > 0$, for any multi-index $s \in \mathbb{N}^{d'}$ with $|s| = \sum_{i=1}^{d'} s_i \leq \lfloor \alpha \rfloor$, the derivative $\partial^s \boldsymbol{g}_l = \frac{\partial^{|s|} \boldsymbol{g}_l}{\partial \theta_{(1)}^{s_1} \ldots \partial \theta_{(d')}^{s_{d'}}}$ exists, where $\theta_{(i)}$ denotes the $i$-th entry of $\boldsymbol{\theta}$. Meanwhile, for any $s$ satisfying $|s| = \lfloor \alpha \rfloor$, we have

$$
\sup_{\boldsymbol{\theta} \neq \boldsymbol{\theta}'} \frac{|\partial^s \boldsymbol{g}_l(\boldsymbol{\theta}) - \partial^s \boldsymbol{g}_l(\boldsymbol{\theta}')|}{\|\boldsymbol{\theta} - \boldsymbol{\theta}'\|_2^{\alpha - \lfloor \alpha \rfloor}} < \infty
$$

**Algorithm 5** Estimation of $\nabla_{\mathbf{x}}$ C-KG

---

**Input:** The conditional mean $\mu_{t-1}(\mathbf{x}; \boldsymbol{\theta}_t)$ and variance $\sigma_{t-1}^2(\mathbf{x}; \boldsymbol{\theta}_t)$ as attained in (2);
Attain $\mu_{t-1}^*(\boldsymbol{\theta}_t) = \max_{\mathbf{x} \in \mathcal{X}} \mu_{t-1}(\mathbf{x}; \boldsymbol{\theta}_t)$;
**for** $i = 1, 2, \dots, \mathfrak{I}$ **do**
    Sample $y_t \sim \mathcal{N}\left(\mu_{t-1}(\mathbf{x}; \boldsymbol{\theta}_t), \sigma_{t-1}^2(\mathbf{x}; \boldsymbol{\theta}_t)\right)$;
    Update $\mu_t(\mathbf{x}'; \boldsymbol{\theta}_t), \mathbf{x}' \in \mathcal{X}$ with $\mathcal{D}_{t-1} \cup \{(\boldsymbol{\theta}_t, \mathbf{x}, y_t)\}$;
    Attain $\mathbf{x}^* = \arg\max_{\mathbf{x}' \in \mathcal{X}} \mu_t(\mathbf{x}'; \boldsymbol{\theta}_t)$;
    $\mathfrak{G}^{(i)} = \nabla_{\mathbf{x}} \mu_t(\mathbf{x}; \boldsymbol{\theta}_t)|_{\mathbf{x} = \mathbf{x}^*}$;
**end for**
Return $\nabla_{\mathbf{x}}$ C-KG$_t(\mathbf{x}; \boldsymbol{\theta}_t) = \frac{1}{\mathfrak{I}} \sum_{i=1}^{\mathfrak{I}} \mathfrak{G}^{(i)}$.

---

for any $\boldsymbol{\theta}, \boldsymbol{\theta}'$ in the interior of $\Theta$.

By the universal approximation theorem of neural networks [28], when the weight parameters are properly chosen, the difference between $\hat{g}_{l;t}(\boldsymbol{\theta})$ and $g_l(\boldsymbol{\theta})$ can be arbitrarily small (denoted by $e_t$) when providing enough layers of neurons. Therefore, in our bandit algorithm, we iteratively reduce the error bound $e_t$ and enlarge the used neural network structure by adding more layers of nodes, since more observations are observed. We note that, since the data (observations $y_t$) come in stream, it is challenging to pre-specify a fixed, precise and relatively small error bound when the data is not sufficient at the beginning. Meanwhile, the training of the neural network may also suffer from overparameterization with a large amounts of layers of nodes when there is no sufficient data. Thus, our strategy of iteratively reducing the error bound and enlarging the neural network is reasonable. In practical implementations, when the neural network is enlarged in some iteration, we can first fix the trained components of the neural network and train the added components with the data. We note that similar strategies have been widely used in transfer learning technology; see [106, 123].

Next, we present the corresponding bandit algorithms of NN-AGP with the neural network error. Specifically, the selection of the next point is based on the NN-AGP model with the approximated NN

$$\hat{f}_t(\mathbf{x}, \boldsymbol{\theta}) = \hat{g}_t(\boldsymbol{\theta})^\top \mathbf{p}(\mathbf{x}).$$

Moreover, we denote

$$\hat{\mu}_{t-1}(\mathbf{x}; \boldsymbol{\theta}_t) = \tilde{\mathcal{K}}_{(\mathbf{x}; \boldsymbol{\theta}_t);t}^\top \left[\tilde{\mathcal{K}}_{(\mathcal{D}_{t-1});t} + \sigma_\epsilon^2 I_{t-1}\right]^{-1} \tilde{\mathbf{y}}_{t-1},$$

$$\tilde{\sigma}_{t-1}^2(\mathbf{x}; \boldsymbol{\theta}_t) = \hat{g}_t(\boldsymbol{\theta}_t)^\top \mathcal{K}(\mathbf{x}, \mathbf{x}) \hat{g}_t(\boldsymbol{\theta}_t) - \tilde{\mathcal{K}}_{(\mathbf{x}; \boldsymbol{\theta}_t);t}^\top \left[\tilde{\mathcal{K}}_{(\mathcal{D}_{t-1});t} + \sigma_\epsilon^2 I_{t-1}\right]^{-1} \tilde{\mathcal{K}}_{(\mathbf{x}; \boldsymbol{\theta}_t);t}.$$

Compared with the posterior mean and variance in (2), here we use $\hat{g}_t(\boldsymbol{\theta})$ in stead of $g(\boldsymbol{\theta})$. Besides, we also note that, $\hat{\mu}_{t-1}(\mathbf{x}; \boldsymbol{\theta}_t)$ depends on the observations $\mathbf{y}_t$ while $\tilde{\sigma}_{t-1}^2(\mathbf{x}; \boldsymbol{\theta}_t)$ does not. In this way, we provide the acquisition function as

$$\mathbf{x}_t = \arg\max_{\mathbf{x} \in \mathcal{X}} \left\{ \hat{\mu}_{t-1}(\mathbf{x}; \boldsymbol{\theta}_t) + \left(\tilde{\beta}_t^{1/2} + \frac{\tilde{e}_t \sqrt{t}}{\sigma_\epsilon}\right) \tilde{\sigma}_{t-1}(\mathbf{x}; \boldsymbol{\theta}_t) \right\}. \tag{10}$$

Here, $\tilde{\beta}_t$ is an increasing sequence to address the exploitation-exploration trade-off, similar as $\beta_t$ in NN-AGP-UCB, and $\tilde{e}_t$ is a decreasing sequence that depends on the neural network error bound $e_t$. Both $\tilde{\beta}_t$ and $\tilde{e}_t$ will be illustrated. We note that, by taking the neural network error into consideration, the exploration term will be enlarged. In other words, the bandit algorithm is performed more conservatively. In this way, we name the bandit algorithm that uses the acquisition function (10) as *NN-AGP-UCB+*. In Section 9, we present the numerical experiments of NN-AGP-UCB+. We note that NN-AGP-UCB+ does not always outperform NN-AGP-UCB in practical since the enlarged exploration terms is overly conservative. At the end of this section, we provide the theoretical support of NN-AGP-UCB+.

**Theorem 4.** *Suppose $\delta \in (0, 1)$ and the following.*

1. *The decision variable $x \in \mathcal{X} \subseteq [0, r]^d$ and $\mathcal{X}$ is convex and compact. The contextual variable $\boldsymbol{\theta} \in \Theta \subseteq [0, 1]^{d'}$ and $\Theta$ is convex and compact; each entry of $g(\boldsymbol{\theta})$ is an $\alpha$-Hölder function ($\alpha > 1$); $\mathbf{p}(\mathbf{x})$ is sampled from a known MGP prior as in (4) and the variance of the noise $\sigma_\epsilon^2$ is known.*

2. *In terms of the neural network $\boldsymbol{g}_t(\boldsymbol{\theta})$, we use the ReLU activation function. Meanwhile, in the $t$-th iteration, weight parameters are properly chosen such that each entry satisfies*

$$\sup_{\boldsymbol{\theta} \in \Theta} |\hat{\boldsymbol{g}}_{l;t}(\boldsymbol{\theta}) - \boldsymbol{g}_l(\boldsymbol{\theta})| \leqslant e_t,$$

*where neural network error sequence is selected as $e_t = \mathcal{O}\left(\frac{1}{t^{1+\Delta}}\right), \Delta > 0$.*

3. *For the components of the MGP, there exist constants $\{a_q\}_{q=1}^Q, \{b_q\}_{q=1}^Q, \{\tilde{a}_l\}_{l=1}^m, \left\{\tilde{b}_l\right\}_{l=1}^m$ satisfying*

$$\mathbb{P}\left\{\sup_{x \in \mathcal{X}} \left|\frac{\partial u_q(x)}{\partial x_j}\right| > L_q\right\} \leqslant a_q e^{-(L_q/b_q)^2}; \mathbb{P}\left\{\sup_{x \in \mathcal{X}} \left|\frac{\partial v_l(x)}{\partial x_j}\right| > \tilde{L}_l\right\} \leqslant \tilde{a}_l e^{-(\tilde{L}_l/\tilde{b}_l)^2}$$

*$\forall L_q, \tilde{L}_l > 0$ and $\forall j = 1, 2, \ldots, d$, $q = 1, 2, \ldots, Q$ and $l = 1, 2, \ldots, m$.*

4. *We choose as a hyper-parameter in (10)*

$$\tilde{\beta}_t = 2\log\left(t^2 2\pi^2/(3\delta)\right) + 2d\log\left(\tilde{\tilde{M}} t^2 dbr\sqrt{\log(4da/\delta)}\right),$$

*where $d$ and $r$ are the dimension and the upper bound of the decision variable, $a = \sum_{q=1}^Q a_q + \sum_{l=1}^m \tilde{a}_l$ and $b = \sum_{q=1}^Q b_q + \sum_{l=1}^m \tilde{b}_l$. Meanwhile,*

$$\tilde{\tilde{M}} = \sup_{\boldsymbol{\theta} \in \Theta; t}\left\{\left\{\left|\sum_{l=1}^m \hat{\boldsymbol{g}}_{l;t}(\boldsymbol{\theta})a_{l,q}\right|\right\}_{q=1}^Q, \{|\hat{\boldsymbol{g}}_{l;t}(\boldsymbol{\theta})|\}_{l=1}^m\right\}.$$

*Then the cumulative regret is bounded with high probability as*

$$\mathbb{P}\left\{\mathcal{R}_T = \mathcal{O}\left(\sqrt{T\gamma_T \tilde{\beta}_T}\right) + \mathcal{O}\left(T(\gamma_T)^{\frac{1}{4}}\left(\tilde{\beta}_T\right)^{\frac{1}{2}}\right), \forall T \geqslant 1\right\} \geqslant 1 - \delta.$$

*Here $C = \left(\left(\sum_{q=1}^Q \sum_{l=1}^m a_{l,q}^2\right) + 1\right)\sup_{\boldsymbol{\theta} \in \Theta} \|\boldsymbol{g}(\boldsymbol{\theta})\|_2^2$. In addition, $\gamma_T$ is the maximum information gain associated with the NN-AGP $f(\mathbf{x}; \boldsymbol{\theta})$, defined by (7). We also note that $\tilde{e}_t = \mathcal{C}_1 e_t$ and the neural network in the $t$-th iteration $\hat{\boldsymbol{g}}_t$ has (i) no more than $\mathcal{C}_2(1 - \log(e_t))$ layers and (ii) at most $\mathcal{C}_3 e_t^{-\frac{d'}{\alpha}}(1 - \log(e_t))$ neurons and weight parameters, where $\mathcal{C}_1, \mathcal{C}_2, \mathcal{C}_3$ are all constants.*

Compared with the results in **Theorem 1**, the regret bound of NN-AGP-UCB+ involves additional term $\mathcal{O}\left(T(\gamma_T)^{\frac{1}{4}}\left(\tilde{\beta}_T\right)^{\frac{1}{2}}\right)$, which results from the neural network approximation error. The proof of **Theorem 4** is contained in Section 8.

## 8 Proof

We present the detailed discussions on **Theorem 1**, **Theorem 2** and **Theorem 4** in this section, as well as the consistency of training the NN-AGP model from the data. We note that all these theoretical results are based on the assumption that $\mathbf{p}(\mathbf{x})$ is a linear combination of independent GP realizations ($u_q$'s and $v_l$'s). It is a common assumption that $u_q$'s and $v_l$'s are realizations from GP's, while an alternative review is to assume that each $u_q$ or $v_l$ is a deterministic function that lives in reproducing kernel Hilbert space (RKHS), which is consistent with relevant literature [95]. The difference between modeling the reward function as a GP sample or an element in an RKHS reflects the difference between Bayesianists and frequentists, as discussed in [95]. In our work, we adopt a Bayesian view since it helps us better understand the construction of the acquisition function. If the reward function $f(\mathbf{x}; \boldsymbol{\theta})$ is assumed as a linear combination of deterministic functions $u_q$'s and $v_l$'s that are from RKHS, martingale-based technologies introduced in [31] can be employed to derive the regret bound as well. This technology leads to a slightly tighter bound with less restrictive assumptions on $\mathcal{X}$, while it is beyond the scope of this work.

## 8.1 Proof of Theorem 1

In this section, we present the detailed proof of **Theorem 2**. The employed methodologies borrow the ideas from [95] and [68].

**Lemma 1** ([95]). *The information gain for the points selected can be expressed in terms of the predictive variances. Specifically,*

$$I\left(\mathbf{y}_T; f_T\right) = \frac{1}{2} \sum_{t=1}^{T} \log\left(1 + \sigma_\epsilon^{-2} \sigma_{t-1}^2\left(\mathbf{x}_t; \boldsymbol{\theta}_t\right)\right).$$

**Lemma 2.** *When $\beta_t$'s are selected nondecreasing,*

$$\sum_{t=1}^{T} 4\beta_t \sigma_{t-1}^2\left(\mathbf{x}_t; \boldsymbol{\theta}_t\right) \leqslant \frac{8C\beta_T\gamma_T}{\log\left(1 + C\sigma_\epsilon^{-2}\right)}, \forall T \geqslant 1.$$

*Here $C = \left(\left(\sum_{q=1}^{Q} \sum_{l=1}^{m} a_{l,q}^2\right) + 1\right) \sup_{\boldsymbol{\theta} \in \Theta} \|\boldsymbol{g}(\boldsymbol{\theta})\|_2^2$ is a constant.*

*Proof.* Note that

$$\tilde{K}\left((\mathbf{x}, \boldsymbol{\theta}), (\mathbf{x}', \boldsymbol{\theta}')\right) = \sum_{q=1}^{Q} \boldsymbol{g}(\boldsymbol{\theta})^\top \mathbf{A}_q \boldsymbol{g}(\boldsymbol{\theta}') k_q\left(\mathbf{x}, \mathbf{x}'\right) + \sum_{l=1}^{m} g_l(\boldsymbol{\theta}) g_l(\boldsymbol{\theta}') \tilde{k}_l(\mathbf{x}, \mathbf{x}').$$

With the regular conditions that all the kernel functions of $u_q$'s and $v_l$'s are less than one,

$$\sigma_{t-1}^2\left(\mathbf{x}_t; \boldsymbol{\theta}_t\right) \leqslant \tilde{K}\left((\mathbf{x}_t, \boldsymbol{\theta}_t), (\mathbf{x}_t, \boldsymbol{\theta}_t)\right)$$

$$\leqslant \sum_{q=1}^{Q} \boldsymbol{g}\left(\boldsymbol{\theta}_t\right)^\top \mathbf{A}_q \boldsymbol{g}\left(\boldsymbol{\theta}_t\right) + \sum_{l=1}^{m} \left(g_l\left(\boldsymbol{\theta}_t\right)\right)^2$$

$$\leqslant \left(\left(\sum_{q=1}^{Q} \sum_{l=1}^{m} a_{l,q}^2\right) + 1\right) \sup_{\boldsymbol{\theta} \in \Theta} \|g(\boldsymbol{\theta})\|_2^2$$

$$= C.$$

Indeed, since $\mathbf{A}_q = \begin{pmatrix} a_{1,q} \\ \vdots \\ a_{m,q} \end{pmatrix} \begin{pmatrix} a_{1,q} & \cdots & a_{m,q} \end{pmatrix}$, the only non-zero eigenvalue of $\mathbf{A}_q$ is $\begin{pmatrix} a_{1,q} & \cdots & a_{m,q} \end{pmatrix} \begin{pmatrix} a_{1,q} \\ \vdots \\ a_{m,q} \end{pmatrix} = \sum_{l=1}^{m} a_{l,q}^2$. This is the reason why the last inequality holds.

Next, since $\beta_t$'s are nondecreasing as $t$ increases,

$$4\beta_t \sigma_{t-1}^2\left(\mathbf{x}_t; \boldsymbol{\theta}_t\right) \leqslant 4\beta_T \sigma_\epsilon^2 \left(\sigma_\epsilon^{-2} \sigma_{t-1}^2\left(\mathbf{x}_t; \boldsymbol{\theta}_t\right)\right)$$

$$\leqslant \beta_T \frac{8C}{\log\left(1 + C\sigma_\epsilon^{-2}\right)} \frac{1}{2} \log\left(1 + \sigma_\epsilon^{-2} \sigma_{t-1}^2\left(\mathbf{x}_t; \boldsymbol{\theta}_t\right)\right).$$

The second inequality holds since $s/\log(1 + s)$ is an increasing function with $s$ and $\sigma_\epsilon^{-2} \sigma_{t-1}^2\left(\mathbf{x}_t; \boldsymbol{\theta}_t\right) \leqslant C\sigma_\epsilon^{-2}$. Thus, summing up the inequalities with $t$ to $T$, we have

$$\sum_{t=1}^{T} 4\beta_t \sigma_{t-1}^2\left(\mathbf{x}_t; \boldsymbol{\theta}_t\right) \leqslant \frac{8C\beta_T\gamma_T}{\log\left(1 + C\sigma_\epsilon^{-2}\right)}.$$

□

**Lemma 3** ([95]). *$\forall \delta \in (0, 1)$, choose $\beta_t = 2\log(\pi_t/\delta)$, where $\sum_{t \geqslant 1} \pi_t^{-1} = 1$.*

$$\mathbb{P}\left\{\forall t, |f(\mathbf{x}_t; \boldsymbol{\theta}_t) - \mu_{t-1}(\mathbf{x}_t; \boldsymbol{\theta}_t)| \leqslant \beta_t^{1/2} \sigma_{t-1}(\mathbf{x}_t; \boldsymbol{\theta}_t)\right\} \geqslant 1 - \delta.$$

**Lemma 4** ([95]). *Suppose that candidate decision variables* $\mathbf{x}_t$ *are finite in each round, that is,* $|\mathcal{X}_t|$ *is finite.* $\forall \delta \in (0,1)$, *choose* $\beta_t = 2\log(|\mathcal{X}_t| \pi_t/\delta)$, *where* $\sum_{t \geqslant 1} \pi_t^{-1} = 1$.

$$\mathbb{P}\left\{\forall t, \forall \mathbf{x} \in \mathcal{X}_t, |f(\mathbf{x}; \boldsymbol{\theta}_t) - \mu_{t-1}(\mathbf{x}; \boldsymbol{\theta}_t)| \leqslant \beta_t^{1/2} \sigma_{t-1}(\mathbf{x}; \boldsymbol{\theta}_t)\right\} \geqslant 1 - \delta.$$

The difference between **Lemma 3** and **Lemma 4** is that **Lemma 3** considers a specific $\mathbf{x}_t$ while **Lemma 4** considers a uniform bound on $\mathcal{X}_t$.

To connect the continuous decision variable space $\mathcal{X}$ to results with finite selections of $\mathbf{x}_t$, we then discretize $\mathcal{X}$ based on the smoothness condition imposed in **Theorem 1**. Recall that

$$f(\mathbf{x}; \boldsymbol{\theta}) = \sum_{q=1}^{Q}\left(\sum_{l=1}^{m} \boldsymbol{g}_l(\boldsymbol{\theta}) a_{l,q} u_q(\mathbf{x})\right) + \sum_{l=1}^{m} \boldsymbol{g}_l(\boldsymbol{\theta}) v_l(\mathbf{x}),$$

where $u_q$'s and $v_l$'s are independent scalar Gaussian processes. Meanwhile, for smoothness, we assume that for the components of the MGP, there exist constants $\{a_q\}_{q=1}^{Q}, \{b_q\}_{q=1}^{Q}, \{\tilde{a}_l\}_{l=1}^{m}, \left\{\tilde{b}_l\right\}_{l=1}^{m}$ satisfying

$$\mathbb{P}\left\{\sup_{x \in \mathcal{X}}\left|\frac{\partial u_q(x)}{\partial x_j}\right| > L_q\right\} \leqslant a_q e^{-(L_q/b_q)^2},$$

$$\mathbb{P}\left\{\sup_{x \in \mathcal{X}}\left|\frac{\partial v_l(x)}{\partial x_j}\right| > \tilde{L}_l\right\} \leqslant \tilde{a}_l e^{-(\tilde{L}_l/\tilde{b}_l)^2},$$

$\forall L_q, \tilde{L}_l > 0$ and $\forall j = 1, 2, \ldots, d$ for $q = 1, 2, \ldots, Q$ and $l = 1, 2, \ldots, m$.

For any fixed $k$, we could select $L_q$ and $\tilde{L}_l$ such that,

$$k = \frac{L_1^2}{b_1^2} = \ldots = \frac{L_Q^2}{b_Q^2} = \frac{\tilde{L}_1^2}{\tilde{b}_1^2} = \ldots = \frac{\tilde{L}_m^2}{\tilde{b}_m^2}.$$

Let $L = \tilde{M}(L_1 + \ldots + \tilde{L}_m)$, we have

$$\mathbb{P}\left\{\sup_{\mathbf{x} \in \mathcal{X}}\left|\frac{\partial f}{\partial \mathbf{x}_j}\right| > L\right\} \leqslant \mathbb{P}\left\{\sup_{\mathbf{x} \in \mathcal{X}}\left\{\sum_{q=1}^{Q}\left|\frac{\partial u_q(\mathbf{x})}{\partial \mathbf{x}_j}\right| + \sum_{l=1}^{m}\left|\frac{\partial v_l(\mathbf{x})}{\partial \mathbf{x}_j}\right|\right\} > \frac{L}{\tilde{M}}\right\}$$

$$\leqslant \sum_{q=1}^{Q}\mathbb{P}\left\{\sup_{x \in \mathcal{X}}\left|\frac{\partial u_q(x)}{\partial x_j}\right| > L_q\right\} + \sum_{l=1}^{m}\mathbb{P}\left\{\sup_{x \in \mathcal{X}}\left|\frac{\partial v_l(x)}{\partial x_j}\right| > \tilde{L}_l\right\}$$

$$\leqslant \sum_{q=1}^{Q} a_q e^{-(L_q/b_q)^2} + \sum_{l=1}^{m} \tilde{a}_l e^{-\left(\tilde{L}_l/\tilde{b}_l\right)^2}$$

$$\leqslant a e^{-k},$$

where $a = a_1 + \ldots + a_Q + \tilde{a}_1 + \ldots + \tilde{a}_m$. That is,

$$\mathbb{P}\left\{\forall \boldsymbol{\theta}, \forall \mathbf{x}, \mathbf{x}', |f(\mathbf{x}; \boldsymbol{\theta}) - f(\mathbf{x}'; \boldsymbol{\theta})| \leqslant L\|\mathbf{x} - \mathbf{x}'\|_1\right\} \geqslant 1 - d a e^{-k}, \tag{11}$$

where $L = \tilde{M} b \sqrt{k}, b = b_1 + \ldots + b_Q + \tilde{b}_1 + \ldots + \tilde{b}_m$.

In this way, we discretize $\mathcal{X}$ as $\mathcal{X}_t$ of size $(\tau_t)^d$ in each round so that for all $\mathbf{x} \in \mathcal{X}$, we can find

$$\|\mathbf{x} - [\mathbf{x}]_t\|_1 \leq rd/\tau_t,$$

where $[\mathbf{x}]_t$ denotes a point in $\mathcal{X}_t$ that is the closest to $\mathbf{x}$. A sufficient discretization has each coordinate in $\mathcal{X}$ with $\tau_t$ uniformly spaced points.

**Lemma 5.** $\forall \delta \in (0,1)$, *choose* $\beta_t = 2\log(2\pi_t/\delta) + 2d \log\left(dt^2 r \tilde{M} b \sqrt{\log\left(\frac{2ad}{\delta}\right)}\right)$, *where* $\sum_{t \geqslant 1} \pi_t^{-1} = 1$.

$$\mathbb{P}\left\{\forall t, |f(\mathbf{x}_t^*; \boldsymbol{\theta}_t) - \mu_{t-1}([\mathbf{x}_t^*]_t; \boldsymbol{\theta}_t)| \leqslant \beta_t^{1/2} \sigma_{t-1}([\mathbf{x}_t^*]_t; \boldsymbol{\theta}_t) + \frac{1}{t^2}\right\} \geqslant 1 - \delta.$$

*Proof.* For any $\delta \in (0, 1)$, we let $k = \log\left(\frac{2ad}{\delta}\right)$ in (11). We have

$$\mathbb{P}\left\{|f(\mathbf{x}; \boldsymbol{\theta}) - f\left([\mathbf{x}]_t; \boldsymbol{\theta}\right)| \leqslant \frac{1}{t^2}\right\} \geqslant 1 - \frac{\delta}{2},$$

with

$$\tau_t = dt^2 r \tilde{M} b \sqrt{\log\left(\frac{2ad}{\delta}\right)}.$$

Choose $\beta_t = 2\log\left(2\left(\tau_t\right)^d \pi_t/\delta\right) = 2\log\left(2\pi_t/\delta\right) + 2d\log\left(dt^2 r \tilde{M} b \sqrt{\log\left(\frac{2ad}{\delta}\right)}\right)$, we have

$$\mathbb{P}\left\{|f(\mathbf{x}_t^*; \boldsymbol{\theta}) - f\left([\mathbf{x}_t^*]_t; \boldsymbol{\theta}\right)| \leqslant \frac{1}{t^2}\right\} \geqslant 1 - \frac{\delta}{2}.$$

Based on **Lemma 4**, we have

$$\mathbb{P}\left\{\forall t, |f([\mathbf{x}_t^*]_t; \boldsymbol{\theta}_t) - \mu_{t-1}([\mathbf{x}_t^*]_t; \boldsymbol{\theta}_t)| \leqslant \beta_t^{1/2}\sigma_{t-1}([\mathbf{x}_t^*]_t; \boldsymbol{\theta}_t)\right\} \geqslant 1 - \frac{\delta}{2}.$$

for the reason that $\beta_t$ here is larger than the required $\beta_t$ in **Lemma 4**. Thus,

$$\mathbb{P}\left\{\forall t, |f(\mathbf{x}_t^*; \boldsymbol{\theta}_t) - \mu_{t-1}([\mathbf{x}_t^*]_t; \boldsymbol{\theta}_t)| \leqslant \beta_t^{1/2}\sigma_{t-1}([\mathbf{x}_t^*]_t; \boldsymbol{\theta}_t) + \frac{1}{t^2}\right\} \geqslant 1 - \delta.$$

$\square$

**Lemma 6.** *Choose* $\beta_t = 2\log\left(4\pi_t/\delta\right) + 2d\log\left(dt^2 r \tilde{M} b \sqrt{\log\left(\frac{4ad}{\delta}\right)}\right)$, *where* $\sum_{t \geqslant 1} \pi_t^{-1} = 1$.

$$\mathbb{P}\left\{\forall t, r_t \leqslant 2\beta_t^{1/2}\sigma_{t-1}(\mathbf{x}_t; \boldsymbol{\theta}_t) + \frac{1}{t^2}\right\} \geqslant 1 - \delta.$$

*Proof.* Select $\delta/2$ in **Lemma 3** and **Lemma 5**. With probability at least $1 - \delta$, we have

$$\forall t, |f(\mathbf{x}_t; \boldsymbol{\theta}_t) - \mu_{t-1}(\mathbf{x}_t; \boldsymbol{\theta}_t)| \leqslant \beta_t^{1/2}\sigma_{t-1}(\mathbf{x}_t; \boldsymbol{\theta}_t)$$

and

$$\forall t, |f(\mathbf{x}_t^*; \boldsymbol{\theta}_t) - \mu_{t-1}([\mathbf{x}_t^*]_t; \boldsymbol{\theta}_t)| \leqslant \beta_t^{1/2}\sigma_{t-1}([\mathbf{x}_t^*]_t; \boldsymbol{\theta}_t) + \frac{1}{t^2},$$

since $\beta_t$ here is larger than that required in **Lemma 3**. Thus,

$$\begin{aligned}
r_t &= f(\mathbf{x}_t^*; \boldsymbol{\theta}_t) - f(\mathbf{x}_t; \boldsymbol{\theta}_t) \\
&\leqslant \mu_{t-1}([\mathbf{x}_t^*]_t; \boldsymbol{\theta}_t) + \beta_t^{1/2}\sigma_{t-1}([\mathbf{x}_t^*]_t; \boldsymbol{\theta}_t) + \frac{1}{t^2} - f(\mathbf{x}_t; \boldsymbol{\theta}_t) \\
&\leqslant \beta_t^{1/2}\sigma_{t-1}(\mathbf{x}_t; \boldsymbol{\theta}_t) + \frac{1}{t^2} + \mu_{t-1}(\mathbf{x}_t; \boldsymbol{\theta}_t) - f(\mathbf{x}_t; \boldsymbol{\theta}_t) \\
&\leqslant 2\beta_t^{1/2}\sigma_{t-1}(\mathbf{x}_t; \boldsymbol{\theta}_t) + \frac{1}{t^2},
\end{aligned}$$

$\square$

Based on these results, we provide the proof of **Theorem 1** in the main text.

*Proof.* Recall that $\mathcal{R}_T = \sum_{t=1}^{T} r_t$ and

$$\sum_{t=1}^{T} 4\beta_t \sigma_{t-1}^2(\mathbf{x}_t; \boldsymbol{\theta}_t) \leqslant \frac{8C\beta_T \gamma_T}{\log\left(1 + C\sigma_\epsilon^{-2}\right)}.$$

Based on **Lemma 6**, with probability at least $1 - \delta$, we have

$$\mathcal{R}_T \leqslant \sqrt{T \sum_{t=1}^{T} 4\beta_t \sigma_{t-1}^2 \left(\mathbf{x}_t; \boldsymbol{\theta}_t\right) + \frac{\pi^2}{6}}$$

$$\leqslant \sqrt{\frac{8C\beta_T \gamma_T T}{\log \left(1 + C\sigma_\epsilon^{-2}\right)} + \frac{\pi^2}{6}}.$$

The first inequality holds because of the Cauchy–Schwarz inequality. Choose $\pi_t = \frac{2\pi^2 t^2}{3\delta}$, we attain the results in **Theorem 1**.

$\square$

At the end of this section, we compare our result with that of [68]. First, we impose the smoothness conditions on the separate components of the MGP (for further discussion on this condition refer to **Theorem 5** in [53]. [68] impose a similar condition directly on the joint GP with composite kernels. Thus, we offer a more explicit condition to verify. Second, we discretize the decision variable space, instead of the joint space of the decision variable and the contextual variable. Therefore, the upper bound on the cumulative regret $\mathcal{R}_T$ increases as the dimension of the decision variable space increases and is not related to the size of the contextual variable space. In comparison, the cumulative regret bound derived in [68] increases when the dimension of either the decision variable or the contextual variable increases. Therefore, NN-AGP-UCB adopts the advantage of a smaller upper bound on the cumulative regret when the dimension of the contextual variable is relatively high, which is also supported by experiment results in Section 9.

## 8.2 Proof of Theorem 2

The discussion of **Theorem 2** is based on the Mercer decomposition and is inspired by [100]. To begin with, we first present the Mercer Theorem.

**Theorem 5** (Mercer Theorem [47]). *Suppose $K(\mathbf{x}, \mathbf{x}')$ is a continuous symmetric non-negative definite kernel defined on $\mathcal{X} \times \mathcal{X}$. The kernel function $K(\mathbf{x}, \mathbf{x}')$ is called positive semi-definite (PSD) if any Gram-matrix generated by the kernel function is PSD. That is, for any sequence $\{\mathbf{x}_1 < \mathbf{x}_2 < \ldots < \mathbf{x}_n\}$, the matrix*

$$\begin{pmatrix} K\left(\mathbf{x}_1, \mathbf{x}_1\right) & \cdots & K\left(\mathbf{x}_1, \mathbf{x}_n\right) \\ \vdots & \ddots & \vdots \\ K\left(\mathbf{x}_n, \mathbf{x}_1\right) & \cdots & K\left(\mathbf{x}_n, \mathbf{x}_n\right) \end{pmatrix} \geqslant \mathbf{0}.$$

*Furthermore, there exists an orthonormal basis $\{e_i\}_i$ consisting of eigenfunctions such that the corresponding sequence of eigenvalues $\{\lambda_i\}$ is non-negative. The eigenfunctions corresponding to non-zero eigenvalues are continuous on $\mathcal{X}$ and $K$ has the representation*

$$K(\mathbf{x}, \mathbf{x}') = \sum_{i=1}^{\infty} \lambda_i e_i(\mathbf{x}) e_i(\mathbf{x}'),$$

*where the convergence is absolute and uniform. Specifically, the eigenfunctions and eigenvalues satisfy that*

$$\int_{\mathcal{X}} e_i(\mathbf{x}) e_j(\mathbf{x}) \, \mathrm{d}\mathbf{x} = \delta_{ij} = \begin{cases} 1, & i = j \\ 0, & i \neq j \end{cases}$$

$$\int_{\mathcal{X}} e_i(\mathbf{x}) K\left(\mathbf{x}, \mathbf{x}'\right) \, \mathrm{d}\mathbf{x} = \lambda_i e_i(\mathbf{x}').$$

Next, we present the Mercer decomposition for our NN-AGP model. Recall that the MGP employed in the NN-AGP model is determined by the linear transformation of multiple independent scalar Gaussian processes

$$\mathbf{p}_l(\mathbf{x}) = \sum_{q=1}^{Q} a_{l,q} u_q(\mathbf{x}) + v_l(\mathbf{x}).$$

Here, $\mathbf{p}_l(\mathbf{x})$ is the $l$-th element of $\mathbf{p}(\mathbf{x})$, where $\{u_q(\mathbf{x})\}_{q=1}^Q$ and $\{v_l(\mathbf{x})\}_{l=1}^m$ are independent scalar-output Gaussian processes. In addition, $a_{l,q}$'s are coefficient parameters. In this way, the correlation between different entries in the MGP $\mathbf{p}(\mathbf{x})$ is captured by $\{u_q(\mathbf{x})\}_{q=1}^Q$ through $a_{l,q}$. Meanwhile, $v_l(\mathbf{x})$ represents specific independent features of $\mathbf{p}_l(\mathbf{x})$ itself, for $l = 1, 2, \ldots, m$. Suppose the kernel function of $u_q(\mathbf{x})$'s and $v_l(\mathbf{x})$'s are $k_q(\mathbf{x}, \mathbf{x}')$'s and $\tilde{k}_l(\mathbf{x}, \mathbf{x}')$'s, the matrix-valued kernel function of $\mathbf{p}(\mathbf{x})$ is

$$\mathcal{K}(\mathbf{x}, \mathbf{x}') = \sum_{q=1}^Q \mathbf{A}_q k_q(\mathbf{x}, \mathbf{x}') + \mathrm{Diag}\left\{ \tilde{k}_1(\mathbf{x}, \mathbf{x}'), \ldots, \tilde{k}_m(\mathbf{x}, \mathbf{x}') \right\}.$$

Here, $\mathbf{A}_q$ denotes the semi-definite matrix, of which the $(l, l')$-th entry is $a_{l,q}a_{l',q}$. In this way, the kernel function for the NN-AGP is

$$\tilde{K}((\mathbf{x}, \boldsymbol{\theta}), (\mathbf{x}', \boldsymbol{\theta}')) = \sum_{q=1}^Q g(\boldsymbol{\theta})^\top \mathbf{A}_q g(\boldsymbol{\theta}') k_q(\mathbf{x}, \mathbf{x}') + \sum_{l=1}^m g_l(\boldsymbol{\theta}) g_l(\boldsymbol{\theta}') \tilde{k}_l(\mathbf{x}, \mathbf{x}').$$

Thus, the kernel function of the NN-AGP model is decomposed into a summation, where each term is a product of the two kernel functions.

**Proposition 3** ([68]). *Suppose a kernel function $K(\mathbf{x}, \mathbf{x}')$ can be represented by a summation of kernel functions. That is, $K(\mathbf{x}, \mathbf{x}') = \sum_{i=1}^n K_i(\mathbf{x}, \mathbf{x}')$. Then*

$$\gamma_T(K) \leqslant \sum_{i=1}^n \gamma_T(K_i).$$

*Here $\gamma_T\left(K_{(\,\cdot\,)}\right)$ denotes the maximum information gain associated with kernel function $K_{(\,\cdot\,)}$.*

That is, the maximum information gain of $\tilde{K}((\mathbf{x}, \boldsymbol{\theta}), (\mathbf{x}', \boldsymbol{\theta}'))$ is bounded by the summation of maximum information gains of each term. Thus, for ease of notion, we focus on the scenario when $Q = 1$ and there are no $v_l$'s. We also relax the restriction on $\mathbf{A}_1$ so that $\mathbf{A}_1 = \mathbf{A}$ is a positive semi-definite matrix and not necessarily a rank-one matrix.

**Proposition 4** (Mercer decomposition of $\tilde{k}$). *Given a positive semi-definite matrix $\mathbf{A}$ and a continuous function $g(\boldsymbol{\theta}), \boldsymbol{\theta} \in \Theta$,*

$$\tilde{k}(\boldsymbol{\theta}, \boldsymbol{\theta}') = g(\boldsymbol{\theta})^\top \mathbf{A} g(\boldsymbol{\theta}')$$

*defines a positive semi-definite (PSD) kernel function. Furthermore, when $\Theta$ is compact, the kernel function $\tilde{k}(\boldsymbol{\theta}, \boldsymbol{\theta}')$ has a finite-rank Mercer decomposition*

$$\tilde{k}(\boldsymbol{\theta}, \boldsymbol{\theta}') = \sum_{i=1}^m \mu_i \phi_i(\boldsymbol{\theta}) \phi_i(\boldsymbol{\theta}').$$

*Proof.* Since $\mathbf{A}$ is a PSD matrix, it has a Cholesky decomposition as $\mathbf{A} = LL^\top$, where $L$ is a lower triangular matrix. Denote $\tilde{g}(\boldsymbol{\theta}) = L^\top g(\boldsymbol{\theta})$, we have $\tilde{k}(\boldsymbol{\theta}, \boldsymbol{\theta}') = \tilde{g}(\boldsymbol{\theta})^\top \tilde{g}(\boldsymbol{\theta}')$. In this way, the covariance matrix of any sequence $\{\boldsymbol{\theta}_\tau\}_{\tau=1}^t$ is

$$\begin{pmatrix} \tilde{k}(\boldsymbol{\theta}_1, \boldsymbol{\theta}_1) & \cdots & \tilde{k}(\boldsymbol{\theta}_1, \boldsymbol{\theta}_t) \\ \vdots & \ddots & \vdots \\ \tilde{k}(\boldsymbol{\theta}_t, \boldsymbol{\theta}_1) & \cdots & \tilde{k}(\boldsymbol{\theta}_t, \boldsymbol{\theta}_t) \end{pmatrix} = \begin{pmatrix} \tilde{g}(\boldsymbol{\theta}_1)^\top \\ \cdots \\ \tilde{g}(\boldsymbol{\theta}_t)^\top \end{pmatrix} (\tilde{g}(\boldsymbol{\theta}_1) \quad \cdots \quad \tilde{g}(\boldsymbol{\theta}_t)) \geqslant \mathbf{0}.$$

Thus, $\tilde{k}(\boldsymbol{\theta}, \boldsymbol{\theta}')$ defines a PSD kernel function. With a slight abuse of notation, we let $\tilde{k}(\boldsymbol{\theta}, \boldsymbol{\theta}') = \sum_{l=1}^m g_l(\boldsymbol{\theta}) g_l(\boldsymbol{\theta}')$ and will re-define $\tilde{g}$ and $\tilde{g}_l$ in the following part. Since $\Theta$ is bounded, we define

$$\langle g_l, g_{l'} \rangle = \int_\Theta g_l(\boldsymbol{\theta}) g_{l'}(\boldsymbol{\theta}) \, \mathrm{d}\boldsymbol{\theta}$$

and

$$\|g_l\| = \sqrt{\langle g_l, g_l \rangle}.$$

Next, we let $\tilde{g}_1(\boldsymbol{\theta}) = \boldsymbol{g}(\boldsymbol{\theta})$. For $l = 2, \ldots, m$, we sequentially let

$$\tilde{g}_l(\boldsymbol{\theta}) = \boldsymbol{g}_l(\boldsymbol{\theta}) - \sum_{i=1}^{l-1} \frac{\langle \boldsymbol{g}_l, \tilde{\boldsymbol{g}}_i \rangle}{\|\tilde{\boldsymbol{g}}_i\|^2} \tilde{\boldsymbol{g}}_i(\boldsymbol{\theta}).$$

Without loss of generality, we assume that $\|\tilde{g}_l(\boldsymbol{\theta})\| = 1$. In fact, $\{\boldsymbol{g}_l\}_{l=1}^m$ composes a basis of a reproducing kernel Hilbert space, and the above procedure is exactly the Gram-Schmidt process of the basis. It can be easily verified that

$$\langle \boldsymbol{g}_l, \boldsymbol{g}_{l'} \rangle = \int_\Theta \boldsymbol{g}_l(\boldsymbol{\theta}) \boldsymbol{g}_{l'}(\boldsymbol{\theta}) \, \mathrm{d}\boldsymbol{\theta} = \delta_{ll'},$$

where $\delta_{ll'} = 1$ when $l = l'$ and $\delta_{ll'} = 0$ otherwise. Since the aforementioned procedure is entirely based on linear operations, we denote

$$\begin{pmatrix} \boldsymbol{g}_1 \\ \boldsymbol{g}_2 \\ \vdots \\ \boldsymbol{g}_m \end{pmatrix} = M \begin{pmatrix} \tilde{\boldsymbol{g}}_1 \\ \tilde{\boldsymbol{g}}_2 \\ \vdots \\ \tilde{\boldsymbol{g}}_m \end{pmatrix}.$$

In this way, we have that

$$\begin{aligned} \tilde{k}(\boldsymbol{\theta}, \boldsymbol{\theta}') &= \tilde{\boldsymbol{g}}(\boldsymbol{\theta})^\top M^\top M \tilde{\boldsymbol{g}}(\boldsymbol{\theta}) \\ &= \tilde{\boldsymbol{g}}(\boldsymbol{\theta})^\top Q^\top \Lambda Q \tilde{\boldsymbol{g}}(\boldsymbol{\theta}). \end{aligned}$$

Here $\tilde{\boldsymbol{g}}(\boldsymbol{\theta}) = (\tilde{g}_1(\boldsymbol{\theta}), \tilde{g}_2(\boldsymbol{\theta}), \ldots, \tilde{g}_m(\boldsymbol{\theta}))^\top$. $Q^\top \Lambda Q$ is the eigendecomposition of a PSD matrix $M^\top M$. That is, $\Lambda$ is a diagonal matrix and $Q$ is an orthogonal matrix. Let $\mu_i$ the $i$-th entry of $\Lambda$ and $\phi_i(\boldsymbol{\theta}$ the $i$-th entry of $Q \tilde{\boldsymbol{g}}(\boldsymbol{\theta})$. It can be easily verified that $\{\mu_i, \phi_i\}_{i=1}^m$ satisfies the conditions in the Mercer theorem. That is, we get the Mercer decomposition of $\tilde{k}(\boldsymbol{\theta}, \boldsymbol{\theta}')$. $\qquad\square$

Next, we show that the kernel function of NN-AGP adopts a Mercer decomposition as well.

**Proposition 5** (Mercer decomposition of NN-AGP). *Suppose that the kernel function with respect to decision variable $k(\mathbf{x}, \mathbf{x}')$ is PSD, and therefore adopts a Mercer decomposition*

$$k(\mathbf{x}, \mathbf{x}') = \sum_{j=1}^\infty \lambda_j \psi_j(\mathbf{x}) \psi_j(\mathbf{x}'),$$

*the kernel function of the NN-AGP then adopts a Mercer decomposition*

$$\tilde{K}((\mathbf{x}, \boldsymbol{\theta}), (\mathbf{x}', \boldsymbol{\theta}')) = \sum_{j=1}^\infty \sum_{i=1}^m \mu_i \lambda_j \phi_i(\boldsymbol{\theta}) \psi_j(\mathbf{x}) \phi_i(\boldsymbol{\theta}') \psi_j(\mathbf{x}'),$$

*if both $\Theta$ and $\mathcal{X}$ are compact, $g(\boldsymbol{\theta})$ is a continuous mapping.*

*Proof.* Recall that $\tilde{K}((\mathbf{x}, \boldsymbol{\theta}), (\mathbf{x}', \boldsymbol{\theta}')) = \tilde{k}(\boldsymbol{\theta}, \boldsymbol{\theta}') k(\mathbf{x}, \mathbf{x}')$. Based on the previous proposition and the Mercer decomposition of $k(\mathbf{x}, \mathbf{x}')$ (the convergence is uniform), we have

$$\tilde{K}((\mathbf{x}, \boldsymbol{\theta}), (\mathbf{x}', \boldsymbol{\theta}')) = \sum_{j=1}^\infty \sum_{i=1}^m \mu_i \lambda_j \phi_i(\boldsymbol{\theta}) \psi_j(\mathbf{x}) \phi_i(\boldsymbol{\theta}') \psi_j(\mathbf{x}').$$

Meanwhile, it can be easily verified that $\{(\mu_i \lambda_j, \phi_i(\boldsymbol{\theta}) \psi_j(\mathbf{x}))\}$ for $i = 1, 2, \ldots, m$ and $j = 1, 2, \ldots, \infty$, are eigenvalues and eigen-functions of $\tilde{K}((\mathbf{x}, \boldsymbol{\theta}), (\mathbf{x}', \boldsymbol{\theta}'))$, which also completes the proof of **Proposition 2** in the main text. $\qquad\square$

**Lemma 7.** *The NN-AGP model adopts a representation*

$$f(\mathbf{x}; \boldsymbol{\theta}) = \sum_{j=1}^\infty \sum_{i=1}^m \xi_{ji} \lambda_j^{\frac{1}{2}} \mu_i^{\frac{1}{2}} \phi_i(\boldsymbol{\theta}) \psi_j(\mathbf{x}),$$

*where $\xi_{ji}$'s are i.i.d. standard normal random variables.*

Based on the Mercer decomposition of $\tilde{K}\left((\mathbf{x}, \boldsymbol{\theta}), (\mathbf{x}', \boldsymbol{\theta}')\right)$, we derive the maximum information gain of the NN-AGP based on the technologies in [100]. Specifically, we select the first $D$ largest eigenvalues of the kernel function $k(\mathbf{x}, \mathbf{x}')$. Recall that in terms of the kernel function $\tilde{k}(\boldsymbol{\theta}, \boldsymbol{\theta}')$, there are at most $m$ non-zero eigenvalues. In this way, we project the NN-AGP onto the subspace and attain

$$\hat{f}(\mathbf{x}; \boldsymbol{\theta}) = \sum_{j=1}^{D} \sum_{i=1}^{m} \xi_{ji} \lambda_j^{\frac{1}{2}} \mu_i^{\frac{1}{2}} \phi_i(\boldsymbol{\theta}) \psi_j(\mathbf{x})$$

and the residual part

$$\hat{f}_r(\mathbf{x}; \boldsymbol{\theta}) = \sum_{j=D+1}^{\infty} \sum_{i=1}^{m} \xi_{ji} \lambda_j^{\frac{1}{2}} \mu_i^{\frac{1}{2}} \phi_i(\boldsymbol{\theta}) \psi_j(\mathbf{x}).$$

We then derive the bound of the maximum information gain by first considering the two separate terms.

**Lemma 8.** *Suppose that 1) $\tilde{K}\left((\mathrm{x}, \boldsymbol{\theta}), (\mathrm{x}', \boldsymbol{\theta}')\right)$ satisfies the conditions in **Proposition 2**; 2) $\forall \mathbf{x}, \mathbf{x}' \in \mathcal{X}, |k(\mathbf{x}, \mathbf{x}')| \leqslant \bar{k}$ for some $\bar{k} > 0$ and 3) $\forall j \in \mathbb{N}, \forall \mathbf{x} \in \mathcal{X}, |\psi_j(\mathbf{x})| \leqslant \psi$, for some $\psi > 0$.*

$$\gamma_T \leqslant \frac{1}{2} m D \log\left(1 + \frac{\bar{\bar{k}}\bar{k}T}{\sigma_\epsilon^2 m D}\right) + \frac{\delta_{mD}T}{2\sigma_\epsilon^2}. \tag{12}$$

*Here, $\delta_{mD} = \sum_{j=D+1}^{\infty} \sum_{i=1}^{m} \lambda_j \mu_i \psi^2 \phi^2$; $\sigma_\epsilon^2$ is the variance of the noise $\epsilon$; $\bar{\mu} = \frac{1}{m} \sum_{i=1}^{m} \mu_i$ denotes the mean of the eigenvalues of the kernel function $\tilde{k}(\boldsymbol{\theta}, \boldsymbol{\theta}')$; and $\bar{\bar{k}} = \sup_{\boldsymbol{\theta}, \boldsymbol{\theta}' \in \Theta} \left|\tilde{k}(\boldsymbol{\theta}, \boldsymbol{\theta}')\right|$ and $\phi = \sup_{\boldsymbol{\theta} \in \Theta} |\phi(\boldsymbol{\theta})|$.*

This is a direct result from **Theorem 3** in [100]. In terms of the right-hand side of (12), the first term bounds the maximum information gain associated with the projected GP while the second term bounds the remaining part of the GP. The NN-AGP can be projected onto such a sub-space for the reason that the kernel function with respect to $\boldsymbol{\theta}$ is a finite-rank kernel function. Otherwise, when both the kernel functions have infinite Mercer decompositions, the truncation of a product of two infinite series and the quantification of the residuals requires more cautious discussions.

Consider now a scalar GP with the kernel function $k(\mathbf{x}, \mathbf{x}')$, denoted as $p(\mathbf{x})$. The observations $y_t' = p(\mathbf{x}_t) + \epsilon_t$, where $\epsilon_t \sim \mathcal{N}(0, \sigma_\epsilon^2)$. That is, we do not consider the contextual variable here. In this way, the maximum information gain of $p(\mathbf{x})$ is bounded as

$$\gamma_{\mathbf{x};T} \leqslant \frac{1}{2} D \log\left(1 + \frac{\bar{k}T}{\sigma_\epsilon^2 D}\right) + \frac{\delta_D T}{2\sigma_\epsilon^2},$$

where $\delta_D = \sum_{j=D+1}^{\infty} \lambda_j \psi^2$. Compared with the result in (12), we have that

$$\gamma_T = \mathcal{O}\left(m\gamma_{\mathbf{x};T}\right).$$

Next, we specifically consider two types of the kernel function $k(\mathbf{x}, \mathbf{x}')$ as in **Definition 1**. Consider the eigenvalues $\{\lambda_j\}_{j=1}^{\infty}$ of the kernel function $k(\mathbf{x}, \mathbf{x}')$ in decreasing order.

1. For some $C_p > 0, \alpha_p > 1, k$ is said to have a $(C_p, \alpha_p)$ polynomial eigendecay, if for all $j \in \mathbb{N}$, we have $\lambda_j \leqslant C_p j^{-\alpha_p}$. Examples include the Matérn kernel.

2. For some $C_{e,1}, C_{e,2}, \alpha_e > 0, k$ is said to have a $(C_{e,1}, C_{e,2}, \alpha_e)$ exponential eigendecay, if for all $j \in \mathbb{N}$, we have $\lambda_j \leqslant C_{e,1} \exp\left(-C_{e,2} j^{\alpha_e}\right)$. Examples include the radial basis function kernel.

Based on the discussions above, we finally present the proof of **Theorem 2**.

*Proof.* Under the polynomial eigendecay $(C_p, \alpha_p)$, we have

$$\begin{aligned}
\delta_{mD} &\leqslant m\bar{\mu}\phi^2\psi^2 \sum_{j=D+1}^{\infty} C_p j^{-\beta_p} \\
&\leqslant m\bar{\mu}\phi^2\psi^2 \int_{z=D}^{\infty} C_p z^{-\beta_p} \,\mathrm{d}z \\
&= m\bar{\mu} C_p D^{1-\beta_p} \psi^2 \phi^2.
\end{aligned}$$

We select

$$D = \left\lceil \left( \frac{\bar{\mu}\psi^2\phi^2 C_p T}{\log\left(1 + \frac{\bar{\bar{k}}\bar{k}T}{m\sigma_\epsilon^2}\right)\sigma_\epsilon^2} \right)^{1/\alpha_p} \right\rceil,$$

so that

$$\frac{\delta_{mD} T}{\sigma_\epsilon^2} \leqslant mD \log\left(1 + \frac{\bar{\bar{k}}\bar{k}T}{\sigma_\epsilon^2 m}\right).$$

In this way, we have that

$$\gamma_T \leqslant m\left( \left( \frac{\bar{\mu}\phi^2\psi^2 C_p T}{\sigma_\epsilon^2} \log^{-1}\left(1 + \frac{\bar{\bar{k}}\bar{k}T}{m\sigma_\epsilon^2}\right) \right)^{\frac{1}{\alpha_p}} + 1 \right) \log\left(1 + \frac{\bar{\bar{k}}\bar{k}T}{m\sigma_\epsilon^2}\right).$$

Under the exponential eigendecay $(C_{e,1}, C_{e,2}, \alpha_e)$, we have

$$\delta_{mD} \leqslant m\bar{\mu}\phi^2\psi^2 \sum_{m=D+1}^{\infty} C_{e,1} \exp\left(-C_{e,2}m^{\alpha_e}\right)$$

$$\leqslant m\bar{\mu}\phi^2\psi^2 \int_{z=D}^{\infty} C_{e,1} \exp\left(-C_{e,2}z^{\alpha_e}\right) \mathrm{d}z.$$

When $\alpha_e = 1$,

$$\int_{z=D}^{\infty} \exp\left(-C_{e,2}z^{\alpha_e}\right) \mathrm{d}z = \int_{z=D}^{\infty} \exp\left(-C_{e,2}z\right) \mathrm{d}z$$

$$= \frac{1}{C_{e,2}} \exp\left(-C_{e,2}D\right).$$

In this way, we select

$$D = \left\lceil \frac{1}{C_{e,2}} \log\left( \frac{C_{e,1} m\bar{\mu}\phi^2\psi^2 T}{\sigma_\epsilon^2 C_{e,2}} \right) \right\rceil,$$

so that $\delta_{mD} T/\sigma_\epsilon^2 \leqslant 1$.

When $\alpha_e \neq 1$,

$$\int_{z=D}^{\infty} \exp\left(-C_{e,2}z^{\alpha_e}\right) \mathrm{d}z = \frac{1}{\alpha_e} \int_{z=D^{\alpha_e}}^{\infty} z^{\frac{1}{\alpha_e}-1} \exp\left(-C_{e,2}z\right) \mathrm{d}z$$

$$= \frac{1}{\alpha_e} \int_{z=D^{\alpha_e}}^{\infty} z^{\frac{1}{\alpha_e}-1} \exp\left(-C_{e,2}\frac{z}{2}\right) \exp\left(-C_{e,2}\frac{z}{2}\right) \mathrm{d}z$$

$$\leqslant \frac{1}{\alpha_e} \int_{z=D^{\alpha_e}}^{\infty} \left( \frac{2}{C_{e,2}} \left(\frac{1}{\alpha_e} - 1\right) \right)^{\frac{1}{\alpha_e}-1} \exp\left( -\left(\frac{1}{\alpha_e} - 1\right) \right) \exp\left(-C_{e,2}\frac{z}{2}\right) \mathrm{d}z$$

$$= \frac{2}{C_{e,2}\alpha_e} \left( \frac{2}{C_{e,2}} \left(\frac{1}{\alpha_e} - 1\right) \right)^{\frac{1}{\alpha_e}-1} \exp\left( -\left(\frac{1}{\alpha_e} - 1\right) \right) \exp\left(-C_{e,2}\frac{D^{\alpha_e}}{2}\right).$$

In this way, we select

$$D = \left\lceil \left( \frac{2}{C_{e,2}} \left( \log(T) + \log\left( \frac{2C_{e,1} m\bar{\mu}\phi^2\psi^2}{\sigma_\epsilon^2 \alpha_e C_{e,2}} \right) + \left(\frac{1}{\alpha_e} - 1\right) \left( \log\left( \frac{2}{C_{e,2}} \left(\frac{1}{\alpha_e} - 1\right) \right) - 1 \right) \right) \right)^{\frac{1}{\alpha_e}} \right\rceil,$$

so that $\delta_{mD} T/\sigma_\epsilon^2 \leqslant 1$. Thus, whether $\alpha_e = 1$ or not, the second term in the upper bound of $\gamma_T$ as in (12) is bounded by $1/2$, a constant. Therefore,

$$\gamma_T \leqslant mD \log\left(1 + \frac{\bar{\bar{k}}\bar{k}T}{m\sigma_\epsilon^2}\right),$$

when $T$ is sufficiently large. To summarize the results for the exponential eigendecay, we have

$$\gamma_T \leqslant m \left( \left( \left( \frac{2}{C_{e,2}} \left( \log(T) + C_{\alpha_e} \right) \right)^{\frac{1}{\alpha_e}} + 1 \right) \log \left( 1 + \frac{\bar{\bar{k}} \bar{k} T}{m \sigma_\epsilon}^2 \right) \right),$$

where

$$C_{\alpha_e} = \log \left( \frac{C_{e,1} m \bar{\mu} \phi^2 \psi^2}{\sigma_\epsilon^2 C_{e,2}} \right)$$

if $\alpha_e = 1$, and

$$C_{\alpha_e} = \log \left( \frac{2 C_{e,1} m \bar{\mu} \phi^2 \psi^2}{\sigma_\epsilon^2 \alpha_e C_{e,2}} \right) + \left( \frac{1}{\alpha_e} - 1 \right) \left( \log \left( \frac{2}{C_{e,2}} \left( \frac{1}{\alpha_e} - 1 \right) \right) - 1 \right)$$

otherwise.

$\square$

## 8.3 Proof of Theorem 4

In this section, we present the detailed proof of **Theorem 4**. For the following discussion, we assume that the conditions described in **Theorem 4** are satisfied.

First, we begin with a lemma that indicates the neural network approximation error $\|\hat{g}_t(\boldsymbol{\theta}) - g(\boldsymbol{\theta})\|$ can be arbitrarily small, providing enough layers of nodes. Here $\hat{g}_t(\boldsymbol{\theta})$ is the employed neural networks in the $t$-th round and $g(\boldsymbol{\theta})$ is the ground-truth function, which is assume to be an $\alpha$-Hölder function as in **Theorem 4**.

**Lemma 9.** *With properly chosen weight parameters, there exists a ReLU neural network, such that*

$$\sup_{\boldsymbol{\theta} \in \Theta} |\hat{g}_{l;t}(\boldsymbol{\theta}) - g_l(\boldsymbol{\theta})| \leqslant e_t, \tag{13}$$

*with a given $e_t \in (0,1)$. Here the neural network in the $t$-th iteration $\hat{g}_t$ has (i) no more than $\mathcal{C}_2 (1 - \log(e_t))$ layers and (ii) at most $\mathcal{C}_3 e_t^{-\frac{d'}{\alpha}} (1 - \log(e_t))$ neurons and weight parameters, where $\mathcal{C}_2, \mathcal{C}_3$ are both constants.*

**Lemma 9** follows directly from **Lemma 2** in [28]. For further discussions on neural network generalization error, we refer to [117].

Based on **Lemma 9**, we then provide that, with a high probability, the error between the ground-truth NN-AGP $f(\mathbf{x}; \boldsymbol{\theta})$ and the NN-AGP with the approximated neural network $\hat{f}_t(\mathbf{x}; \boldsymbol{\theta})$ can be bounded

**Lemma 10.** *With probability at least $1 - \delta$, we have that*

$$\forall \mathbf{x} \in \mathcal{X}, \boldsymbol{\theta} \in \Theta, \left| \hat{f}_t(\mathbf{x}; \boldsymbol{\theta}) - f(\mathbf{x}; \boldsymbol{\theta}) \right| \leqslant \mathcal{C}_1 e_t,$$

*where $\mathcal{C}_1$ is a constant.*

*Proof.* Note that,

$$\left| \hat{f}_t(\mathbf{x}; \boldsymbol{\theta}) - f(\mathbf{x}; \boldsymbol{\theta}) \right| = \left| \sum_{q=1}^{Q} \left( \sum_{l=1}^{m} (\hat{g}_{l;t}(\boldsymbol{\theta}) - g_l(\boldsymbol{\theta})) a_{l,q} u_q(\mathbf{x}) \right) + \sum_{l=1}^{m} (\hat{g}_{l;t}(\boldsymbol{\theta}) - g_l(\boldsymbol{\theta})) v_l(\mathbf{x}) \right|$$

$$\leqslant \left( \sum_{q=1}^{Q} \left( \sum_{l=1}^{m} |a_{l,q}| |u_q(\mathbf{x})| \right) + \sum_{l=1}^{m} |v_l(\mathbf{x})| \right) e_t.$$

Recall that, based on the condition (6), all $u_q$'s and $v_l$'s are Lipschitz with probability at least $1 - \delta$. This also implies that $u_q$'s and $v_l$'s are bounded as well, since $\mathcal{X}$ is compact. Thus, we denote

$$\tilde{C}_1 \doteq \sup_{\mathbf{x} \in \mathbf{X}} \left( \sum_{q=1}^{Q} \left( \sum_{l=1}^{m} |a_{l,q}| |u_q(\mathbf{x})| \right) + \sum_{l=1}^{m} |v_l(\mathbf{x})| \right)$$

and $\tilde{e}_t = \mathcal{C}_1 e_t$, and accomplish the proof. $\square$

Next, we focus on the misspecified NN-AGP $\hat{f}_t(\mathbf{x}; \boldsymbol{\theta}) = \hat{g}_t(\boldsymbol{\theta})^\top \mathbf{p}(\mathbf{x})$. Recall that, the ground-truth observations satisfy that

$$y_t = f(\mathbf{x}_t; \boldsymbol{\theta}_t) + \epsilon_t.$$

We then construct "virtual" observations as

$$\tilde{y}_t = \hat{f}_t(\mathbf{x}_t; \boldsymbol{\theta}_t) + \epsilon_t.$$

Consequently, we have that

$$|y_t - \tilde{y}_t| = \left| f(\mathbf{x}; \boldsymbol{\theta}) - \hat{f}_t(\mathbf{x}; \boldsymbol{\theta}) \right| \leqslant \tilde{e}_t$$

with high probability, based on **Lemma 10**. Besides, suppose we are now in the $t$-th round. Then the inference of the reward function uses the "virtual" observations. Specifically, the posterior mean and covariance that is associated with the NN-AGP model in the $t$-th round $\hat{f}_t(\mathbf{x}; \boldsymbol{\theta}_t)$ are denoted as

$$\tilde{\mu}_{t-1}(\mathbf{x}; \boldsymbol{\theta}_t) = \tilde{\mathcal{K}}_{(\mathbf{x};\boldsymbol{\theta}_t);t}^\top \left[ \tilde{\mathcal{K}}_{(\mathcal{D}_{t-1});t} + \sigma_\epsilon^2 I_{t-1} \right]^{-1} \tilde{\mathbf{y}}_{t-1},$$

$$\tilde{\sigma}_{t-1}^2(\mathbf{x}; \boldsymbol{\theta}_t) = \hat{g}_t(\boldsymbol{\theta}_t)^\top \mathcal{K}(\mathbf{x}, \mathbf{x}) \hat{g}_t(\boldsymbol{\theta}_t) - \tilde{\mathcal{K}}_{(\mathbf{x};\boldsymbol{\theta}_t);t}^\top \left[ \tilde{\mathcal{K}}_{(\mathcal{D}_{t-1});t} + \sigma_\epsilon^2 I_{t-1} \right]^{-1} \tilde{\mathcal{K}}_{(\mathbf{x};\boldsymbol{\theta}_t);t}.$$

Compared with the quantities used in the acquisition function (10) in NN-AGP-UCB+, the difference lies in between the posterior means $\hat{\mu}_{t-1}$ and $\tilde{\mu}_{t-1}$, where the real/"virtual" observations are incorporated. Actually, we have the following lemma to quantify the difference.

**Lemma 11.** *It is satisfied that, $\forall t$*

$$\left| \tilde{\mu}_t(\mathbf{x}; \boldsymbol{\theta}_t) - \hat{\mu}_t(\mathbf{x}; \boldsymbol{\theta}_t) \right| \leqslant \frac{\tilde{e}_t \sqrt{t}}{\sigma_\epsilon} \tilde{\sigma}_t(\mathbf{x}; \boldsymbol{\theta}_t),$$

*if $|y_t - \tilde{y}_t| \leqslant \tilde{e}_t$.*

The proof of **Lemma 11** is similar to **Lemma 2** in [14]. In addition, since $\tilde{\mu}_{t-1}(\mathbf{x}; \boldsymbol{\theta}_t)$ and $\tilde{\sigma}_{t-1}^2(\mathbf{x}; \boldsymbol{\theta}_t)$ denote the posterior mean and variance of $\hat{f}_t(\mathbf{x}; \boldsymbol{\theta}_t)$, we could still employ the union bound and discretization technologies in **Lemma 5** and **Lemma 6**, and reach the following lemma.

**Lemma 12.** *Choose $\tilde{\beta}_t = 2 \log(4\pi_t/\delta) + 2d \log \left( dt^2 r \tilde{\tilde{M}} b \sqrt{\log\left(\frac{4ad}{\delta}\right)} \right)$, we have*

$$\mathbb{P} \left\{ \begin{array}{l} \forall t, \left| \hat{f}_t(\mathbf{x}_t^*; \boldsymbol{\theta}_t) - \tilde{\mu}_{t-1}([\mathbf{x}_t^*]_t; \boldsymbol{\theta}_t) \right| \leqslant \tilde{\beta}_t^{1/2} \tilde{\sigma}_{t-1}([\mathbf{x}_t^*]_t; \boldsymbol{\theta}_t) + \frac{1}{t^2} \\ \left| \hat{f}_t(\mathbf{x}_t^*; \boldsymbol{\theta}_t) - \tilde{\mu}_{t-1}(\mathbf{x}_t^*; \boldsymbol{\theta}_t) \right| \leqslant \tilde{\beta}_t^{1/2} \tilde{\sigma}_{t-1}(\mathbf{x}_t^*; \boldsymbol{\theta}_t) \end{array} \right\} \geqslant 1 - \delta,$$

*Here, $\tilde{\tilde{M}} = \sup_{\boldsymbol{\theta} \in \Theta; t} \left\{ \{ |\sum_{l=1}^m \hat{g}_{l;t}(\boldsymbol{\theta}) a_{l,q}| \}_{q=1}^Q, \{ |\hat{g}_{l;t}(\boldsymbol{\theta})| \}_{l=1}^m \right\}$ and $[\mathbf{x}]_t$ denotes a point in $\mathcal{X}_t$ that is the closest to $\mathbf{x}$.*

The difference lies on that, in the previous proof, there is only one NN-AGP model $f(\mathbf{x}; \boldsymbol{\theta})$ in all the iterations. However, when the approximation of the neural network is taken into consideration, the kernel function (therefore, the NN-AGP) changes in every iteration. Although the NN-AGP model changes every time, the union bound can be employed as well. In addition, the GP's $u_q$'s and $v_l$'s remain the same to be discretized. In this way, we reach **Lemma 12**.

Next, we discuss the difference between $\tilde{\sigma}_t^2(\mathbf{x}_t; \boldsymbol{\theta}_t)$ and $\sigma_t^2(\mathbf{x}_t; \boldsymbol{\theta}_t)$. Specifically,

$$\tilde{\sigma}_t^2(\mathbf{x}_t; \boldsymbol{\theta}_t) - \sigma_t^2(\mathbf{x}_t; \boldsymbol{\theta}_t) = \tilde{K}_t((\mathbf{x}_t; \boldsymbol{\theta}_t), (\mathbf{x}_t; \boldsymbol{\theta}_t)) - \tilde{\mathcal{K}}_{(\mathbf{x}_t;\boldsymbol{\theta}_t);t}^\top \left[ \tilde{\mathcal{K}}_{(\mathcal{D}_t);t} + \sigma_\epsilon^2 I_t \right]^{-1} \tilde{\mathcal{K}}_{(\mathbf{x}_t;\boldsymbol{\theta}_t);t}$$

$$- \tilde{K}((\mathbf{x}_t; \boldsymbol{\theta}_t), (\mathbf{x}_t; \boldsymbol{\theta}_t)) + \tilde{\mathcal{K}}_{(\mathbf{x}_t;\boldsymbol{\theta}_t)}^\top \left[ \tilde{\mathcal{K}}_{\mathcal{D}_t} + \sigma_\epsilon^2 I_t \right]^{-1} \tilde{\mathcal{K}}_{(\mathbf{x}_t;\boldsymbol{\theta}_t)}$$

$$\leqslant \tilde{K}_t((\mathbf{x}_t; \boldsymbol{\theta}_t), (\mathbf{x}_t; \boldsymbol{\theta}_t)) - \tilde{K}((\mathbf{x}_t; \boldsymbol{\theta}_t), (\mathbf{x}_t; \boldsymbol{\theta}_t))$$

$$+ \tilde{\mathcal{K}}_{(\mathbf{x}_t;\boldsymbol{\theta}_t)}^\top \left[ \tilde{\mathcal{K}}_{\mathcal{D}_t} + \sigma_\epsilon^2 I_t \right]^{-1} \tilde{\mathcal{K}}_{(\mathbf{x}_t;\boldsymbol{\theta}_t)}.$$

In terms of the last term $\tilde{\mathcal{K}}_{(\mathbf{x}_t;\boldsymbol{\theta}_t)}^\top \left[\tilde{\mathcal{K}}_{\mathcal{D}_t} + \sigma_\epsilon^2 I_t\right]^{-1} \tilde{\mathcal{K}}_{(\mathbf{x}_t;\boldsymbol{\theta}_t)}$, we have that

$$\tilde{\mathcal{K}}_{(\mathbf{x}_t;\boldsymbol{\theta}_t)}^\top \left[\tilde{\mathcal{K}}_{\mathcal{D}_t} + \sigma_\epsilon^2 I_t\right]^{-1} \tilde{\mathcal{K}}_{(\mathbf{x}_t;\boldsymbol{\theta}_t)} \leqslant \frac{C\sqrt{t}}{\sigma_\epsilon} \sigma_t\left(\mathbf{x}_t;\boldsymbol{\theta}_t\right).$$

Here, $C = \left(\left(\sum_{q=1}^Q \sum_{l=1}^m a_{l,q}^2\right) + 1\right) \sup_{\boldsymbol{\theta}\in\Theta} \|g(\boldsymbol{\theta})\|_2^2$ denotes the upper bound of $\tilde{K}\left((\mathbf{x}_t;\boldsymbol{\theta}_t),(\mathbf{x}_t;\boldsymbol{\theta}_t)\right)$. The proof of this inequality can be referred to **Theorem 2** in [31]. On the other, in terms of $\tilde{\tilde{K}}_t\left((\mathbf{x}_t;\boldsymbol{\theta}_t),(\mathbf{x}_t;\boldsymbol{\theta}_t)\right) - \tilde{K}\left((\mathbf{x}_t;\boldsymbol{\theta}_t),(\mathbf{x}_t;\boldsymbol{\theta}_t)\right)$, the difference comes from the bias using neural networks $\hat{g}_t$ to approximate the mapping $g$. That is,

$$\tilde{\tilde{K}}_t\left((\mathbf{x}_t;\boldsymbol{\theta}_t),(\mathbf{x}_t;\boldsymbol{\theta}_t)\right) - \tilde{K}\left((\mathbf{x}_t;\boldsymbol{\theta}_t),(\mathbf{x}_t;\boldsymbol{\theta}_t)\right)$$

$$\leqslant \sum_{q=1}^Q \left(\hat{g}_t\left(\boldsymbol{\theta}_t\right) - g\left(\boldsymbol{\theta}_t\right)\right)^\top \mathbf{A}_q \left(\hat{g}_t\left(\boldsymbol{\theta}_t\right) - g\left(\boldsymbol{\theta}_t\right)\right) + \|\hat{g}_t\left(\boldsymbol{\theta}_t\right) - g\left(\boldsymbol{\theta}_t\right)\|^2$$

$$\leqslant \left(\left(\sum_{q=1}^Q \sum_{l=1}^m a_{l,q}^2\right) + 1\right) \|\hat{g}_t\left(\boldsymbol{\theta}_t\right) - g\left(\boldsymbol{\theta}_t\right)\|^2$$

$$\leqslant \left(\left(\sum_{q=1}^Q \sum_{l=1}^m a_{l,q}^2\right) + 1\right) m e_t^2.$$

In this way, by letting $\left(\left(\sum_{q=1}^Q \sum_{l=1}^m a_{l,q}^2\right) + 1\right) m e_t^2 \doteq e_t'$, we have

$$\tilde{\sigma}_t^2\left(\mathbf{x}_t;\theta_t\right) \leqslant \frac{C\sqrt{t}}{\sigma_\epsilon} \sigma_t\left(\mathbf{x}_t;\theta_t\right) + \sigma_t^2\left(\mathbf{x}_t;\theta_t\right) + e_t'.$$

Lastly, we present the proof of **Theorem 4**.

*Proof.* First, we notice that

$$\mathcal{R}_T \leqslant \tilde{\mathcal{R}}_T + 2\sum_{t=1}^T \tilde{e}_t,$$

since $\left|f\left(\mathbf{x};\boldsymbol{\theta}\right) - \hat{f}_t\left(\mathbf{x};\boldsymbol{\theta}\right)\right| \leqslant \tilde{e}_t$. Here $\tilde{\mathcal{R}}_T = \sum_{t=1}^T \tilde{r}_t$ denotes the "virtual" cumulative regrets based on the NN-AGP with neural network $\hat{g}_t\left(\boldsymbol{\theta}\right)$. That is, $\mathcal{R}_T$ and $\tilde{\mathcal{R}}_T$ are in the same order of $T$, since $\sum_{t=1}^\infty \tilde{e}_t < \infty$.

Next, we focus on the "virtual" regret $\tilde{r}_t$. Specifically, based on **Lemma 12**, with probability at least $1 - \delta$

$$\tilde{r}_t = \hat{f}_t\left(\mathbf{x}_t^*;\boldsymbol{\theta}_t\right) - \hat{f}_t\left(\mathbf{x}_t;\boldsymbol{\theta}_t\right)$$

$$\leqslant \tilde{\mu}_{t-1}\left([\mathbf{x}_t^*]_t;\boldsymbol{\theta}_t\right) + \left(\tilde{\beta}_t^{1/2} + \frac{\tilde{e}_t\sqrt{t}}{\sigma_\epsilon}\right) \tilde{\sigma}_{t-1}\left([\mathbf{x}_t^*]_t;\boldsymbol{\theta}_t\right) + \frac{1}{t^2} - \hat{f}_t\left(\mathbf{x}_t;\boldsymbol{\theta}_t\right)$$

$$\leqslant \tilde{\mu}_{t-1}\left(\mathbf{x}_t^*;\boldsymbol{\theta}_t\right) + \left(\tilde{\beta}_t^{1/2} + \frac{\tilde{e}_t\sqrt{t}}{\sigma_\epsilon}\right) \tilde{\sigma}_{t-1}\left(\mathbf{x}_t^*;\boldsymbol{\theta}_t\right) + \frac{1}{t^2} - \hat{f}_t\left(\mathbf{x}_t;\boldsymbol{\theta}_t\right)$$

$$\leqslant 2\left(\tilde{\beta}_t^{1/2} + \frac{\tilde{e}_t\sqrt{t}}{\sigma_\epsilon}\right) \tilde{\sigma}_{t-1}\left(\mathbf{x}_t^*;\boldsymbol{\theta}_t\right) + \frac{1}{t^2}.$$

Note that

$$\sum_{t=1}^T \tilde{\sigma}_{t-1}\left(\mathbf{x}_t;\boldsymbol{\theta}_t\right) \leqslant \sqrt{T\sum_{t=1}^T \tilde{\sigma}_{t-1}^2\left(\mathbf{x}_t;\boldsymbol{\theta}_t\right)}$$

$$\leqslant \sqrt{T\left(\frac{2C\gamma_T}{\log\left(1+C\sigma_\epsilon^{-2}\right)} + \frac{C^{\frac{3}{2}}T}{\sigma_\epsilon}\sqrt{\frac{2\gamma_T}{\log\left(1+C\sigma_\epsilon^{-2}\right)}} + \sum_{t=1}^T e_t'\right)}$$

$$= \mathcal{O}\left(\sqrt{T\gamma_T}\right) + \mathcal{O}\left(T\left(\gamma_T\right)^{\frac{1}{4}}\right),$$

where $\sum_{t=1}^{\infty} e'_t < \infty$. In addition, since $e_t = \mathcal{O}(\frac{1}{t^{1+\Delta}})$ and $\Delta > 0$, $\tilde{e}_t \sqrt{t}$ is bounded by some constant, say $\mathcal{B}$. Thus,

$$\tilde{\mathcal{R}}_T = \sum_{t=1}^{T} \tilde{r}_t$$

$$\leqslant 2 \left( \tilde{\beta}_T^{1/2} + \mathcal{B} \right) \sum_{t=1}^{T} \tilde{\sigma}_{t-1} \left( \mathbf{x}_t; \boldsymbol{\theta}_t \right)$$

$$= \mathcal{O} \left( \sqrt{T \gamma_T \tilde{\beta}_T} \right) + \mathcal{O} \left( T \left( \gamma_T \right)^{\frac{1}{4}} \left( \tilde{\beta}_T \right)^{\frac{1}{2}} \right).$$

$\square$

## 8.4 Proof of consistency

In this section, we discuss the consistency of training the NN-AGP model from the data. The consistency requires specific conditions on the sampling strategies of $\mathbf{x}_t$ and $\boldsymbol{\theta}_t$, which might be difficult to verify during the contextual GP bandits. However, we still present it as a sanity check here for two reasons. First, we note that the NN-AGP model can also be employed in supervised learning tasks when the function to be approximated involves both user-selected inputs $\mathbf{x}$ and observed contextual variables $\boldsymbol{\theta}$. Examples of these tasks can be found in [60]. In these task, the NN-AGP model still adopts the advantage of explicit kernel expression with respect to user-selected inputs $\mathbf{x}$ and approximation accuracy with respect to observed contextual variables $\boldsymbol{\theta}$. Meanwhile, the required conditions for consistency can be satisfied in these tasks. Second, existing theoretical results on GP bandits (as well as **Theorem 1** in our work) largely assume that the surrogate model is well-specified and does not require updating from the observations. That is, the discussion on the consistency of training NN-AGP from the data does not conflict with existing theoretical results on GP bandits.

To begin with, we first set up the notation used in the proof. Recall that the training objective of NN-AGP is to maximize the likelihood function, which is in the form of

$$L_t \left( \mathbf{W}, \Phi, \sigma_\epsilon^2 \right) = -\ln \left[ (2\pi)^{t/2} \right] - \frac{1}{2} \ln \left[ \left| \tilde{K}_{\mathcal{D}_t} + \sigma_\epsilon^2 I_t \right| \right] - \frac{1}{2} \mathbf{y}_t^\top \left[ \tilde{K}_{\mathcal{D}_t} + \sigma_\epsilon^2 I_t \right]^{-1} \mathbf{y}_t.$$

It is equivalent to considering the optimization problem

$$\left( \hat{\mathbf{W}}_t, \hat{\Phi}_t, \hat{\sigma}_{\epsilon;t}^2 \right) = \arg \min_{(\mathbf{W}, \Phi, \sigma_\epsilon^2)} \ell_t \left( \mathbf{W}, \Phi, \sigma_\epsilon^2 \right),$$

where $\ell_t \left( \mathbf{W}, \Phi, \sigma_\epsilon^2 \right) = \frac{1}{t} \left( \ln \left[ \left| \tilde{K}_{\mathcal{D}_t} + \sigma_\epsilon^2 I_t \right| \right] + \mathbf{y}_t^\top \left[ \tilde{K}_{\mathcal{D}_t} + \sigma_\epsilon^2 I_t \right]^{-1} \mathbf{y}_t \right)$. For ease of notation, we let $\mathbf{K}_t = \tilde{K}_{\mathcal{D}_t} + \sigma_\epsilon^2 I_t$ and $\mathbf{K}_t^*$ denotes the covariance matrix with the ground-truth parameter plug-in. In addition, for the parameters involved in the MGP $\Phi$, we separate them as $\Phi_K$ to denote the parameters in the kernel function and the weight parameters $a = \{a_l\}_{l=1}^m$. We generally denote $\tilde{\Phi} = \left( \Phi_K, \mathbf{W}, a, \sigma_\epsilon^2 \right)$. In the proof, the ground truth parameters or the quantities that are with ground truth parameters will be indicated by a superscript "$*$". We will also sometimes hide the subscript "$t$" that indicates the iteration for ease of notation.

**Assumption 1.** *We assume that*

1. *The set of parameters to be optimized $\tilde{\Phi} \in S_{\tilde{\Phi}}$ is a compact set. Especially, $\sigma_\epsilon^2 \in [\sigma_a^2, \sigma_b^2]$ and $\sigma_a^2 > 0$. The ground-truth parameters are contained in the set of parameters to be optimized. That is, $\tilde{\Phi}^* \in S_{\tilde{\Phi}}$.*

2. *The kernel function of $u(\mathbf{x})$ is a stationary kernel function. That is, $k(\mathbf{x}, \mathbf{x}') = k(\|\mathbf{x} - \mathbf{x}'\|)$. In addition, it is also satisfied that*

$$\max_{s=0,1,2,3} \max_{j_1,\ldots,j_s} \sup_{\tilde{\Phi} \in S_{\tilde{\Phi}}} \left| \frac{\partial^s}{\partial \tilde{\Phi}_{j_1}, \ldots, \partial \tilde{\Phi}_{j_s}} k \left( \|\mathbf{x} - \mathbf{x}'\| \right) \right| \leqslant \frac{C_{sup}}{1 + \|\mathbf{x} - \mathbf{x}'\|^{d + C_{inf}}}$$

*for some positive fixed constants $C_{inf}$ and $C_{sup}$.*

3. *The sampling strategy satisfies that , there exists a fixed constant $\Delta$*

$$\inf_{\substack{\tau,\tau'\in\mathbb{N}\\\tau\neq\tau'}} \|\mathbf{x}_\tau - \mathbf{x}_{\tau'}\| \geqslant \Delta$$

*for the decision variable; $g_l(\boldsymbol{\theta}_t) = \mathcal{O}(1/t)$ and $\frac{\partial}{\partial\mathbf{W}_j}g_l(\boldsymbol{\theta}) = \mathcal{O}(1/t)$ for the contextual variable with any $\mathbf{W}$ involved with the neural network $g(\boldsymbol{\theta})$.*

4. *The ground-truth parameters are well-separated from other potential values of parameters. That is, for $\forall\epsilon > 0$*

$$\liminf_{t\to\infty} \inf_{\substack{\tilde{\Phi}\in S_{\tilde{\Phi}}\\\|\tilde{\Phi}-\tilde{\Phi}^*\|\geqslant\epsilon}} \frac{1}{t}\sum_{\tau,\tau'=1}^{t} \left(\tilde{K}\left((\mathbf{x}_\tau,\boldsymbol{\theta}_\tau),(\mathbf{x}_{\tau'},\boldsymbol{\theta}_{\tau'})\right) - \tilde{K}^*\left((\mathbf{x}_\tau,\boldsymbol{\theta}_\tau),(\mathbf{x}_{\tau'},\boldsymbol{\theta}_{\tau'})\right) + \delta_{\tau\tau'}\left(\sigma_\epsilon^2 - \sigma_\epsilon^{2*}\right)\right)^2 > 0.$$

**Theorem 6** (Consistency of Learning NN-AGP). *Under **Assumption 1**, the training of the NN-AGP through (5) is consistent. That is,*

$$\lim_{t\to\infty} \left(\hat{\mathbf{W}}_t, \hat{\Phi}_t, \sigma^2_{\hat{\epsilon};t}\right) \xrightarrow{P} \left(\mathbf{W}^*, \Phi^*, \sigma_\epsilon^{2*}\right).$$

*Here, $\left(\mathbf{W}^*, \Phi^*, \sigma_\epsilon^{2*}\right)$ denotes the ground-truth values of the parameters in the NN-AGP model and the noise, and "$\xrightarrow{P}$" denotes the convergence in probability.*

*Proof.* Note that, the ground-truth parameters minimize the mean value of the loss function $\ell_t\left(\mathbf{W},\Phi,\sigma_\epsilon^2\right)$. That is,

$$\left(\mathbf{W}^*, \Phi^*, \sigma_\epsilon^{2*}\right) = \arg\min_{(\mathbf{W},\Phi,\sigma_\epsilon^2)} \mathbb{E}\left\{\ell_t\left(\mathbf{W},\Phi,\sigma_\epsilon^2\right)\right\}.$$

Therefore, in order to prove the consistency, we require the uniform convergence of the likelihood function, that is

$$\lim_{t\to\infty} \sup_{(\mathbf{W},\Phi,\sigma_\epsilon^2)} \left|\ell_t\left(\mathbf{W},\Phi,\sigma_\epsilon^2\right) - \mathbb{E}\left\{\ell_t\left(\mathbf{W},\Phi,\sigma_\epsilon^2\right)\right\}\right| \xrightarrow{P} 0. \tag{14}$$

To begin with, we first establish the point-wise convergence of the loss function.

$$\begin{aligned}
\mathrm{Var}\left\{\ell_t\left(\mathbf{W},\Phi,\sigma_\epsilon^2\right)\right\} &= \frac{1}{t^2}\,\mathrm{Var}\left\{\mathbf{y}_t^\top \mathbf{K}_t^{-1}\mathbf{y}_t\right\} \\
&= \frac{2}{t^2}\,\mathrm{Tr}\left\{K_t^{-1}K_t^*K_t^{-1}K_t^*\right\}.
\end{aligned}$$

For $\forall t$, the maximum eigenvalue of the matrix $K_t^*$ is bounded as

$$\begin{aligned}
\lambda_{sup}\left\{K_t^*\right\} &\leqslant \lambda_{sup}\left\{\tilde{K}^*_{D_t}\right\} + \sigma_\epsilon^{2*} \\
&\leqslant \max_{\tau=1,\ldots,t}\sum_{\tau'=1}^{t}\left|k^*\left(\mathbf{x}_\tau,\mathbf{x}_{\tau'}\right)\right|\left|\tilde{k}\left(\boldsymbol{\theta}_\tau,\boldsymbol{\theta}_{\tau'}\right)\right| + \sigma_\epsilon^{2*} \\
&\leqslant \max_{\tau=1,\ldots,t}\sum_{\tau'=1}^{t}\frac{C_{sup}}{1+\|\mathbf{x}_\tau-\mathbf{x}_{\tau'}\|^{d+C_{inf}}}\tilde{\mathcal{K}} + \sigma_\epsilon^{2*},
\end{aligned} \tag{15}$$

where $\tilde{\mathcal{K}}$ is the upper bound of $\left|\tilde{k}\left(\boldsymbol{\theta}_\tau,\boldsymbol{\theta}_{\tau'}\right)\right|$. The second inequality comes from the Gershgorin circle theorem [62]. Based on condition 2 and condition 3 in **Assumption 1**, there exists a constant $A_1$ such that $\lambda_{sup}\left\{K_t^*\right\} \leqslant A_1$; see aslo [8]. On the other hand, $\lambda_{sup}\left\{K_t^{-1}\right\} = (\lambda_{inf}\{K_t\})^{-1} \leqslant \left(\sigma_a^2\right)^{-1} = A_2$, for $\forall t, \tilde{\Phi}$. Thus, we have

$$\mathrm{Var}\left\{\ell_t\left(\mathbf{W},\Phi,\sigma_\epsilon^2\right)\right\} \leqslant \frac{2A_1^2A_2^2}{t}.$$

Thus, we have the point-wise convergence of $\ell_t \left( \mathbf{W}, \Phi, \sigma_\epsilon^2 \right)$ to its mean value. Next, we show that the convergence is uniform. To prove the uniform convergence, we consider the gradient of $\ell_t(\tilde{\Phi})$. We let $\tilde{K}_{D_t} = K_{\Phi,t} \odot K_{\mathbf{W},t}$, where "$\odot$" denotes the Hadamard product of two matrices. $K_{\Phi,t}$ and $K_{\mathbf{W},t}$ are the covariance matrix associated with $k\left(\mathbf{x}, \mathbf{x}'\right)$ and $\tilde{k}\left(\boldsymbol{\theta}, \boldsymbol{\theta}'\right)$. In this way,

$$\frac{\partial \ell_t \left( \tilde{\Phi} \right)}{\partial \tilde{\Phi}_j} = \frac{1}{t} \operatorname{tr} \left\{ K_t^{-1} \frac{\partial K_t}{\partial \tilde{\Phi}_j} \right\} - \frac{1}{t} \mathbf{y}_t^\top K_t^{-1} \frac{\partial K_t}{\partial \tilde{\Phi}_j} K_t^{-1} \mathbf{y}_t.$$

In terms of the gradient, we specifically have

$$\frac{\partial K_t}{\partial \Phi_{K;j}} = \frac{\partial K_{\Phi,t}}{\partial \Phi_{K;j}} \odot K_{\mathbf{W},t} + \sigma_\epsilon^2 I_t$$

$$\frac{\partial K_t}{\partial \mathbf{W}_j} = K_{\Phi,t} \odot 2 \begin{pmatrix} \left( \frac{\partial}{\mathbf{w}_j} \boldsymbol{g}(\boldsymbol{\theta}_1) \right)^\top a \\ \vdots \\ \left( \frac{\partial}{\mathbf{w}_j} \boldsymbol{g}(\boldsymbol{\theta}_t) \right)^\top a \end{pmatrix} \left( \boldsymbol{g}(\boldsymbol{\theta}_1)^\top a, \ldots, \boldsymbol{g}(\boldsymbol{\theta}_t)^\top a \right) + \sigma_\epsilon^2 I_t$$

$$\frac{\partial K_t}{\partial a_j} = K_{\Phi,t} \odot 2 \begin{pmatrix} \boldsymbol{g}_j(\boldsymbol{\theta}_1) \\ \vdots \\ \boldsymbol{g}_j(\boldsymbol{\theta}_t) \end{pmatrix} \left( \boldsymbol{g}(\boldsymbol{\theta}_1)^\top a, \ldots, \boldsymbol{g}(\boldsymbol{\theta}_t)^\top a \right) + \sigma_\epsilon^2 I_t$$

$$\frac{\partial K_t}{\partial \sigma_\epsilon^2} = I_t.$$

We want to show that there exists a constant $A_3$ that bounds the singular value of the gradient of $K_t$ such that

$$\rho_{sup} \left( \frac{\partial K_t}{\partial \tilde{\Phi}_j} \right) \leqslant A_3, \ \forall t.$$

Note that, given any two matrices $K_1$ and $K_2$, the singular values satisfy that

$$\rho_{sup} \left( K_1 \odot K_2 \right) \leqslant \rho_{sup} \left( K_1 \right) \rho_{sup} \left( K_2 \right)$$

and

$$\rho_{sup} \left( K_1 + K_2 \right) \leqslant \rho_{sup} \left( K_1 \right) + \rho_{sup} \left( K_2 \right),$$

see [79].

Specifically, $\rho_{sup} \left\{ \frac{\partial K_t}{\partial \Phi_{K;j}} \right\}$ is bounded by a constant, based on condition 2, as is proved in [8] with a similar argument of (15). $\rho_{sup} \left\{ K_{\Phi;t} \right\}$ is bounded by a constant with a similar argument as in (15) as well. In addition, $\sum_{\tau=1}^\infty \left[ \boldsymbol{g}(\boldsymbol{\theta}_\tau)^\top a \right]^2 < \infty$, $\sum_{\tau=1}^\infty \left[ \left( \frac{\partial}{\mathbf{w}_j} \boldsymbol{g}(\boldsymbol{\theta}_\tau) \right)^\top a \right] \left[ \boldsymbol{g}(\boldsymbol{\theta}_\tau)^\top a \right] < \infty$ and $\sum_{\tau=1}^\infty \boldsymbol{g}_l(\boldsymbol{\theta}_\tau) \left[ \boldsymbol{g}(\boldsymbol{\theta}_\tau)^\top a \right] < \infty$ based on the condition that $\boldsymbol{g}(\boldsymbol{\theta}_t) = \mathcal{O}(1/t)$ and $\frac{\partial}{\partial \mathbf{W}_j} \boldsymbol{g}_l(\boldsymbol{\theta}) = \mathcal{O}(1/t)$ in **Assumption 1**, which further bounds the maximum singular value of matrices $K_{\mathbf{W},t}$,

$$\begin{pmatrix} \left( \frac{\partial}{\mathbf{w}_j} \boldsymbol{g}(\boldsymbol{\theta}_1) \right)^\top a \\ \vdots \\ \left( \frac{\partial}{\mathbf{w}_j} \boldsymbol{g}(\boldsymbol{\theta}_t) \right)^\top a \end{pmatrix} \left( \boldsymbol{g}(\boldsymbol{\theta}_1)^\top a, \ldots, \boldsymbol{g}(\boldsymbol{\theta}_t)^\top a \right) \text{ and } \begin{pmatrix} \boldsymbol{g}_j(\boldsymbol{\theta}_1) \\ \vdots \\ \boldsymbol{g}_j(\boldsymbol{\theta}_t) \end{pmatrix} \left( \boldsymbol{g}(\boldsymbol{\theta}_1)^\top a, \ldots, \boldsymbol{g}(\boldsymbol{\theta}_t)^\top a \right). \text{ In this}$$

way, we have that $\rho_{sup} \left( \frac{\partial K_t}{\partial \tilde{\Phi}_j} \right) \leqslant A_3, \ \forall t$. Thus,

$$\max_j \sup_{\tilde{\Phi} \in S_{\tilde{\Phi}}} \left| \frac{\partial \ell_t \left( \tilde{\Phi} \right)}{\partial \tilde{\Phi}_j} \right| \leqslant A_2 A_3 + A_2^2 A_3 \frac{\mathbf{y}_t^\top \mathbf{y}_t}{t} = \mathcal{O}_p(1) \tag{16}$$

since $\frac{\mathbf{y}_t^\top \mathbf{y}_t}{t}$ is a non-negative random variable with bounded expectation. With a similar argument, we also have

$$\max_j \sup_{\tilde{\Phi} \in S_{\tilde{\Phi}}} \left| \frac{\partial \mathbb{E} \left\{ \ell_t \left( \tilde{\Phi} \right) \right\}}{\partial \tilde{\Phi}_j} \right| = \mathcal{O}(1). \tag{17}$$

Because of (16) and (17), along with the point-wise convergence, we attain the uniform convergence of the loss function $\ell_t(\tilde{\Phi})$; see [81] for detailed discussions.

To guarantee consistency of the learning procedure, we also require that the ground-truth parameters can be specified when minimizing the loss function. It can be verified that, there exists a constant $A_4$

$$
\mathbb{E}\left\{\ell_t\left(\tilde{\Phi}\right)\right\} - \mathbb{E}\left\{\ell_t\left(\tilde{\Phi}^*\right)\right\}
$$
$$
\geqslant A_4 \frac{1}{t} \sum_{\tau,\tau'=1}^{t} \left( \tilde{K}\left((\mathbf{x}_\tau, \boldsymbol{\theta}_\tau), (\mathbf{x}_{\tau'}, \boldsymbol{\theta}_{\tau'})\right) - \tilde{K}^*\left((\mathbf{x}_\tau, \boldsymbol{\theta}_\tau), (\mathbf{x}_{\tau'}, \boldsymbol{\theta}_{\tau'})\right) + \delta_{\tau\tau'}\left(\sigma_\epsilon^2 - \sigma_\epsilon^{2*}\right)\right)^2
$$

where the detailed proof can be found in [7]. In this way, based on condition 4 in **Assumption 1**, for $\forall \epsilon > 0$,

$$
\liminf_{t\to\infty} \inf_{\substack{\tilde{\Phi}\in S_{\tilde{\Phi}} \\ \|\tilde{\Phi}-\tilde{\Phi}^*\|\geqslant\epsilon}} \mathbb{E}\left\{\ell_t\left(\tilde{\Phi}\right)\right\} - \mathbb{E}\left\{\ell_t\left(\tilde{\Phi}^*\right)\right\} \geqslant A_5,
$$

for some constant $A_5$. Thus, along with the uniform consistency of the loss function (14), we attain the consistency of the training procedure (5), which follows a regular argument on consistency of $M$-estimation; see [101].

$\square$

# 9 Experimental details & additional experiments

## 9.1 Experimental details

In this section, we describe the experiment settings in the main context in detail. In terms of the training of NN-AGP through (5) and maximizing the likelihood function of a joint GP, we apply the alternating direction method of multipliers (ADMM) with a learning rate $10^{-4}$; see [20]. All the experiments are based on running PyTorch and Python 3.8 on Nvidia GeForce RTX 3090 (GPU) with 24GB of RAM. An implementation is provided at `https://github.com/Oceanjinghai/NN-AGP-UCB`.

### 9.1.1 Synthetic reward

In terms of the joint GP model, we consider both additive kernels and multiplicative kernels, of which each separate kernel is the radial basis function (RBF) kernel. In terms of the NN-AGP model, we select $m = 2, 3, 5$. Besides, we select the ICM model with the RBF kernel as the MGP component ($Q = 1$) and an FCN with 2 hidden layers with 64 and 32 nodes. The parameters of the MGP ($a_{l,q}$'s) are updated through learning the NN-AGP model in (5).

In addition, both NeuralUCB [121] and NN-UCB [65] are designed for contextual bandits with $K$ arms. To adapt them into the problem we consider in Section 2, we take $\mathbf{z} = (\boldsymbol{\theta}, \mathbf{x})$ as a joint input to the neural networks representing the arm. We discretize the joint space $\Theta \times \mathcal{X}$ with 10 points in each dimension with equal distances. The best arm is selected with some of the dimension fixed by $\boldsymbol{\theta}_t$. In terms of NN's used in both algorithms, we select an FCN with 3 hidden layers of 64, 32, and 32 nodes.

### 9.1.2 Queuing problem with time sequence contextual variables

We consider a discrete-time queuing problem, where decision-making in each time period is required. In each epoch, a contextual variable $\boldsymbol{\theta}_t$ is first revealed to the agent. In some application scenarios, the contextual variable might includes traffic conditions and weather conditions that affect the arrival process of the queuing system. The number of customers who will come to the queue, denoted as $N_t$ is drawn from a Poisson distribution $\mathrm{Poisson}(u_t)$. Here $u_t = \exp\left(\sum_{\tau=1}^{t} a^\top \boldsymbol{\theta}_\tau\right)$, and $a \sim \mathcal{N}(\mathbf{1}/4, \frac{1}{4}I_{d'})$ is the weight generated and fixed in advance. In this way, the number of customers who will come to the queue in this round depends on the entire sequence of contextual variables.

On the other hand, the decision variable is composed of two parts $p_{1;t}$ and $p_{2;t}$, denoting the service price and the service rate respectively. The service price indicates the reward that completes serving a customer. On the other hand, a customer comes to queue, sees the service price, and then determines to join the queue with the probability of $p(p_{1;t})$, where $p(\cdot)$ is a decreasing function with respect to $p_{1;t}$. In addition, in each iteration, the number of service completion $N'(t)$ is a Poisson random variable with the mean of the service rate $p_{2;t}$, that is $N'(t) \sim \text{Poisson}(p_{2;t})$. The higher the service rate, the higher the service cost will be. After each iteration, the customers who decide to join the queue and do not receive the service will leave as well, resulting in a penalty. In this way, the observed reward in each iteration is

$$y_t = p_{1;t} \max\{N(t), N'(t)\} - c_1 p_{2;t} - c_2 \max\{N(t) - N'(t), 0\},$$

where on the right-hand side the first term is the reward of completing service, the second term is the service cost and the last term denotes the penalty of not satisfying customers. Such decision problems in a queuing system are also considered in [29]. For the NN-AGP model, we select the long short-term memory (LSTM) [58] neural network to approximate the mapping $g_t(\boldsymbol{\theta}_1, \ldots, \boldsymbol{\theta}_t)$. The training of LSTM (as well as MGP) is accomplished by maximizing the likelihood function. Since we do not have the ground-truth value of the expected maximum reward, we instead record the cumulative rewards to compare the performance of different methods.

Specifically, in terms of the experiment results contained in the main text, we set $c_1 = 0.5$ and $c_2 = 0.3$. In terms of the NN-AGP model, we select $m = 2, 3, 5$. Besides, we select the ICM model with the RBF kernel as the MGP component and an LSTM with 1) sequence length = 10; 2) hidden size = 64; 3) projection size = $m$; 4) batch size = 1. We utilize the implementation from https://pytorch.org/docs/stable/generated/torch.nn.LSTM.html. Besides, we select the ICM model with the RBF kernel as the MGP component ($Q = 1$). For CGP-UCB, we utilize the RBF kernel for the scenario when we only utilize the current contextual variable. We also apply the Wasserstein subsequence kernel [13] that is specifically designed for time series, of which the implementation can be found at https://github.com/BorgwardtLab/WTK.

### 9.1.3 Pricing with a diffusion network

We consider a pricing problem with a diffusion network, which is explored in [77]. Specifically, we represent the network at time $t$ as a graph $\boldsymbol{\theta}_t = (V_t, E_t)$, where $V_t := \{1, 2 \ldots, |V_t|\}$ is the set of all users (nodes) and $E_t := \{1, 2 \ldots, |E_t|\}$ is the set of all directed edges. A directed edge $(i, j) \in E_t$, where $i, j \in V$, implies that user $i$ is influenced by user $j$, and we call $j$ an in-neighbor of $i$. We use $\mathcal{N}_{i;t}$ to denote the set of all in-neighbors for agent $i$ at time $t$ and $n_{i;t} := |\mathcal{N}_{i;t}|$ to denote her in-degree (i.e., the number of in-neighbors).

In each iteration, the user $i \in V$ will decide whether to adopt the service based on her realized utility in period $t$ : $Y_i(t) := \mathbb{I}\{u_i(t) \geq 0\} \in \{0, 1\}$, where $u_i(t)$ is the utility of user $i$ to adopt the service in period $t$, and is defined as

$$u_i(t) = v_i - \alpha \mathbf{x}_t + \beta \cdot \frac{\sum_{j \in \mathcal{N}_{i;t}} Y_j(t-1)}{n_{i;t}} + \epsilon_i(t).$$

Here $\mathbf{x}_t$ is the service price (decision variable); $v_i$ denotes the user (node) preference while $\alpha$ and $\beta$ are intrinsic network parameters; and $\epsilon_t$ is the i.i.d. Gaussian noise. In each iteration, the graph structure $\boldsymbol{\theta}_t = (V_t, E_t)$ is presented to the agent to determine a price $\mathbf{x}_t$. In this experiment, we consider maximizing the total profit brought by users' service adoption in the network, and therefore the reward function is

$$f(\mathbf{x}_t; \theta) = \mathbf{x}_t \times \mathbb{E}\left[\sum_i Y_i(t; \theta)\right],$$

where $x_t$ denotes the price of the service. An increase in prices is likely to have a negative impact on the adoption rate of service.

For the NN-AGP model, we select the graph convolutional neural network (GCN) [92] to approximate the mapping $g(\boldsymbol{\theta})$. The training of GCN (as well as MGP) is accomplished by maximizing the likelihood function. Since we do not have the ground-truth value of the expected maximum reward, we instead record the cumulative rewards to compare the performance of different methods.

In terms of the experiment results in the main text, we let $\mathcal{X} = [0, 30]$ and $\boldsymbol{\theta}_t$ represents an undirected graph with 5 and 10 nodes where each edge exists with probability 1/2. Besides, we set $\alpha = \beta = 1$

| | 50-th round | 100-th round | 300-th round |
|---|---|---|---|
| NN-AGP-UCB (m=2) | 0.07/0.25 | 0.10/0.81 | 0.27/1.31 |
| NN-AGP-UCB (m=5) | 0.11/0.29 | 0.14/0.89 | 0.35/1.42 |
| CGP-UCB (additive kernel) | 0.01/0.26 | 0.02/0.77 | 0.04/1.24 |
| NN-UCB | 0.14/2.28 | 0.35/4.13 | 0.40/6.51 |
| NeuralUCB | 0.11/1.64 | 0.23/3.62 | 0.37/5.45 |

Table 1: The mean of training time/ execution time of bandit algorithms associated with the first set of experiments in Section 4.1.

and $v_i = 3$ for each node. In terms of the NN-AGP model, we select $m = 3$. Besides, we select the ICM model with the RBF kernel as the MGP component and a GCN with convolution size = 3. We utilize the implementation from https://pytorch-geometric.readthedocs.io/en/latest/modules/nn.html. Besides, we select the ICM model with the RBF kernel as the MGP component ($Q = 1$). For CGP-UCB, we utilize the RBF kernel for the vectorized contextual variable. We also apply the Gaussian RBF kernel between vertex histogram [59] that is specifically designed for graphs. The experimental results indicate that our NN-AGP-UCB has a greater advantage than CGP-UCB when the contextual variable exhibits more complexity (with more nodes).

## 9.2 Additional experiments

### 9.2.1 Computational time

In this section, we record the computational time of the algorithms. We record 1) the training time that constructs the surrogate model based on the historical data and 2) the execution time that selects the decision variable after the contextual variable is revealed. We record the time (seconds) for exactly one round in the 50-th, 100-th, and 300-th rounds. We take the first set of experiments in Section 4.1 as an example and present the results in **Table 1**.

We notice that CGP-UCB is the most efficient in training time since it employs a pre-specified GP model which does not update during iterations. The training procedure of CGP-UCB only requires matrix operations, which can be implemented efficiently. On the other hand, all the algorithms that involve NN require learning NN from data and longer training time than CGP-UCB. In terms of the execution time, NN-AGP-UCB requires similar time as CGP-UCB, since the selection of the decision variable of NN-AGP-UCB is based on GP as well. We also note that both NN-UCB and NeuralUCB are initially designed for finite selections of decision variables. Thus, the computational cost of the execution time is largely due to searching for the optimal decision variable from the discretized feasible set. In addition, we consider sparse NN-AGP to alleviate the computational burden for future work; see also a discussion in Section 10.1.

### 9.2.2 Sensitivity on reward function structure

As suggested by [68], commonly selected composite kernel functions of the joint Gaussian process in CGP-UCB are additive kernels and multiplicative kernels. In this section, we show through experiments that the performance of CGP-UCB is sensitive on whether the form of the composite kernel is consistent with the structure of the reward function, while our NN-AGP-UCB achieves acceptable performance through the experiments.

Specifically, we consider two synthetic reward functions in the form of

$$R_3(\mathbf{x}, \boldsymbol{\theta}) = \sin\left(\|\mathbf{x}\|_2\right) \left|\cos\left(\|\boldsymbol{\theta}\|_2\right)\right|;$$
$$R_4(\mathbf{x}, \boldsymbol{\theta}) = \sin\left(\|\mathbf{x}\|_2\right) + \cos\left(\|\boldsymbol{\theta}\|_2\right).$$

That is, $R_3(R_4)$ is a multiplicative(additive) function with contextual/decision variables. We consider $d = 2$ and $d' = 3$. We let $\mathcal{X} = [-\sqrt{2}, \sqrt{2}]^2$ and $\Theta = [-1, 1]^3$. In terms of the joint GP model, we consider both additive kernels and multiplicative kernels, of which each separate kernel is the radial basis function (RBF) kernel. In terms of the NN-AGP model, we select $m = 3$. Besides, we select

the ICM model with the RBF kernel as the MGP component and an FCN with 2 hidden layers of 64 and 32 nodes.

The experiment results (mean performance of 15 times experiments) are contained in **Figure 7** and **Figure 8**, which provide the insights as follows. First, for both reward functions, the NN-AGP-UCB approach does not outperform the classical CGP-UCB approach in initial iterations, since the neural networks require sufficient data to learn. Second, as the size of the data increases, NN-CGP-UCB outperforms CGP-UCB in both experiments, owing to the strong approximation power of FCN. Last, the performance of the CGP-UCB is sensitive to the form of the composite kernels and the structure of reward functions. That is, for reward $R_3(\mathbf{x}, \boldsymbol{\theta})$ with a multiplicative structure, CGP-UCB with the multiplicative kernel outperforms CGP-UCB with the additive kernel, while for reward $R_4(\mathbf{x}, \boldsymbol{\theta})$ with an additive structure, CGP-UCB with the additive kernel outperforms CGP-UCB with the multiplicative kernel. On the other hand, the performance of the NN-AGP-UCB remains acceptable and outperforms baseline approaches no matter the structure of the reward function.

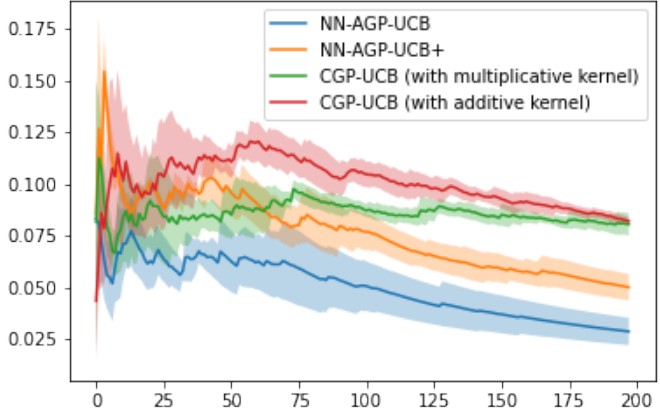

Figure 7: Average regrets of NN-AGP-UCB and CGP-UCB with multiplicative reward function $R_3(\mathbf{x}; \boldsymbol{\theta})$.

### 9.2.3 Advantage with higher-dimensional contextual variables

In the previous section, we present the experimental results when $d = 1$ and $d' = 3$. Here, in terms of the additive reward function $R_4$, we increase the dimension of the contextual variable $d'$ and present the results (mean performance of 15 times experiments) in **Figure 8**, **Figure 9** and **Figure 10**. Experimental results indicate that the superiority of our NN-AGP-UCB becomes more significant as the dimension of the contextual variable increases, considering the larger gaps (the scale of vertical axis in each figure is different) between the average regrets.

Moreover, we also contain the results of NN-AGP-UCB+ in **Figure 7**, **Figure 8**, **Figure 9** and **Figure 10**. The experimental results indicate that NN-AGP-UCB+ does not generally outperform NN-AGP-UCB, since NN-AGP-UCB+ is overly-conservative. On the other hand, NN-AGP-UCB+ still outperforms the baseline CGP-UCB with both multiplicative/additive kernels.

In addition, we also conduct additional experiments with higher-dimensional contextual variables, while the reward function exhibits a sparse structure. We consider that the observed contextual variable $\boldsymbol{\theta}$ are randomly selected with equal probability from $\Theta = [-1/2, 1/2]^{50}$. That is $d' = 50$. Meanwhile, we select the reward function

$$\tilde{R}_3(\mathbf{x}, \boldsymbol{\theta}) = 2\sin\left(\|\mathbf{x}\|_2\right)|\cos\left(\|\boldsymbol{\theta}_{\text{eff}}\|_2\right)|$$

as a sparse version of the reward function $R_3(\mathbf{x}, \boldsymbol{\theta})$. Here, $\mathbf{x} \in [-\sqrt{2}, \sqrt{2}]^2$ and $\boldsymbol{\theta}_{\text{eff}}$ denotes the first 20 dimensions of $\boldsymbol{\theta}$. That is, the remaining 30 dimensions of $\boldsymbol{\theta}$ will not affect the reward function while the user does not know. The experimental results are contained in **Figure 11**. The results on average regret indicate the superiority of our approach in high-dimensional scenarios, even when

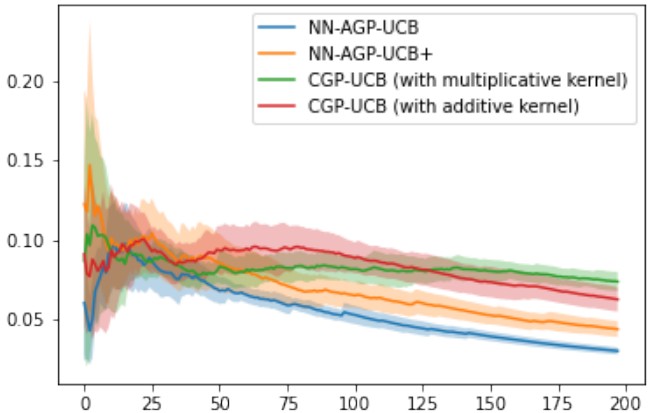

Figure 8: Average regrets of NN-AGP-UCB and CGP-UCB with additive reward function $R_4(\mathbf{x}; \boldsymbol{\theta})$ when $d' = 3$.

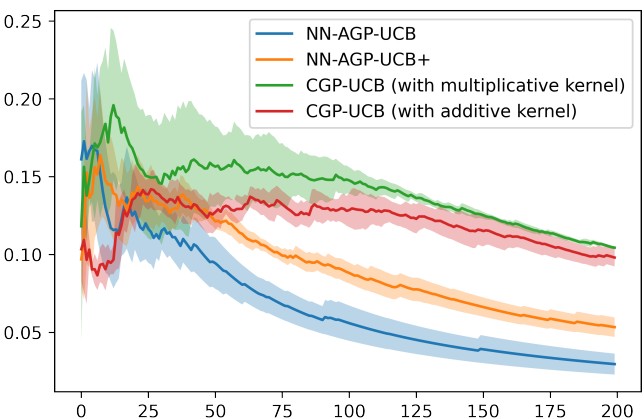

Figure 9: Average regrets of NN-AGP-UCB and CGP-UCB with additive reward function $R_4(\mathbf{x}; \boldsymbol{\theta})$ when $d' = 6$.

the dimension of the NN-AGP model $m \ll d'$. That is to say, by utilizing the neural network, the NN-AGP model effectively extracts the information from the contextual variable and propagates it to the MGP component.

### 9.2.4 Regression tasks with complex functions

As is discussed in the main context, the NN-AGP model inherits the strong approximation power from neural networks, which leads to the better performance on contextual GP bandit problems. To support this intuition, we conduct experiments to compare the prediction performance of NN-AGP and a joint GP. That is, we select in advance all the points to be samples and attain the observations. We then use these observations to train both NN-AGP and a joint GP. For both models, the prediction value of the unknown function is the posterior mean. Here, we select the Ackley function [2] as a representative to be approximated

$$f(\mathbf{x}; \boldsymbol{\theta} = (a, b, c)) = -a \exp\left(-b\sqrt{\frac{1}{d}\sum_{i=1}^{d} \mathbf{x}_{(i)}^2}\right) - \exp\left(\frac{1}{d}\sum_{i=1}^{d} \cos\left(c\mathbf{x}_{(i)}\right)\right) + a + \exp(1).$$

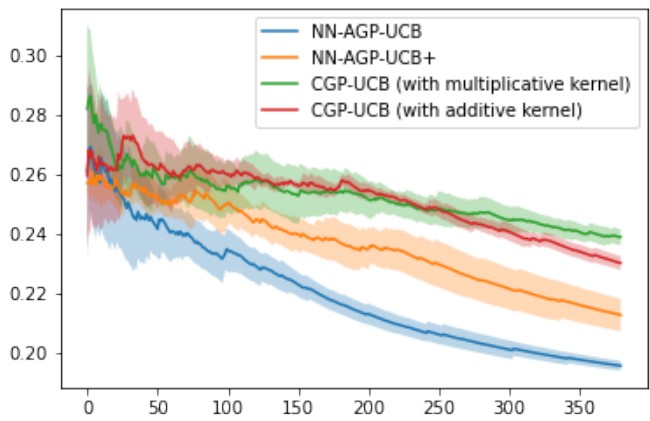

Figure 10: Average regrets of NN-AGP-UCB and CGP-UCB with additive reward function $R_4(\mathbf{x}; \boldsymbol{\theta})$ when $d' = 9$.

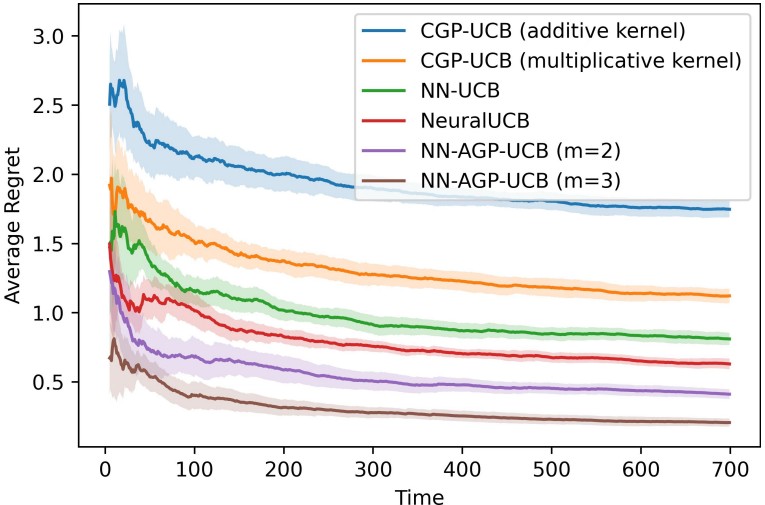

Figure 11: Average regrets of NN-AGP and baseline approaches with the high-dimensional reward function $\tilde{R}_3(\mathbf{x}; \boldsymbol{\theta})$.

A plot of the Ackley function is contained in **Figure 12**. We let $\mathcal{X} = [-32.768, 32.768]^2$. In terms of the contextual variable, we set $a \in [15, 25], b \in [0.15, 0.25]$ and $c \in [1.5\pi, 2.5\pi]$. In terms of the joint GP model, we consider both additive kernels and multiplicative kernels, of which each separate kernel is the radial basis function (RBF) kernel. In terms of the NN-AGP model, we select $m = 3$. Besides, we select the ICM model with the RBF kernel as the MGP component ($Q = 1$) and an FCN with 2 hidden layers with 64 and 32 nodes.

We present the experimental results in **Figure 13**. The experiments are performed 15 times and the experimental results indicate that our NN-AGP model achieves a better performance on the approximation accuracy, which is quantified by rooted mean square error (RMSE) as well as the corresponding standard deviation.

Since the NN-AGP model achieves a better performance in approximating the highly-nonstructural function, we would expect that NN-AGP-UCB would also achieve a better performance in contextual GP bandits when the reward function is highly-nonstructural as well.

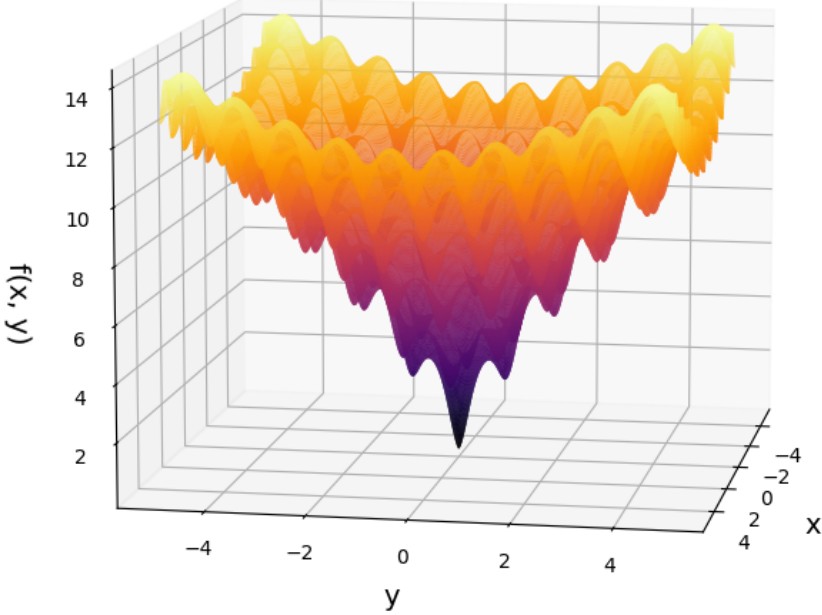

Figure 12: An Ackley function $f(x, y) = -20 \exp\left(-0.2\sqrt{0.5\left(x^2 + y^2\right)}\right) - \exp(0.5(\cos(2\pi x) + \cos(2\pi y))) + e + 20$.

### 9.2.5 Air-quality monitoring sites

In this set of experiments, we consider sequentially selecting the site that will record the worst air-quality, among multiple air-quality monitoring sites. That is, each $\mathbf{x}$ denotes an air-quality monitoring site and $|\mathcal{X}|$ denotes the number of sites. We use the data collected by Beijing Municipal Environmental Monitoring Center[2]; see also [120]. The data set includes hourly air pollutants data from 12 nationally-controlled air-quality monitoring sites ($|\mathcal{X}| = 12$). The time period is from March 1st, 2013 to February 28th, 2017. The recorded quantities in each iteration include

- PM2.5: PM2.5 concentration (ug/m$^3$)
- PM10: PM10 concentration (ug/m$^3$)
- SO2: SO2 concentration (ug/m$^3$)
- NO2: NO2 concentration (ug/m$^3$)
- CO: CO concentration (ug/mm$^3$)
- O3: O3 concentration (ug/m$^3$)
- TEMP: temperature (degree Celsius)
- PRES: pressure (hPa)
- DEWP: dew point temperature (degree Celsius)

---

[2]The data can be found at https://archive.ics.uci.edu/ml/datasets/Beijing+Multi-Site+Air-Quality+Data.

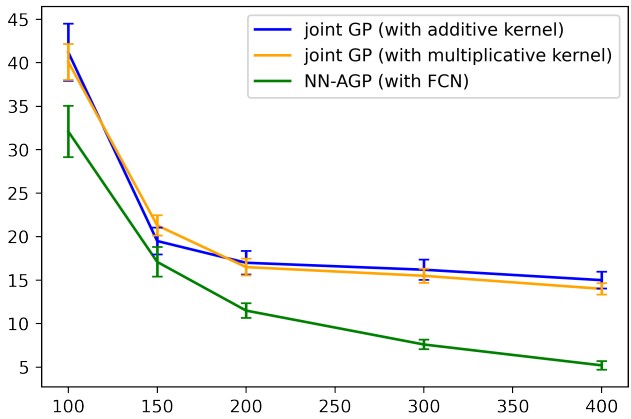

Figure 13: RMSE's of NN-AGP and joint GP with Ackley function.

- RAIN: precipitation (mm)
- wd: wind direction
- WSPM: wind speed (m/s).

In our experiment, we regard PM 2.5 as the unknown reward and the remaining quantities as the observed contextual variables in each round, that is $d' = 11$. That is, we would like to select the site that records the largest PM 2.5. In terms of the decision variable, we simulate an $\mathbf{x}^{(i)} \sim \mathrm{Unif}(0,1)$ for $i = 1, 2, \ldots, |\mathcal{X}|$, and use this generated random number to represent the site in all rounds. That is, $\mathcal{X} = \left\{ \mathbf{x}^{(1)}, \mathbf{x}^{(2)}, \ldots, \mathbf{x}^{(|\mathcal{X}|)} \right\}$. Since PM 2.5 in all the sites is contained in the data set, the regret in each round is then the maximum PM 2.5 minus the PM 2.5 recorded in the selected site.

The setting of the approaches (NN-AGP-UCB, CGP-UCB, NN-UCB and NeuralUCB) is consistent with that in Section 9.1.1. The experiment results are presented in **Figure 14**. The experiments are performed 15 times. Although we use a same data set, the uncertainty comes from randomly selecting the decision variables for initialization. The experimental results indicate that our approach is applicable to real-world applications when the selection of the decision variable is finite, and outperforms existing approaches. We also note that, all the compared approaches exhibit fluctuation at the same time since the air-quality is influenced by human factors in certain period.

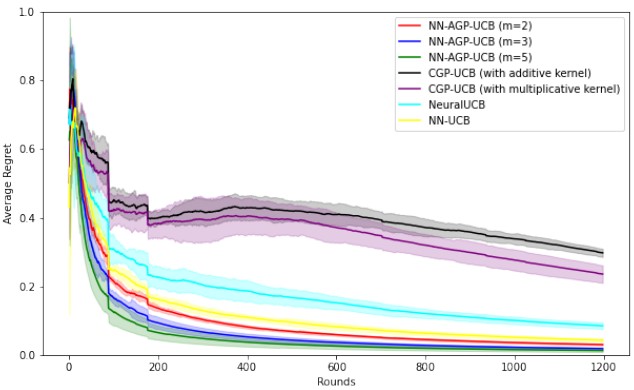

Figure 14: Average regrets of NN-AGP-UCB and baseline approaches with the real-data collected from air-quality monitoring sites.

# 10 Limitations & future work

In this section, we describe the limitations of NN-AGP, and propose potential future work to address these limitations.

## 10.1 Sparse NN-AGP

As is widely known, the Gaussian process model suffers from a computational complexity of $\mathcal{O}\left(t^3\right)$ ($t$ denotes the sample size of data) and the NN-AGP model encounters the same challenge as well, because of the GP expression with respect to the decision variable. In this way, we briefly introduce the sparse NN-AGP model, which alleviates the computational burdens of the NN-AGP. A more detailed discussion on this direction will be contained in future work.

In terms of the MGP in the NN-AGP, we specifically consider the scenario when $Q = 1$ and there is no $v_l$'s. Denote $\mathbf{p} = \left(\mathbf{p}\left(\mathbf{x}_1\right), \mathbf{p}\left(\mathbf{x}_2\right), \ldots, \mathbf{p}\left(\mathbf{x}_N\right)\right)^\top$. In addition, $\mathbf{u} = \left(\mathbf{p}\left(\mathbf{z}_1\right), \mathbf{p}\left(\mathbf{z}_2\right), \ldots, \mathbf{p}\left(\mathbf{z}_M\right)\right)^\top$ are inducing points of the MGP on pseudo-inputs $\mathbf{Z} = \{\mathbf{z}_m\}_{m=1}^M$. In this way,

$$\left(\begin{array}{c} \mathbf{p} \\ \mathbf{u} \end{array}\right) \sim \mathcal{MN}\left(\left(\begin{array}{c} \mathbf{0} \\ \mathbf{0} \end{array}\right), \left(\begin{array}{cc} \mathbf{K_{pp}} & \mathbf{K_{up}^\top} \\ \mathbf{K_{up}} & \mathbf{K_{uu}} \end{array}\right), \mathbf{A}\right).$$

That is, the matrix $\left(\begin{array}{c} \mathbf{p} \\ \mathbf{u} \end{array}\right)$ is sampled from a matrix-variate normal distribution; see [55]. Here $\mathbf{K}_{(\,\cdot\,,\,\cdot\,)}$ is the covariance matrix generated by the kernel function $k\left(\mathbf{x}, \mathbf{x}'\right)$. Meanwhile, $\mathbf{A} = \mathbf{a}\mathbf{a}^\top$, where $\mathbf{a} = (a_1, a_2, \ldots, a_m)$.

**Remark 1.** *A matrix-valued random element* $\mathbf{X} \sim \mathcal{MN}\left(\mathbf{M}, \mathbf{U}, \mathbf{V}\right)$ *is equivalent with that*

$$\mathrm{vec}\left\{\mathbf{X}\right\} \sim \mathcal{N}\left(\mathrm{vec}\left\{\mathbf{M}\right\}, \mathbf{V} \otimes \mathbf{U}\right),$$

*where* $\mathrm{vec}\{\,\cdot\,\}$ *is the "vectorize" operator. Both* $\mathbf{U}$ *and* $\mathbf{V}$ *serve as the covariance matrix, where* $\mathbf{U}$ *captures the covariance among rows (samples) while* $\mathbf{V}$ *captures that among columns (dimensions). Meanwhile, for the p.d.f. of* $\mathbf{X}$, *we denote is as* $p(\mathbf{X}) = \mathcal{MN}\left(\mathbf{X} \mid \mathbf{M}, \mathbf{U}, \mathbf{V}\right)$. *This notation is also used for the multivariate Gaussian case.*

Define $\psi_{\mathbf{u}}\left(\mathbf{x}\right) = \mathbf{K_{uu}^{-1}}k_{\mathbf{u}}(\mathbf{x})$, where $k_{\mathbf{u}}(\mathbf{x})$ denotes the covariance vector between $\mathbf{p}(\mathbf{x})$ and $\mathbf{u}$. Meanwhile, let

$$\mathbf{\Phi} = \left(\psi_{\mathbf{u}}\left(\mathbf{x}_1\right), \psi_{\mathbf{u}}\left(\mathbf{x}_2\right), \ldots, \psi_{\mathbf{u}}\left(\mathbf{x}_N\right)\right).$$

We then have

$$p\left(\mathbf{p} \mid \mathbf{u}\right) = \mathcal{MN}\left(\mathbf{p} \mid \mathbf{\Psi}^\top\mathbf{u}, \mathbf{K_{pp}} - \mathbf{\Psi}^\top\mathbf{K_{uu}}\mathbf{\Psi}, \mathbf{A}\right).$$

Supppose that a variational prior is imposed on the inducing points as

$$q_v\left(\mathbf{u}\right) = \mathcal{MN}\left(\mathbf{u} \mid \mathbf{B}, \mathbf{L}\mathbf{L}^\top, \mathbf{A}\right).$$

The selection of $\mathbf{B}$ and $\mathbf{L}$ is postponed. Based on the conditional distribution $p(\mathbf{p} \mid \mathbf{u})$, we have the joint variational distribution $q_v(\mathbf{p}, \mathbf{u}) = p\left(\mathbf{p} \mid \mathbf{u}\right)q_v(\mathbf{u})$. By marginalizing, we have

$$q_v\left(\mathbf{p}\right) = \mathcal{MN}\left(\mathbf{p} \mid \mathbf{\Psi}^\top\mathbf{B}, \mathbf{K_{pp}} - \mathbf{\Psi}^\top\left(\mathbf{K_{uu}} - \mathbf{L}\mathbf{L}^\top\right)\mathbf{\Psi}, \mathbf{A}\right) \tag{18}$$

With the variational distribution of $q_v\left(\mathbf{p}\right)$, the inference of the MGP at any new point $\mathbf{x}^*$ with a given $\boldsymbol{\theta}$ is

$$\hat{f}(\mathbf{x}^*; \boldsymbol{\theta}) \sim \mathcal{N}\left(\boldsymbol{g}(\boldsymbol{\theta})^\top\psi_{\mathbf{u}}\left(\mathbf{x}^*\right)^\top\mathbf{B}, \boldsymbol{g}(\boldsymbol{\theta})^\top\left(k\left(\mathbf{x}^*, \mathbf{x}^*\right) - \psi_{\mathbf{u}}\left(\mathbf{x}^*\right)^\top\left(\mathbf{K_{uu}} - \mathbf{L}\mathbf{L}^\top\right)\psi_{\mathbf{u}}\left(\mathbf{x}^*\right)\right)\mathbf{A}\boldsymbol{g}(\boldsymbol{\theta})\right).$$

In this way, the inference at a new point $(\boldsymbol{\theta}, \mathbf{x}^*)$ requires the computational complexity of $\mathcal{O}\left(tM^2\right)$ instead of $\mathcal{O}\left(t^3\right)$. Next, we describe the procedure of deciding the parameters of the variational prior $q_v(\mathbf{u})$, that is $(\mathbf{B}, \mathbf{L})$, and the location of inducing points $\mathbf{Z}$. The selection is through the variational inference approach, which minizes the kullback-leibler(KL)-divergence between $q_v\left(\mathbf{p}, \mathbf{u}\right)$ and $p\left(\mathbf{p}, \mathbf{u} \mid \mathbf{y}\right)$. Specifically

$$\begin{aligned}
\mathrm{KL}\left[q_v(\mathbf{p}, \mathbf{u})\|p(\mathbf{p}, \mathbf{u} \mid \mathbf{y})\right] &= \mathbb{E}_{q_v(\mathbf{p}, \mathbf{u})}\left[\log\frac{q_v(\mathbf{p}, \mathbf{u})}{p(\mathbf{p}, \mathbf{u} \mid \mathbf{y})}\right] \\
&= \log p(\mathbf{y}) + \mathbb{E}_{q_v(\mathbf{p}, \mathbf{u})}\left[\log\frac{q_v(\mathbf{p}, \mathbf{u})}{p(\mathbf{p}, \mathbf{u}, \mathbf{y})}\right] \\
&= \log p(\mathbf{y}) - \mathrm{ELBO}(\boldsymbol{v}, \mathbf{Z}),
\end{aligned}$$

where the evidence lower bound (ELBO) is defined as

$$\text{ELBO}(v, \mathbf{Z}) \triangleq \mathbb{E}_{q_v(\mathbf{p}, \mathbf{u})} \left[ \log \frac{p(\mathbf{p}, \mathbf{u}, \mathbf{y})}{q_v(\mathbf{p}, \mathbf{u})} \right].$$

Since $\log p(\mathbf{y})$ is fixed and not affected by the variational distribution, minimizing the KL-divergence is then equivalent with maximizing ELBO, which is further decomposed as

$$\text{ELBO}(v, \mathbf{Z}) = \mathbb{E}_{q_v(\mathbf{p})}[\log p(\mathbf{y} \mid \mathbf{p})] - \text{KL}\left[q_v(\mathbf{u}) \| p(\mathbf{u})\right].$$

Note that

$$\log p(\mathbf{y} \mid \mathbf{p}) = \log p(\mathbf{y} \mid \mathbf{p}) = -\frac{N}{2} \log \left(2\pi\sigma_\epsilon^2\right) - \frac{1}{2\sigma_\epsilon^2} \left(\mathbf{y} - \mathbf{f}\right)^\top \left(\mathbf{y} - \mathbf{f}\right),$$

where $\mathbf{f} = \left(g(\boldsymbol{\theta}_1)^\top \mathbf{p}(\mathbf{x}_1), g(\boldsymbol{\theta}_2)^\top \mathbf{p}(\mathbf{x}_2), \ldots, g(\boldsymbol{\theta}_N)^\top \mathbf{p}(\mathbf{x}_N)\right)^\top$ is not a linear function with respect to $\mathbf{p}$. Thus, we employ the Markov chain Monte Carlo method to evaluate $\mathbb{E}_{q_v(\mathbf{p})}[\log p(\mathbf{y} \mid \mathbf{p})]$. In addition, the KL-divergence adopts a closed-form expression as

$$\text{KL}\left[q_v(\mathbf{u}) \| p(\mathbf{u})\right] = \frac{1}{2}\left(\text{vec}\left\{\mathbf{B}\right\}^\top \text{vec}\left\{\mathbf{K}_{\mathbf{uu}}^{-1}\mathbf{B}\mathbf{A}^{-1}\right\} + m\,\text{tr}\left\{\mathbf{K}_{\mathbf{uu}}^{-1}\mathbf{L}\mathbf{L}^\top\right\} - m\ln\frac{\left|\mathbf{L}\mathbf{L}^\top\right|}{\left|\mathbf{K}_{\mathbf{uu}}\right|} - Mm\right).$$

In this way, the stochastic gradient descent method can be employed to maximize ELBO. We note that the locations of the inducing points $\mathbf{u}$ can also be optimized as well. For more detailed discussions on the sparse Gaussian process, we refer to [99, 57]. In addition, we compare sparse NN-AGP with sparse joint GP with contextual/decision variables. In terms of NN-AGP, the sparsity is built on a GP where the input dimension is $d$. In comparison, for the joint GP, the sparsity is built on a GP where the input dimension is $d + d'$. Intuitively, sparse NN-AGP requires fewer inducing points to achieve a prescribed accuracy, which alleviates the computational complexity. We admit that further discussions are required in future work.

## 10.2 Transfer learning with NN-AGP

It is widely accepted that incorporating NN into bandit problems generally requires sufficient data to approximate the unknown reward function. Thus, the cold-start issue is brought to, in principle, all bandit algorithms that uses NN. Compared with the algorithms that fully rely on NN (e.g., NeuralUCB and NN-UCB), our NN-AGP-UCB actually suffers less from the cold-start issue, which is supported by numerical results in Section 4.1. The reason is that, in existing NN-based bandit algorithms, NN is responsible for approximating the entire reward function. In comparison, for the NN-AGP model, NN is focused to only be used for approximating the mapping from the contextual variable to the reward function and the approximation regarding the decision variable is supported by GP. It has been widely accepted that GP generally requires less data than NN in practical applications, and therefore NN-AGP helps to ease the cold-start issue.

Moreover, to further address the cold-start issue brought to NN-AGP, the transfer learning technology [106] can be incorporated into the bandit algorithm. We conduct numerical experiments and present the results in **Figure 15**. Specifically, we consider an unknown reward function $f_T$ and we also have access to functions $f_s, s = 1, 2, \ldots, 5$ that have a similar structure with $f_T$. We first sample each $f_s$ for 50 or 100 rounds and learn an NN-AGP model with these samples. The NN component in NN-AGP helps to transfer the knowledge from $f_s$ to $f_T$. That is, during the initial rounds of NN-AGP-UCB with $f_T$, we first fix the input layer of the pretrained NN and update the remaining layers with the new data, which is a widely-used transfer learning method named freezing.

Experimental results indicate that transfer learning from similar tasks helps to address the cold-start issue, and NN-AGP-UCB with/without transfer learning will converge to the similar regrets as the round increases. We also not that, to the best of our knowledge, there has not been sufficient work on transfer learning with NN-based bandit algorithms. These NN-based bandit algorithms largely rely on the neural tangent kernel (NTK) to address the exploration-exploitation trade-off when selecting the decision variable. However, it remains an open question on how to transfer the knowledge between different domains with NTK. In comparison, the exploration-exploitation trade-off in NN-AGP-UCB is supported by GP, and existing transfer learning technologies with NN can be easily adapted into our algorithm. Other methodologies for addressing cold-start in learning NN in an online setting can also be employed; see [105, 108].

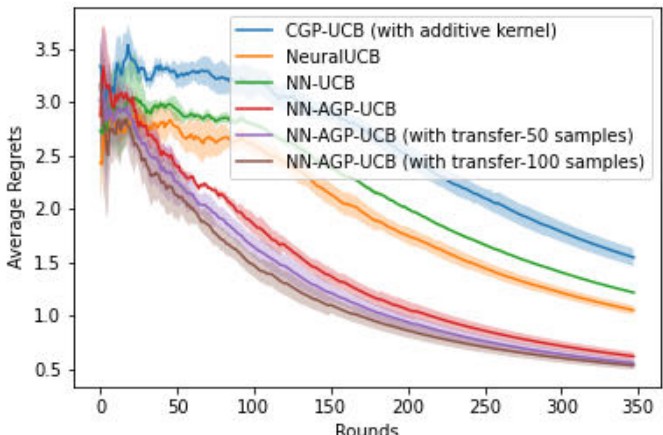

Figure 15: Average regrets of $f_T\left(\mathbf{x};\boldsymbol{\theta}\right) = \exp\left\{\cos\left(\|\mathbf{x}\|_2\right) + \sin\left(\|\boldsymbol{\theta}\|_2\right)\right\}$ with $\mathcal{X} = [-1,1]^2$ and $\Theta = [-1,1]^3$. In each similar task, samples are generated by $f_s\left(\mathbf{x};\boldsymbol{\theta}\right) = \exp\left\{\cos\left(\|\mathbf{x}\|_2\right) + k_s\sin\left(\|\boldsymbol{\theta}\|_2\right)\right\}$, where $k_s$ is randomly selected from $\{1,2,\ldots,10\}$ with equal probability for $s = 1,2,\ldots,5$.