# OpenReview forum: "Contextual Gaussian Process Bandits with Neural Networks"
_NeurIPS.cc/2023/Conference — NeurIPS 2023 poster_

### Official Review · Reviewer_9ysH · 2023-07-05

**Soundness:** 3 good
**Presentation:** 3 good
**Contribution:** 3 good
**Rating:** 7
**Confidence:** 3

**Summary:**

The paper proposes extension of the Gaussian Process (GP) based contextual bandits to the contextual case, where the reward function is time and context dependent. The proposed method models the context dependency via a multi-output GP and its relationship to the actions via a neural network. The inner product of the outputs of the GP and the neural network determines the reward.

**Strengths:**

The paper is well-written, its language is clear, experiments thorough, results strong, and the theoretical analysis sound.

Despite the theoretical depth of the work, the paper studies two challenging and interesting use cases as bandit problems: queuing with LSTMs and pricing with graph convolutional neural nets. This gives a paper a solid standing. The graph application comes nicely together with the kernelized nature of the reward definition.

The results reported in Figures 3-6 are intriguing.

**Weaknesses:**

Algorithm 1 has some novel aspects such as the way mu and sigma are calculated but the methodological novelty is rather incremental.

**Questions:**

It does not look clear to me what the purpose of Section 3.3. What do we expect from studying the maximum information gain aspect?

Why is the proposed method not compared against NeuralUCB also in queuing and pricing tasks?

**Limitations:**

The paper does not discuss the limitations of the proposed approach.

---

> ### Author Rebuttal · Authors · 2023-08-09
>
>
> We thank the reviewer for the valuable time, questions and comments. We find them very helpful. Please find our response to the comments below.
>
> $\textbf{Comment 1: }$Algorithm 1 has some novel aspects such as the way mu and sigma are calculated but the methodological novelty is rather incremental.
>
> $\textbf{Response:}$ We thank the reviewer for the comment. We recognize the extensive work done on GP-based methods. However, the novelty of our proposed NN-AGP model lies in the innovative integration of both GP and NN models, which not only enhances the performance but also broadens the applicability of the bandit algorithm.
>
> In addition, our approach uniquely retains the GP structure regarding the decision variable $\mathbf{x}$, maintaining the key statistical properties and explicit uncertainty quantification inherent to GP. This structure ensures that bandit algorithms utilizing NN-AGP can seamlessly leverage the power of established GP-based acquisition functions designed for different settings and scenarios. As a case in point, while our main text focuses on the UCB acquisition function, we also delve into other acquisition functions in Section 6 of the supplements. In conclusion, we specifically retain the existing methodologies for GP while incorporating the NN model to enhance the bandit algorithm performance.
>
>
>
>
>
>
>
> $\textbf{Comment 2: }$It does not look clear to me what the purpose of Section 3.3. What do we expect from studying the maximum information gain aspect?
>
>
> $\textbf{Response:}$ We appreciate this constructive comment. In Section 3.2, we prove that the cumulative regret in terms of rewards of the bandit algorithm NN-AGP-UCB is upper-bounded by $\tilde{\mathcal{O}}\left(\sqrt{T\gamma_T}\right)$. Here $\gamma_T$ is the maximum information gain of the proposed model NN-AGP, and we study it in Section 3.3 to provide the upper bound on the cumulative regret as a function of the total number of rounds $T$.
>
>
> $\textbf{Comment 3: }$Why is the proposed method not compared against NeuralUCB also in queuing and pricing tasks?
>
> $\textbf{Response: }$We appreciate this constructive comment. In queuing and pricing tasks, the contextual variables are complex and represented by time series and graphs. We select the long short-term memory (LSTM) model and the graph convolutional neural network (GCN) to model the mappings from these contextual variables in our NN-AGP-UCB algorithm. In contrast, NeuralUCB (as well as NN-UCB) is designed for vector-valued contextual variables and employs fully connected networks (FCN). Therefore, NeuralUCB and NN-UCB are not applicable in queuing and pricing tasks because of the complex contextual variables.
>
>
>
>
> $\textbf{Comment 4: }$The paper does not discuss the limitations of the proposed approach.
>
> $\textbf{Response:}$ We appreciate this constructive comment. We will definitely follow the comment and enhance the discussion in the next iteration. One of the limitations of our work is the cold-start issue during the initial rounds. That is, there are no sufficient data for learning the neural network in NN-AGP. This issue can be addressed by the transfer learning technology [1]. That is, at the beginning of NN-AGP-UCB, we can retain some layers of the neural network in NN-AGP, which are previously learned in similar tasks. This incorporation with transfer learning is also mentioned in the conclusion section.
>
> To illustrate the transfer learning technology with NN-AGP-UCB, we conduct numerical experiments and present the results in $\textbf{Figure 1}$ of the document attached in the global response. Specifically, we consider an unknown reward function $f_{T}$ and we also have access to functions $f_{s},s=1,2,\ldots,5$ that have a similar structure with $f_T$. We first sample each $f_s$ for 50 or 100 rounds and learn an NN-AGP model with these samples. The NN component in NN-AGP helps to transfer the knowledge from $f_s$ to $f_T$. That is, during the initial rounds of NN-AGP-UCB with $f_T$, we first fix the input layer of the pretrained NN and update the remaining layers with the new data, which is a widely-used transfer learning method named freezing. Experimental results indicate that transfer learning from similar tasks helps to address the cold-start issue, and NN-AGP-UCB with/without transfer learning will converge to the similar regrets as the round increases.
>
>
>
>
> Another limitation is due to the computational cost of NN-AGP. As mentioned in the conclusion section, since NN-AGP retains a GP structure, it suffers from computational complexity with large data sets. For future work, we consider sparse NN-AGP to alleviate the computational burden; see also a discussion in Section 9 of the supplements. We will include a more detailed and explicit discussion on the limitations in the revised main text.
>
> $\textbf{References:}$
>
> $[1]$ Weiss, K., Khoshgoftaar, T.M. and Wang, D., 2016. A survey of transfer learning. Journal of Big data, 3(1), pp.1-40.

---

> > ### Comment · Reviewer_9ysH · 2023-08-16
> > **Answers satisfactory**
> >
> > Thanks for your clear answers to my questions. I keep my view that this is a solid piece of work and also my score unchanged.

---

> > > ### Author Response · Authors · 2023-08-18
> > >
> > > Thanks again for your time and comments!

---

### Official Review · Reviewer_ThqQ · 2023-07-06

**Soundness:** 2 fair
**Presentation:** 3 good
**Contribution:** 2 fair
**Rating:** 5
**Confidence:** 3

**Summary:**

This work studies the contextual bandit problem with the Gaussian process. Authors tried to use neural networks to learn reward functions and introduce one algorithm NN-AGP. Then conduct empirical evaluation and theoretical analysis for it.

**Strengths:**

Recently, the bandit community paid more and more attention to the neural network approximation. It is interesting to combine it with GP in bandits. Authors use extensive experiments to demonstrate its practicability and conduct rigorous regret analysis for it.

**Weaknesses:**

(1) This work is an extension of [37] but very similar is very high. The key difference from [37] is changing the reward function to the inner product of $g(\theta)$ and $p(x)$, where $g$ is the neural network function and $p$ is the Gaussian process. The optimization of $p$ and the UCB-based exploration is adapted from [37]. But I don't see very novel aspects from $g$ except that it can be replaced by different neural network functions. Therefore, I think the overall novelty of this paper is limited.

(2) The required assumption of regret analysis is strong. It requires that the space of $\theta$ is convex and compact, but it is known the parameter space of neural network is non-convex and not compact. Moreover, the analysis is based on [37] and [34,35] (the information gain part). Authors may want to discuss more about [34, 35] from analysis aspect.

**Questions:**

(1) In Figure 2, I am wondering why there is a big jump first and then starts dropping.
(2) How about the running time cost (portion) of GP in this algorithm?

**Limitations:**

Strong assumption and incremental novelty.

---

> ### Author Rebuttal · Authors · 2023-08-09
>
> We thank the reviewer for the valuable time, questions and comments. We find them very helpful. Please find our response to the comments below.
>
> $\textbf{Comment 1: }$This work is an extension of [37] but very similar is very high. The key difference from [37] is changing the reward function to the inner product of $g(\theta)$ and $p(x)$, where $g$ is the neural network function and $p$ is the Gaussian process. The optimization of $p$ and the UCB-based exploration is adapted from [37]. But I don't see very novel aspects from $g$ except that it can be replaced by different neural network functions. Therefore, I think the overall novelty of this paper is limited.
>
> $\textbf{Response:}$ We thank the reviewer for the comment. Please allow us to clarify more of the novelty and associated comparisonal advantage of our work compared to the literature. Our work considers the contextual Gaussian process bandit problem, which was also addressed in [1] ([37] in main text) as well. The novelty of our proposed NN-AGP model lies in the innovative integration of both GP and NN models, which partially addresses the challenge of pre-specifying an appropriate joint GP model for approximating the reward function. Based on the numerical experiments, our approach outperforms the existing GP-based bandit methods by specifying the data-driven kernel function through the lens of neural networks. In addition, by employing different structures of neural networks (e.g., graph neural network), our approach is applicable to diverse application scenarios where the contextual variable is represented in complex forms other than a vector. We hope that the above clarification may partially alleviate the potential concern.
>
> $\textbf{Comment 2: }$The required assumption of regret analysis is strong. It requires that the space of $\boldsymbol{\theta}$ is convex and compact, but it is known the parameter space of neural network is non-convex and not compact. Moreover, the analysis is based on [37] and [34,35] (the information gain part). Authors may want to discuss more about [34, 35] from analysis aspect.
>
> $\textbf{Response:}$ We agree with the reviewer's comment that the parameter space of neural network is non-convex and not compact. In fact,  we would like to clarify that parameter $\boldsymbol{\theta}$ in our work does not represent the neural network parameters. The parameter $\boldsymbol{\theta}$ denotes the contextual variable that is input to the reward function (Line 82, Page 3 in our manuscript).
>
> Compared to assumptions on the neural network parameters, the assumption of convexity and compactness on the set of contextual variables appears to be more common; see also [1]. Indeed, for NN-AGP-UCB, this assumption can be relaxed to that $\sup_{\boldsymbol{\theta}\in \Theta} \\{ \\{  |\sum\_{l=1}^m \boldsymbol{g}\_{l}(\boldsymbol{\theta})a\_{l,q}  |  \\}\_{q=1}^Q , \\{ |  \boldsymbol{g}\_{l}(\boldsymbol{\theta})  | \\}^m\_{l=1}   \\}$ and $\sup_{\boldsymbol{\theta}\in \Theta}||\boldsymbol{g}\left(\boldsymbol{\theta}\right)||_2^2$ exist.
>
>
>
>
>
> In addition, our theoretical analysis borrows ideas from [1] and [2], and we mention it in Section 8 of the supplements, considering the length of main text. We also include a comparison of our theoretical results with those in [1] in the supplements. We will discuss more about [1,2] from analysis aspect in the revised main text.
>
>
>
>
>
>
>
> $\textbf{Comment 3: }$In Figure 2, I am wondering why there is a big jump first and then starts dropping. (2) How about the running time cost (portion) of GP in this algorithm?
>
> $\textbf{Response:}$ We appreciate the opportunity to clarify the appearance of a ''big jump" in Figure 2 of the main text. In fact, the mean of average regrets of all the compared algorithms decrease as the number of rounds increases and there is no ''jump'' in the numerical results. The ''jump'' in Figure 2 is actually the overlap of the shadowed regions of the curves, which indicate the standard deviation of average regrets. In contrast with Figure 1, Figure 2 is associated with a function with higher-dimensional decision and contextual variables. Consequently, there's heightened uncertainty in the initial stages illustrated in Figure 2. We will clarify the figure in the revised manuscript.
>
>
> In terms of the running time, we record 1) the training time that constructs the surrogate model based on the historical data and 2) the execution time that selects the decision variable after the contextual variable is revealed. We record the time (seconds) for exactly one round in the 50-th, 100-th, and 300-th rounds. We take the first set of experiments in Section 4.1 as an example and present the results in $\textbf{Table 1}$ of the document attached in the global response.
>
>
> We notice that CGP-UCB is the most efficient in training time since it employs a pre-specified GP model which does not update during iterations. On the other hand, all the algorithms that involve NN require learning NN from data and longer training time than CGP-UCB. In terms of the execution time, NN-AGP-UCB requires similar time as CGP-UCB, since the selection of the decision variable of NN-AGP-UCB is based on GP as well. To sum up, the training time is largely due to learning NN from the data, while the execution time is for selecting a decision variable based on the GP model. We will add this information to the supplements. In addition, we consider sparse NN-AGP to alleviate the computational burden for future work; see also a discussion in Section 9 of the supplements.
>
>
>
> $\textbf{References:}$
>
> $[1]$ Krause, A. and Ong, C., 2011. Contextual gaussian process bandit optimization. Advances in neural information processing systems, 24.
>
> $[2]$ Srinivas, N., Krause, A., Kakade, S.M. and Seeger, M., 2009. Gaussian process optimization in the bandit setting: No regret and experimental design. arXiv preprint arXiv:0912.3995.

---

> > ### Comment · Reviewer_ThqQ · 2023-08-17
> >
> > Thanks for the response. I am wondering what is the complexity of $\gamma_T$ and how to bound it?

---

> > > ### Author Response · Authors · 2023-08-18
> > >
> > > We appreciate the opportunity to clarify the maximum information gain $\gamma_T$. Specifically, the upper bound of $\gamma_T$ depends on the kernel function used in the GP component of NN-AGP. We consider two general categories of kernel functions: polynomial eigendecay and exponential eigendecay (see details in Definition 1 on Page 6), which include the most commonly used kernel functions. For polynomial eigendecay, $\gamma_T = \mathcal{O} \left ( T^{\frac{1}{\alpha _p} }\log ^{1-\frac{1}{\alpha_p} }\left ( T \right )  \right )$, where $\alpha_p>1$ is a constant. For exponential eigendecay, $\gamma_T = \mathcal{O} \left ( \log ^{1+\frac{1}{\alpha_e} }\left ( T \right )  \right )$, where $\alpha_e>0$ is a constant. More detailed results on the upper bound of $\gamma_T$ are summarized as Theorem 2 on Page 6, and we will provide this simplified expression in the revised version. In order to provide this upper bound of maximum information gain, we first explore the Mercer decomposition and the eigenvalues associated with NN-AGP. Then, we analyze the upper bound of the maximum information gain through the eigenvalues. The details are postponed to Section 8.2 of the supplements considering the length of the main text.

---

> > > > ### Comment · Reviewer_ThqQ · 2023-08-19
> > > >
> > > > Thanks for the response. I have increased the score by 1, but I still have concerns about the methodology novelty and the complexity of $\gamma_T$.

---

> > > > > ### Author Response · Authors · 2023-08-20
> > > > >
> > > > > We really appreciate your time, comment and reading our response. It is encouraging to know that our responses have clarified some of your questions and comments. In the next revision, we plan to discuss the simplified sample complexity of $\gamma_T$ and provide more details as described in the response. We also plan to provide a detailed comparison of our method with existing ones in terms of computational time.

---

### Official Review · Reviewer_ZHPP · 2023-07-07

**Soundness:** 3 good
**Presentation:** 3 good
**Contribution:** 2 fair
**Rating:** 5
**Confidence:** 3

**Summary:**

This paper proposes a reward model, called the neural network-accompanied Gaussian process (NN-AGP) for solving contextual bandit problems where the space of contexts and the space of decision variables may be continuous. This model is an inner product of a neural network and a multi-output GP. The neural network captures the dependence of the reward function on the contextual variables while the GP is to model the mapping from the decision variable to the reward.  The authors propose the NN-AGP-UCB algorithm for this problem in the form of the upper confidence bound strategy. They derive the regret for NN-AGP-UCB as well as the maximum information gain in their setting. Finally, they provide the experiments to evaluate their proposed algorithm for complex reward functions, including those with time-varying dependence on sequenced and graph-structured contextual variables.




**Strengths:**

- The paper introduces a reward model for contextual Gaussian process bandits which is more general than the previous work by Krause el al [37] by employing a neural network to capture the dependence of the reward function on the contextual variables. This allows to use different neural networks appropriate for applications with diverse structures of contextual information. This is also demonstrated in their experimental results.
- The regret analysis and the upper bound of the maximum information gain are provided under this model.
- The experimental results on different structures of context information are also a strong point of this paper.


**Weaknesses:**

- As defined, the reward model is an inner product of a neural network (NN) and a multi-output GP. This reward structure seems not natural compared to the ones which are entirely either GP or NN. In addition, there are also many possible combinations of a NN and a GP to construct a reward model. Therefore, a more general model would be better.
- In the conclusion section, the authors claimed that the advantage of their approach is the approximation accuracy for the reward function and better performance on cumulative rewards/regrets. However, this is not correct. Using a NN to model reward in overparameterized regime allows the approximation accuracy for the reward function. Moreover, they said their approach has better performance on regrets. However, it is not clear which related works they are comparing with.
- It lacks the comparison of the regret bounds of the proposed algorithm and related algorithms like NeuralUCB, NeuralTS, and NeuralLinUCB which use entirely a NN, and algorithms which use entirely a GP to model the reward function.



**Questions:**

- Please see the questions in the Weaknesses section.
- It would be interesting if the authors take into consideration the influence of the proposed NN in their regret analysis, at least in the overparametrized regime.

**Limitations:**

Yes

---

> ### Author Rebuttal · Authors · 2023-08-09
>
> We thank the reviewer for the valuable time, questions and comments. We find them very helpful. Please find our response to the comments below.
>
> $\textbf{Comment 1: }$As defined, the reward model is an inner product of a neural network (NN) and a multi-output GP. This reward structure seems not natural compared to the ones which are entirely either GP or NN. In addition, there are also many possible combinations of a NN and a GP to construct a reward model. Therefore, a more general model would be better.
>
> $\textbf{Responses:}$ We completely agree with the reviewer on the many possible combinations of NN and GP. Our particular choice of combining a neural network and a multi-output GP as an inner product is motivated by the differentiation between the decision variable and the contextual variable. By doing so, we ensure an explicit GP expression for the decision variable $\mathbf{x}$ once the contextual variable is observed in each round. Such design not only facilitates explicit quantification of reward function approximation uncertainty, guiding decision variable selection, but also provides theoretical regret bound guarantees. While there are various ways to combine NN and GP, a more general model might compromise the explicitness of the GP expression for $\mathbf{x}$. Our approach leverages the strong flexibility and approximation accuracy of NN while preserving statistical properties of GP, and appears to have reliable numerical performance. We plan to add a detailed remark to reflect this helpful comment.
>
>
> $\textbf{Comment 2: }$In the conclusion section, the authors claimed that the advantage of their approach is the approximation accuracy for the reward function and better performance on cumulative rewards/regrets. However, this is not correct. Using a NN to model reward in overparameterized regime allows the approximation accuracy for the reward function. Moreover, they said their approach has better performance on regrets. However, it is not clear which related works they are comparing with.
>
> $\textbf{Response:}$ We thank the reviewer for the comment. We would like to clarify that the advantage of approximation accuracy of NN-AGP is over the joint GP model. We also admit that using overparametrized NN to model the reward function generally leads to better approximation accuracy provided sufficient data. However, there are two challenges of the algorithms that are entirely based on NN. First, there is no explicit uncertainty quantification of NN approximation. Thus, addressing the exploration-exploitation trade-off requires further approximation, which affects the performance of bandit algorithms. Second, the overparametrized NN requires sufficient data to train, which might not be applicable in bandit problems, especially in initial rounds. Therefore, our NN-AGP-UCB algorithm achieves better performance on regrets than both the algorithms that entirely rely on GP (CGP-UCB) and the algorithms that entirely rely on NN (NeuralUCB and NN-UCB), which are also supported by experimental results. We will revise the conclusion section to provide a more clear statement and more careful descriptions.
>
>
>
> $\textbf{Comment 3: }$It lacks the comparison of the regret bounds of the proposed algorithm and related algorithms like NeuralUCB, NeuralTS, and NeuralLinUCB which use entirely a NN, and algorithms which use entirely a GP to model the reward function.
>
>
>
>
>
> $\textbf{Response:}$ We thank the reviewer for the comment. In terms of the algorithm which uses entirely a GP model, CGP-UCB [1], we postpone the comparison to the supplements (the end of Section 8.1 on Page 18). Specifically, NN-AGP-UCB has a same bound of $\tilde{\mathcal{O}}\left(\sqrt{T\gamma_T}\right)$ as CGP-UCB, but is superior when the contextual variable dimension is high. In terms of the algorithms which use entirely a NN, we note that NeuralUCB [2], Neural TS [3] and Neural LinUCB [4] all consider the scenarios when the decision variable $\mathbf{x}$ is selected from a finite set. In comparison, we consider that $\mathbf{x}$ is selected from a continuous set (Line 87, Page 2). When performed on a finite feasible set $\mathcal{X}$, our NN-AGP-UCB also has a regret bound of $\tilde{\mathcal{O}}\left(\sqrt{T\gamma_T}\right)$, where $\gamma_T$ further depends on the kernel function of the GP component used in NN-AGP. When the kernel function has an exponential eigendecay (see Definition 1 in Line 242, Page 6), NN-AGP-UCB has a regret bound of $\tilde{\mathcal{O}}\left(\sqrt{T}\right)$, matching the regret bound of NeuralUCB, Neural TS and Neural LinUCB as well.
>
>
>
>
>
> $\textbf{Comment 4: }$It would be interesting if the authors take into consideration the influence of the proposed NN in their regret analysis, at least in the overparametrized regime.
>
> $\textbf{Response:}$ We thank the reviewer for the constructive suggestion. In Section 7 of the supplements, we provide an algorithm NN-AGP-UCB+, which accounts for the neural network approximation error and performs more conservatively than NN-AGP-UCB. We also provide the regret bound of NN-AGP-UCB+. We agree that the overparametrized regime would be an inspiring direction for future work and the neural tangent kernel (NTK) can be employed to analyze the regret.
>
>
>
>
> $\textbf{References:}$
>
> $[1]$ Krause, A. and Ong, C., 2011. Contextual gaussian process bandit optimization. Advances in neural information processing systems, 24.
>
> $[2]$ Zhou, D., Li, L. and Gu, Q., 2020, November. Neural contextual bandits with ucb-based exploration. In International Conference on Machine Learning (pp. 11492-11502). PMLR.
>
> $[3]$ ZHANG, W., Zhou, D., Li, L. and Gu, Q., 2020, October. Neural Thompson Sampling. In International Conference on Learning Representations.
>
> $[4]$ Xu, P., Wen, Z., Zhao, H. and Gu, Q., 2020. Neural contextual bandits with deep representation and shallow exploration. arXiv preprint arXiv:2012.01780.

---

> > ### Comment · Reviewer_ZHPP · 2023-08-22
> >
> > Thanks for your rebuttal. I keep my current score.

---

> > > ### Author Response · Authors · 2023-08-22
> > >
> > > Thank you again for your time and comments.

---

### Official Review · Reviewer_jVXy · 2023-07-13

**Soundness:** 3 good
**Presentation:** 4 excellent
**Contribution:** 3 good
**Rating:** 6
**Confidence:** 3

**Summary:**

This work proposes a NN accompanied GP model. It leverages NN to approximate the unknown reward function regarding the context variable and maintains a GP with the decision variable. By introducing NN, the proposed method offers a better approximation accuracy. Theoretical implications, including maximum information gain and regret bounds, are provided for the proposed method. The effectiveness of the proposed method is also supported by empirical evaluation on both a synthetic and real-world dataset.

**Strengths:**

1. A well-motivated and well-designed algorithm with solid theoretical analysis of the statistical properties of the proposed model.
2. The proposed model is flexible enough to be used together with different types of NN models.
3. Empirical evaluation of a diverse set of tasks is included and shows the promising effectiveness of the proposed method.

**Weaknesses:**

1. Potential limitations of the model: Although NN could help enable a better approximation accuracy on the reward function regarding the context variable, the training of NN models may make the model computationally prohibitive. In addition, NN typically works well with a large amount of data, which may cause a cold-start issue when the proposed model is used.

2. May need to include more metrics in the empirical evaluation, such as computation cost and prediction latency.

**Questions:**

Can the authors provide corresponding justifications or empirical evidence on the potential limitations of the proposed method mentioned in the weaknesses section?

I am willing to adjust my rating based on the authors' response to my question above.

**Limitations:**

See weaknesses.

---

> ### Author Rebuttal · Authors · 2023-08-09
>
>
> We thank the reviewer for the valuable time, questions and comments. We find them very helpful. Please find our response to the comments below.
>
>
> $\textbf{Comment 1: }$Potential limitations of the model: Although NN could help enable a better approximation accuracy on the reward function regarding the context variable, the training of NN models may make the model computationally prohibitive. In addition, NN typically works well with a large amount of data, which may cause a cold-start issue when the proposed model is used.
>
> $\textbf{Response:}$ We admit that incorporating NN into bandit problems generally requires sufficient data to approximate the unknown reward function. Thus, the cold-start issue is brought to, in principle, all bandit algorithms that use NN. Compared with the algorithms that fully rely on NN (e.g., NeuralUCB and NN-UCB), our NN-AGP-UCB actually suffers less from the cold-start issue, which is supported by numerical results in Section 4.1. The reason is that, in existing NN-based bandit algorithms, NN is responsible for approximating the entire reward function. In comparison, for the NN-AGP model, NN is specially used for approximating the mapping from the contextual variable to the reward function, while the approximation regarding the decision variable is supported by GP. It has been widely accepted that GP generally requires less data than NN in practical applications, and therefore NN-AGP helps to ease the cold-start issue.
>
> Moreover, to further address the cold-start issue, the transfer learning technology [1] can be incorporated as mentioned in the conclusion section. We conduct numerical experiments and present the results in $\textbf{Figure 1}$ of the document attached in the global response. Specifically, we consider an unknown reward function $f_{T}$ and we also have access to functions $f_{s},s=1,2,\ldots,5$ that have a similar structure with $f_T$. We first sample each $f_s$ for 50 or 100 rounds and learn an NN-AGP model with these samples. The NN component in NN-AGP helps to transfer the knowledge from $f_s$ to $f_T$. That is, during the initial rounds of NN-AGP-UCB with $f_T$, we first fix the input layer of the pretrained NN and update the remaining layers with the new data, which is a widely-used transfer learning method named freezing.
>
> Experimental results indicate that transfer learning from similar tasks helps to address the cold-start issue, and NN-AGP-UCB with/without transfer learning will converge to the similar regrets as the round increases. We also not that, to the best of our knowledge, there has not been extensive work on transfer learning with NN-based bandit algorithms. These NN-based bandit algorithms largely rely on the neural tangent kernel (NTK) to address the exploration-exploitation trade-off when selecting the decision variable. However, it remains an open question on how to transfer the knowledge between different domains with NTK. In comparison, the exploration-exploitation trade-off in NN-AGP-UCB is supported by GP, and existing transfer learning technologies with NN can be easily adapted into our algorithm. Other methodologies for addressing cold-start in learning NN in an online setting can also be employed; see [2,3]. We will add a  remark addressing this to our revised introduction.
>
>
>
>
>
>
>
> $\textbf{Comment 2: }$May need to include more metrics in the empirical evaluation, such as computation cost and prediction latency.
>
>
> $\textbf{Response:}$ We appreciate this comment. We record 1) the training time that constructs the surrogate model based on the historical data and 2) the execution time that selects the decision variable after the contextual variable is revealed. We record the time (seconds) for exactly one round in the 50-th, 100-th, and 300-th rounds. We take the first set of experiments in Section 4.1 as an example and present the results in $\textbf{Table 1}$ of the document attached in the global response.
>
>
> We notice that CGP-UCB is the most efficient in training time since it employs a pre-specified GP model which does not update during iterations. The training procedure of CGP-UCB only requires matrix operations, which can be implemented efficiently. On the other hand, all the algorithms that involve NN require learning NN from data and longer training time than CGP-UCB. In terms of the execution time, NN-AGP-UCB requires similar time as CGP-UCB, since the selection of the decision variable of NN-AGP-UCB is based on GP as well. We also note that both NN-UCB and NeuralUCB are initially designed for finite selections of decision variables. Thus, the computational cost of the execution time is largely due to searching for the optimal decision variable from the discretized feasible set. We will add this information to the supplements. In addition, we consider sparse NN-AGP to alleviate the computational burden for future work; see also a discussion in Section 9 of the supplements.
>
>
> $\textbf{References:}$
>
> $[1]$ Weiss, K., Khoshgoftaar, T.M. and Wang, D., 2016. A survey of transfer learning. Journal of Big data, 3(1), pp.1-40.
>
> $[2]$ Wei, J., He, J., Chen, K., Zhou, Y. and Tang, Z., 2017. Collaborative filtering and deep learning based recommendation system for cold start items. Expert Systems with Applications, 69, pp.29-39.
>
>
> $[3]$ Wolfe, C.R. and Kyrillidis, A., 2022. Cold Start Streaming Learning for Deep Networks.arXiv preprint arXiv:2211.04624.

---

> > ### Comment · Reviewer_jVXy · 2023-08-18
> > **My questions have been addressed**
> >
> > I thank the authors for providing additional evidences to further support their claims and resolve the concerns. My questions have been addressed accordingly. The second weakness pointed out can be removed. I will keep my score as 6: Weak Accept.

---

> > > ### Author Response · Authors · 2023-08-20
> > >
> > > Thank you again for your time and comments. In the revised version, we will include a detailed discussion on computation cost and prediction latency as suggested.

---

### Author Rebuttal · Authors · 2023-08-09

We would like to thank the reviewers for their review and for providing us with valuable comments to improve our work. We have treated every comment to our best efforts. In addition to the point-to-point responses to each of the reviewer, we also attach a document in this global response, containing 1) a table recording training/ execution time and 2) a figure recording the average regrets in a transfer learning setting.

We record 1) the mean of training time that constructs the surrogate model based on the historical data and 2) the mean of execution time that selects the decision variable after the contextual variable is revealed. We record the time (seconds) for exactly one round in the 50-th, 100-th, and 300-th rounds. We take the first set of experiments in Section 4.1 as an example and present the results in $\textbf{Table 1}$ in the attached document.

We also include the experimental results of transfer learning with NN-AGP-UCB in $\textbf{Figure 1}$ of the attached document. Specifically, we consider an unknown reward function $f_{T}$ and we also have access to unknown functions $f_{s},s=1,2,\ldots,5$ that have a similar structure with $f_T$. We first sample each $f_s$ for 50 or 100 rounds and learn an NN-AGP model with these samples. The NN component in NN-AGP helps to transfer the knowledge from $f_s$ to $f_T$. That is, during the initial rounds of NN-AGP-UCB with $f_T$, we first fix the input layer of the pretrained NN and update the remaining layers with the new data, which is a widely-used transfer learning method named freezing.

---

### Decision · Program_Chairs · 2023-09-21

**Decision:**

Accept (poster)

**Comment:**

A good paper that incorporates neural networks into bandits, hence expanding its applicability and practical use. Please incorporate the reviewers' comments and suggestions in the camera-ready version.